# FlyPrompt: Brain-Inspired Random-Expanded Routing with Temporal-Ensemble Experts for General Continual Learning

**Hongwei Yan**[1] **Guanglong Sun**[1] **Kanglei Zhou**[2] **Qian Li**[3] **Liyuan Wang**[2*] **Yi Zhong**[1*]

[1]School of Life Sciences, IDG/McGovern Institute for Brain Research, Tsinghua University
[2]Department of Psychological and Cognitive Sciences, Tsinghua University
[3]School of Medicine, Shenzhen Campus of Sun Yat-Sen University
{yanhw22,sgl23}@mails.tsinghua.edu.cn, liqian255@mail.sysu.edu.cn
{zhoukanglei, liyuanwang, zhongyithu}@tsinghua.edu.cn

## Abstract

General continual learning (GCL) challenges intelligent systems to learn from single-pass, non-stationary data streams without clear task boundaries. While recent advances in continual parameter-efficient tuning (PET) of pretrained models show promise, they typically rely on multiple training epochs and explicit task cues, limiting their effectiveness in GCL scenarios. Moreover, existing methods often lack targeted design and fail to address two fundamental challenges in continual PET: how to allocate expert parameters to evolving data distributions, and how to improve their representational capacity under limited supervision. Inspired by the fruit fly's hierarchical memory system characterized by sparse expansion and modular ensembles, we propose FlyPrompt, a brain-inspired framework that decomposes GCL into two subproblems: *expert routing* and *expert competence improvement*. FlyPrompt introduces a randomly expanded analytic router for instance-level expert activation and a temporal ensemble of output heads to dynamically adapt decision boundaries over time. Extensive theoretical and empirical evaluations demonstrate FlyPrompt's superior performance, achieving up to 11.23%, 12.43%, and 7.62% gains over state-of-the-art baselines on CIFAR-100, ImageNet-R, and CUB-200, respectively. Our source code is available at FlyGCL.

## 1 Introduction

General Continual Learning (GCL) (Buzzega et al., 2020; De Lange et al., 2021), aims to equip intelligent systems with the ability to learn continuously from non-stationary, single-pass data streams, where tasks may not have clear boundaries and can evolve over time. Unlike traditional Continual Learning (CL) (Wang et al., 2024b; Parisi et al., 2019), which assumes well-defined task boundaries and multiple training epochs, GCL presents a much more challenging problem, as it requires rapid adaptation, robust knowledge retention, and efficient resource usage under conditions of limited supervision and task ambiguity (Fig. 1). The ability to effectively tackle GCL has profound implications for real-world applications such as autonomous agents and personal assistants, where systems must learn from dynamic environments without clear task definitions.

Recent advances[1] in parameter-efficient tuning (PET) of pretrained models (PTMs) have shown promise in CL (Wang et al., 2022d;c; Smith et al., 2023), but they still face fundamental limitations under GCL conditions. Such methods introduce task-specific prompt experts to adapt PTMs incrementally, and typically rely on clear task cues and sufficient gradient updates to allocate and train expert modules (Wang et al., 2024a; 2022b). However, those assumptions no longer hold in GCL (Koh et al., 2021; Moon et al., 2023). We therefore identify two fundamental challenges that remain unresolved: (1) how to dynamically route inputs to appropriate experts without task labels or iterative training, and (2) how to ensure that each expert maintains strong and adaptive representations under sparse and imbalanced supervision. Both remain non-trivial and underexplored.

---

*Corresponding Authors: L. Wang and Y. Zhong.
[1]Due to the page limit, we present a comprehensive summary of related work in Appendix B.

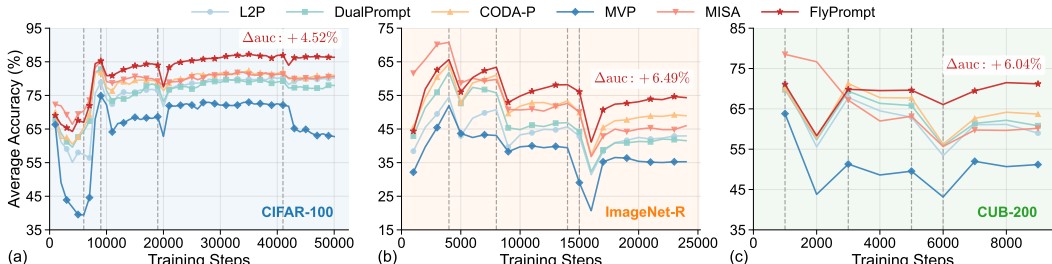

Figure 1: Any-time average accuracy of GCL methods over three datasets using Sup-21K. Dashed lines indicate task transition. $\Delta$auc, the improvement of area-under-curve score by FlyPrompt.

The complexity of GCL has also been extensively studied in biological systems, where organisms have evolved efficient strategies for lifelong learning in dynamic environments. The fruit fly *Drosophila* provides a compelling model: despite having fewer than 100,000 neurons, it exhibits robust memory consolidation, context-aware behavior, and stable learning under minimal supervision (Davis, 2023; Li et al., 2020; Modi et al., 2020; Owald & Waddell, 2015). These capabilities are largely attributed to the mushroom body, a central brain structure that encodes sensory inputs via sparse random projections and organizes learning into modular, hierarchical compartments (Aso et al., 2014b; Dasgupta et al., 2017; Wang et al., 2021; 2022a; 2023b; Wang & Li, 2025) (see Fig. 4*(Left)*). Projection neurons (PNs) from the antennal lobe connect randomly to Kenyon cells (KCs) in mushroom body, yielding high-dimensional sparse codes that support input separation and routing even under noisy or overlapping conditions (Turner et al., 2008; Honegger et al., 2011). Furthermore, different KC subregions exhibit plasticity on distinct timescales (Aso et al., 2014a; Aso & Rubin, 2016), enabling both rapid adaptation and long-term consolidation (Cervantes-Sandoval et al., 2013; Bouzaiane et al., 2015; Wang & Li, 2025). These mechanisms closely mirror the goals of GCL, offering principled inspiration for tackling its core challenges.

Building upon these neurobiological principles and our preliminary analysis in Sec. 2.2, we propose to decompose the GCL challenges into two essential subproblems: (1) *expert routing*, which aims to assign each input to an appropriate subnetwork (expert) under unknown and shifting task boundaries; and (2) *expert competence improvement*, which seeks to enhance the robustness and adaptability of each expert given limited training and imbalanced class exposure. To address these challenges, we introduce **FlyPrompt**, a brain-inspired framework that integrates two key components: (i) a *Random Expanded Analytic Router* (REAR) that mimics the fruit fly's sparse expansion circuit to rapidly assign inputs to experts in a forward-only, closed-form manner; and (ii) a *Task-wise Experts with Temporal Ensemble ($TE^2$)* that captures knowledge across multiple time scales using exponential moving averages, mirroring the compartmental consolidation observed in the mushroom body.

FlyPrompt is supported by both theoretical analysis and empirical validation. Across diverse GCL benchmarks, including CIFAR-100, ImageNet-R, and CUB-200, it consistently outperforms state-of-the-art CL and GCL methods, achieving accuracy improvements of up to 11.23%, 12.43%, and 7.62%, respectively. By integrating biologically grounded design with principled algorithmic structure, FlyPrompt offers an interdisciplinary perspective on addressing the core challenges of GCL and also exemplifies the potential of the emerging field of NeuroAI (Zador et al., 2023).

## 2 PRELIMINARIES

In this section, we formulate GCL, and then evaluate PET methods in an instantiated GCL scenario.

### 2.1 PROBLEM FORMULATION

In CL, a model learns sequential tasks $t \in \{1, \cdots, T\}$, each associated with a dataset $\mathcal{D}_t = (\boldsymbol{x}_t, y_t)$ where $\boldsymbol{x}_t \in \mathcal{X}_t$ and $y_t \in \mathcal{Y}_t$. The model comprises a backbone $f_{\boldsymbol{\theta}}(\cdot)$ and an output head $g_{\boldsymbol{\psi}}(\cdot)$, which together produce predictions $\hat{y} = g_{\boldsymbol{\psi}}(f_{\boldsymbol{\theta}}(\boldsymbol{x}))$. The objective is to learn a unified mapping from input domains $\mathcal{X} = \bigcup_t \mathcal{X}_t$ to label spaces $\mathcal{Y} = \bigcup_t \mathcal{Y}_t$. Classical CL settings impose structural assumptions on the input or label space. Domain-incremental learning (DIL) assumes disjoint input domains with a shared label space ($\mathcal{X}_i \cap \mathcal{X}_j = \emptyset$, $\mathcal{Y}_i = \mathcal{Y}_j$), while task-incremental and class-incremental learning (TIL, CIL) assume disjoint label spaces ($\mathcal{Y}_i \cap \mathcal{Y}_j = \emptyset$), with TIL additionally

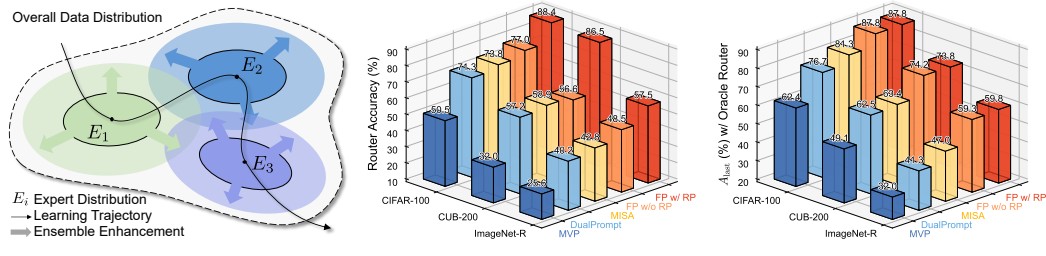

(a) GCL as Multi-Expert Learning.     (b) Prompt router accuracy.     (c) Accuracy with oracle router.

Figure 2: Empirical analysis of GCL. (a) A schematic illustration of GCL viewed as multi-expert collaboration. (b) Prompt selection accuracy for methods with explicit expert routing designs. (c) Final average accuracy ($A_{\text{last}}$, ↑) when using a test-time oracle to provide the correct prompt identity. Results evaluated across three benchmarks with Sup-21K. FP, FlyPrompt. RP, Random Projection.

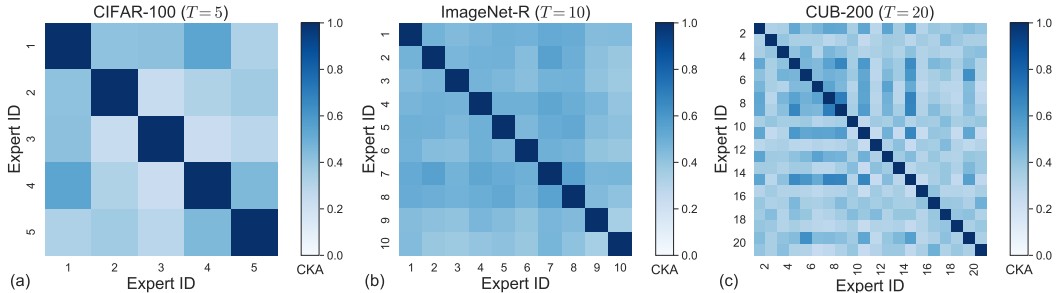

Figure 3: CKA similarity of feature representations between experts of MVP on three datasets.

providing task identity at test time (Van de Ven & Tolias, 2019). Under these assumptions, the learning objective can be decomposed into two orthogonal subproblems: *task identity prediction* (TIP), which selects an appropriate task-specific module, and *within-task prediction* (WTP), which performs classification under the selected module (Kim et al., 2022; Wang et al., 2023a).

GCL, however, lifts these assumptions and operates under substantially more challenging conditions. Tasks arrive as a one-pass data stream, and their label spaces may overlap with non-negligible probability: $\forall i \neq j, P(\mathcal{Y}_i \cap \mathcal{Y}_j \neq \emptyset) > 0$ (Koh et al., 2021). This entangles inter-task and intra-task interference, undermining the TIP–WTP orthogonality. Especially when using pretrained backbones, the strong priors encoded in PTMs already bias prediction before adaptation, causing task identity and class discrimination to co-evolve (Wang et al., 2023a; 2024a). Moreover, GCL introduces additional difficulties such as severe intra-task class imbalance (e.g., long-tailed distributions) and limited training iterations. These problems are compounded by memory constraints (Buzzega et al., 2020; De Lange et al., 2021), where storing past data is restricted or disallowed.

A representative instantiation of GCL is Si-Blurry (Moon et al., 2023), which explicitly partitions the global label space $\mathcal{Y}$ into a disjoint subset $\mathcal{Y}^{\text{D}}$ and a blurry subset $\mathcal{Y}^{\text{B}}$, with $\mathcal{Y} = \mathcal{Y}^{\text{D}} \cup \mathcal{Y}^{\text{B}}$ and $\mathcal{Y}^{\text{D}} \cap \mathcal{Y}^{\text{B}} = \emptyset$. The *disjoint class ratio* $r_{\text{D}} = |\mathcal{Y}^{\text{D}}|/|\mathcal{Y}|$ controls the proportion of task-specific classes, while the *blurry sample ratio* $r_{\text{B}}$ determines how frequently classes in $\mathcal{Y}^{\text{B}}$ reappear across tasks. This flexible design captures the stochasticity and heterogeneity of GCL, which has been validated in recent theoretical and empirical work (Mi et al., 2020; Zhuang et al., 2024; Kang et al., 2025). We therefore adopt Si-Blurry as the default GCL benchmark (see Appendix D for discussion about the task/session boundary information and our empirical rationality of using Si-Blurry).

## 2.2 ANALYSIS OF GCL METHODS WITH EXPERTS

Recent CL and GCL methods increasingly adopt PET techniques on top of PTMs. These methods can be seen as lightweight extensions of architecture-based CL (Zhu et al., 2021; Wang et al., 2023b), where instead of expanding full networks, they introduce trainable modules $\boldsymbol{p}$ (e.g., adapters, prompts, and LoRA) that act as semantic-aware adaptation experts to give instructed outputs $f_{\boldsymbol{\theta}}(\boldsymbol{x}; \boldsymbol{p})$. A common strategy is to maintain a pool of such experts and design a router to assign inputs to the appropriate ones. However, most existing methods, such as L2P (Wang et al., 2022d), DualPrompt (Wang et al., 2022c), MVP (Moon et al., 2023), CODA-P (Smith et al., 2023), and MISA (Kang et al., 2025), train these routing functions synchronously with the stream of incoming

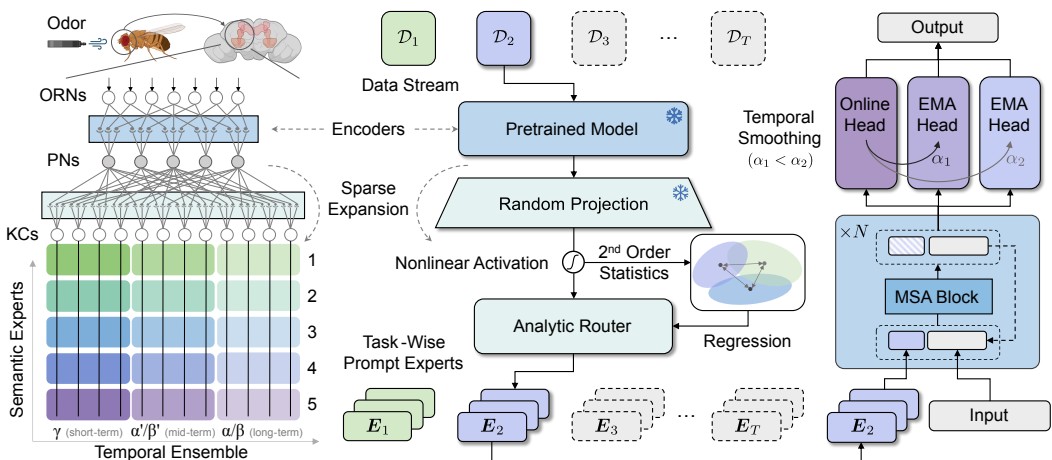

Figure 4: Method overview. Inspired by the fruit fly's olfactory memory system *(Left)*, FlyPrompt incorporates a random-expanded analytic router *(Middle)* and temporal ensemble-based experts *(Right)*. ORNs, olfactory receptor neurons. PNs, projection neurons. KCs, Kenyon cells.

data, making them vulnerable to distributional shifts and limited iterations. These issues are especially pronounced in GCL, where blurry task boundaries and online constraints prohibit iterative tuning, and class imbalance further weakens the quality of per-task representations.

Based on this observation, we propose to decompose the GCL problem into two critical subproblems: **expert routing**, which determines which expert to assign to each input; and **expert competence**, which enhances the quality and robustness of each expert's representation. Compared to the classical TIP-WTP formulation for CL with clearly segmented tasks, GCL's blurry structure and overlapping class distributions make semantic-level experts a more suitable abstraction for adaptation (see Fig. 2a). Moreover, since the same class may appear in multiple tasks, the correspondence between classes and experts is inherently one-to-many, which further complicates routing.

To better understand these challenges, we conduct a preliminary empirical study from a multi-expert collaboration (details in Sec. 4.1). We first evaluate the routing accuracy of methods that explicitly predict expert identity (e.g., DualPrompt, MVP, MISA, and our FlyPrompt) after all GCL training tasks. A prediction is considered correct if the selected expert belongs to the set of experts previously trained on the true label $y_t$, acknowledging the overlap across tasks. As shown in Fig. 2b, existing routers based on similarity or contrastive losses still exhibit considerable limitations in expert selection. Next, we evaluate the final average accuracy under an oracle router that always selects a correct expert for each input (Fig. 2c). The results reveal that, even with perfect routing, previous methods still exhibit inferior performance, highlighting a second bottleneck: the limited competence of individual experts. In a PTM-based context, such competence depends not only on the representation space shaped by each expert module, but also on how well the output head can maintain consistent decision boundaries over time; even when an early expert's encoder is frozen, a single head that keeps adapting to later data can gradually become misaligned with its fixed representation. To verify this point, our analysis of expert-specific representations using centered kernel alignment (CKA; Appendix F.4) in Fig. 3 confirms that experts indeed specialize in distinct feature subspaces, underscoring the need for accurate expert assignment. Together with the observed degradation under an oracle router, these results support decomposing GCL into the interacting subproblems above, and clarify that expert competence must account for both representation quality and decoding robustness.

## 3 FLYPROMPT: A BRAIN-INSPIRED GCL APPROACH

In this section, we propose **FlyPrompt**, an innovative brain-inspired approach designed to tackle the key challenges of GCL by explicitly improving **expert routing** and **expert competence**. As shown in Fig. 4, FlyPrompt consists of two core components: (i) a *Random Expanded Analytic Router* (REAR) that employs fixed random projections and closed-form updates to assign inputs to experts, inspired by the sparse expansion circuits in fruit flies, and (ii) *Task-wise Experts with Temporal Ensemble (TE$^2$)* that adaptively refine class boundaries over time to improve expert-level performance, reflecting modularized ensembles architecture in fruit flies' neural systems.

## 3.1 Random Expanded Analytic Router

Recent studies in CL have explored the use of Random Projection (RP) to construct forward-only learners with desirable properties such as rapid adaptation and immunity to catastrophic forgetting (Zhuang et al., 2022; 2024; McDonnell et al., 2024). When combined with nonlinear activation, RP can significantly improve the linear separability of input features by capturing high-order interactions without backpropagation. Notably, this mechanism aligns closely with the olfactory system of the fruit fly, where projection neurons (PNs) connect sparsely and randomly to a high-dimensional population of Kenyon cells (KCs) in the mushroom body, achieving a nearly 40-fold expansion. Global inhibition then enforces sparsity, allowing robust and efficient pattern separation (Dasgupta et al., 2017; Honegger et al., 2011). This neurobiological design inspires REAR, which mimics the biologically grounded random expansion to enable efficient, non-iterative expert selection in GCL.

Concretely, given a pretrained backbone $f_{\boldsymbol{\theta}}(\cdot)$ that maps input $\boldsymbol{x}$ to an embedding $\boldsymbol{h} = f_{\boldsymbol{\theta}}(\boldsymbol{x}) \in \mathbb{R}^d$ of dimension $d$, we apply a fixed RP followed by a nonlinear activation:

$$\boldsymbol{\varphi}(\boldsymbol{x}) = \sigma\left(f_{\boldsymbol{\theta}}(\boldsymbol{x})\boldsymbol{R}\right) = \sigma(\boldsymbol{h}\boldsymbol{R}) \in \mathbb{R}^M, \tag{1}$$

where $\boldsymbol{R} \in \mathbb{R}^{d \times M}$ is a random matrix with $R_{i,j} \sim \mathcal{N}(0,1)$, $M > d$, and $\sigma(\cdot)$ is an element-wise activation function (e.g., ReLU). The resulting feature $\boldsymbol{\varphi}(\boldsymbol{x})$ is sparse and high-dimensional.

During online training, we associate each task $t$ with a corresponding expert $E_t$. For each incoming batch $\mathcal{B}_i \subset \mathcal{D}_t$ of size $B$, we compute the projected features $\boldsymbol{\Phi}_i \in \mathbb{R}^{B \times M}$, whose row vectors are $\{\boldsymbol{\varphi}(\boldsymbol{x})^\top | (\boldsymbol{x}, y) \in \mathcal{B}_i\}$, and update two statistics: the *Gram matrix* $\boldsymbol{G} \in \mathbb{R}^{M \times M}$ capturing second-order feature correlations; and the *prototype matrix* $\boldsymbol{Q} \in \mathbb{R}^{M \times T}$ storing expert-wise feature sums:

$$\boldsymbol{G} \leftarrow \boldsymbol{G} + \boldsymbol{\Phi}_i^\top \boldsymbol{\Phi}_i, \qquad \boldsymbol{Q} \leftarrow \boldsymbol{Q} + \boldsymbol{\Phi}_i^\top \boldsymbol{C}_t, \tag{2}$$

where $\boldsymbol{C}_t \in \mathbb{R}^{B \times T}$ whose row vectors are the same one-hot embedding $\boldsymbol{c}_t \in \{0,1\}^T$ for expert $E_t$. We then construct a router matrix $\boldsymbol{U} \in \mathbb{R}^{T \times M}$ by minimizing the following objective:

$$\mathcal{L}(\boldsymbol{U}) = \sum_{t=1}^T \sum_{(\boldsymbol{x}_t, y) \in \mathcal{D}_t} \left\| \boldsymbol{\varphi}(\boldsymbol{x}_t)\boldsymbol{U}^\top - \boldsymbol{c}_t \right\|_2^2 + \lambda \|\boldsymbol{U}\|_F^2, \tag{3}$$

where $\lambda > 0$ is the regularization parameter. This objective encourages the router to map samples from task $t$ to the corresponding expert $E_t$ while maintaining numerical stability. Using the accumulated statistics $\boldsymbol{G}$ and $\boldsymbol{Q}$, the closed-form solution to this optimization problem is given by:

$$\widehat{\boldsymbol{U}}^\top = (\boldsymbol{G} + \lambda \boldsymbol{I})^{-1} \boldsymbol{Q}. \tag{4}$$

The calculation of $\widehat{\boldsymbol{U}}$ is only needed once upon evaluation, therefore this optimization process is efficient and lightweight compared to gradient-based routing mechanisms. At inference time, the routing score $\boldsymbol{s}$ and selected expert $\hat{E}$ for an input $\boldsymbol{x}$ given router $\widehat{\boldsymbol{U}}$ is computed as:

$$\boldsymbol{s}(\boldsymbol{x}) = \boldsymbol{\varphi}(\boldsymbol{x})\widehat{\boldsymbol{U}}^\top \in \mathbb{R}^T, \qquad \hat{E}(\boldsymbol{x}) = \arg\max_{t \leq T} s_t(\boldsymbol{x}). \tag{5}$$

This routing mechanism is efficient, biologically motivated, and requires no gradient updates, making it well-suited for GCL's online, single-pass constraints. Unlike prior methods based on random expanded features, such as RanPAC (McDonnell et al., 2024) or ACIL (Zhuang et al., 2022), which apply closed-form ridge regression directly for final classification on fixed representations, REAR uses random projections solely for instance-level expert routing while keeping each expert's prompts and heads fully trainable. Empirical comparison between the analytic router (REAR) and analytic classifier (RanPAC) under GCL benchmarks is shown in Appendix Tab. 13. And we further demonstrate the superiority of REAR upon alternative routing strategies in Appendix Tab. 17. We then summarize the core theoretical guarantee that explains why REAR yields reliable routing in the expanded sparse feature space. Full assumptions and proofs are included in Appendix E.1.

**Theorem 1** (REAR, informal). *With high probability over the random expansion and the data stream, the population excess risk of the ridge router learned from online statistics admits the following decomposition:*

$$\mathcal{R}(\widehat{\boldsymbol{U}}) - \mathcal{R}(\boldsymbol{U}^\star) \lesssim \sqrt{\log(N)/M} + (\sqrt{N}\,\lambda)^{-1} + \lambda,$$

*for suitable universal constants. Therefore, by increasing the expansion dimension $M$ and the number of samples $N$, and choosing the regularization parameter $\lambda$ to balance estimation error and bias, the population excess risk (and, under a fixed margin assumption on expert scores; see Appendix E.1, the misrouting probability) can be made arbitrarily small.*

**Interpretation.** The first term reflects the approximation error due to finite random features; increasing $M$ improves the expressive power of the expansion. The second term captures the variance arising from finite data, which diminishes as $N$ grows or $\lambda$ increases. The third term represents the bias introduced by ridge regularization, which stabilizes learning but limits expressiveness if it is too large. In practice, this decomposition implies that robust and forward-only routing can be achieved by employing sufficiently rich random expansions and moderate regularization, without requiring task-level or iterative refinement.

## 3.2 Task Experts as Temporal Ensembles

To improve the competence of each expert under dynamic distributions, we draw inspiration from the fruit fly's KCs in the mushroom body and their connections to output neurons. This brain structure integrates multi-timescale plasticity and hierarchical processing across subregions, where $\gamma$ KCs mediate short-term memory, $\alpha'/\beta'$ KCs support intermediate memory, and $\alpha/\beta$ KCs are critical for long-term memory consolidation (Krashes et al., 2007; Cervantes-Sandoval et al., 2013; Bouzaiane et al., 2015). These KC subtypes are sequentially recruited during learning and exhibit compartment-specific modulation by dopamine neurons (Aso et al., 2014a; Owald & Waddell, 2015; Aso et al., 2014b; Aso & Rubin, 2016), enabling temporally staged memory formation and retrieval. Inspired by this biological design, we equip each expert in FlyPrompt with a *temporal ensemble of output heads*, implemented using exponential moving averages (EMA) with varying decay rates.

Concretely, instead of using only one shadow head in naïve EMA, each expert $E_t$ in FlyPrompt maintains a set of $n$ EMA heads with decay rates $\{\alpha_j\}_{j=1}^n$, where $\alpha_j \neq \alpha_k$ for all $j \neq k$. Let the online head be parameterized as $\psi = (\boldsymbol{W}, \boldsymbol{b}) \in \mathbb{R}^{|\mathcal{Y}| \times d} \times \mathbb{R}^{|\mathcal{Y}|}$, and the $j$-th EMA head of expert $E_t$ as $(\boldsymbol{W}_t^{(j)}, \boldsymbol{b}_t^{(j)})$. When a new task $t$ begins, its prompt $\boldsymbol{p}_t$ is initialized as the average of previously learned prompts as a warm start:

$$\boldsymbol{p}_t = \frac{1}{t-1}\sum_{i=1}^{t-1} \boldsymbol{p}_i \quad \text{for } t > 1,$$

with random initialization for $t = 1$. This average-prompt warm start provides a more informed initialization under single-pass GCL streams, where each expert only observes limited data, and empirically accelerates convergence and more compatible with blurry boundaries in which classes can reoccur across sessions. The EMA heads of $E_t$ are initialized as clones of the current online head. During training, only online head $\psi$ and prompt $\boldsymbol{p}_t$ are updated using the cross-entropy:

$$\mathcal{L}_t(\boldsymbol{x}, y) = \text{CE}\left(f_{\boldsymbol{\theta}}(\boldsymbol{x}; \boldsymbol{p}_t)\boldsymbol{W}^\top + \boldsymbol{b} + \boldsymbol{m}, y\right), \tag{6}$$

where $\boldsymbol{m} \in \mathbb{R}^{|\mathcal{Y}|}$ is a non-parametric logit mask initialized for each data batch $(\boldsymbol{X}, \boldsymbol{y})$. We set $m_c = 0$ and for any class $c \in \boldsymbol{y}$ encountered in the current batch , and set $m_{c'} = -\infty$ to suppress predictions on unseen labels $c' \notin \boldsymbol{y}$. This masking strategy mitigates interference from class imbalance both across and within tasks (Moon et al., 2023; Kang et al., 2025), evaluated in Tab. 15.

After each update step, the EMA heads are updated as:

$$\boldsymbol{W}_t^{(j)} \leftarrow \alpha_j\,\boldsymbol{W}_t^{(j)} + (1-\alpha_j)\,\boldsymbol{W}, \quad \boldsymbol{b}_t^{(j)} \leftarrow \alpha_j\,\boldsymbol{b}_t^{(j)} + (1-\alpha_j)\,\boldsymbol{b}. \tag{7}$$

At inference, the REAR module first selects an expert $e = \hat{E}(\boldsymbol{x})$. Using the associated prompt $\boldsymbol{p}_e$, we compute logits from the online and EMA heads:

$$\boldsymbol{z}^{(0)} = f_{\boldsymbol{\theta}}(\boldsymbol{x}; \boldsymbol{p}_e)\boldsymbol{W}^\top + \boldsymbol{b}, \tag{8}$$

$$\boldsymbol{z}^{(j)} = f_{\boldsymbol{\theta}}(\boldsymbol{x}; \boldsymbol{p}_e)\boldsymbol{W}_e^{(j)\top} + \boldsymbol{b}_e^{(j)}, \quad \forall j \in \{1, \cdots, n\}. \tag{9}$$

We then ensemble all $n + 1$ heads by computing the SoftMax of each and taking their element-wise maximum, followed by logit masking:

$$\hat{\boldsymbol{z}}(\boldsymbol{x}) = \max_{j \in \{0,\ldots,n\}} \text{softmax}(\boldsymbol{z}^{(j)} + \boldsymbol{m}), \qquad \hat{y}(\boldsymbol{x}) = \arg\max_c \hat{z}_c(\boldsymbol{x}). \tag{10}$$

This temporal ensemble mechanism enables FlyPrompt to integrate stable, long-term information via EMA heads while preserving rapid adaptation through the online head, mirroring biological memory consolidation and facilitating robust inference under non-stationary, imbalanced streams. Here, we also present a theoretical guarantee that supports the use of multiple EMA heads in GCL.

Table 1: Overall performance of representative methods over three GCL benchmarks across PTMs.

| PTM | Method | CIFAR-100 | | ImageNet-R | | CUB-200 | |
|---|---|---|---|---|---|---|---|
| | | $A_{\mathrm{auc}}(\%,\uparrow)$ | $A_{\mathrm{last}}(\%,\uparrow)$ | $A_{\mathrm{auc}}(\%,\uparrow)$ | $A_{\mathrm{last}}(\%,\uparrow)$ | $A_{\mathrm{auc}}(\%,\uparrow)$ | $A_{\mathrm{last}}(\%,\uparrow)$ |
| Sup-21K | Seq FT | $19.71_{\pm3.39}$ | $10.42_{\pm4.92}$ | $7.51_{\pm3.94}$ | $2.29_{\pm0.85}$ | $3.47_{\pm0.41}$ | $1.49_{\pm0.42}$ |
| | Linear Probe | $49.69_{\pm6.09}$ | $23.07_{\pm7.33}$ | $29.24_{\pm1.26}$ | $16.87_{\pm3.14}$ | $28.96_{\pm2.46}$ | $17.33_{\pm3.08}$ |
| | Seq FT w/ SL | $64.90_{\pm7.18}$ | $62.06_{\pm1.89}$ | $47.20_{\pm1.47}$ | $39.60_{\pm2.43}$ | $56.16_{\pm4.32}$ | $56.50_{\pm3.08}$ |
| | L2P | $76.23_{\pm2.73}$ | $79.11_{\pm1.43}$ | $44.40_{\pm1.03}$ | $42.03_{\pm1.72}$ | $64.30_{\pm2.18}$ | $61.42_{\pm2.13}$ |
| | DualPrompt | $76.04_{\pm3.32}$ | $76.62_{\pm0.74}$ | $46.13_{\pm1.94}$ | $40.80_{\pm1.04}$ | $65.03_{\pm2.24}$ | $62.43_{\pm1.78}$ |
| | CODA-P | $79.13_{\pm3.06}$ | $\underline{80.91}_{\pm0.70}$ | $\underline{51.87}_{\pm2.81}$ | $\underline{48.09}_{\pm2.75}$ | $\underline{66.01}_{\pm2.20}$ | $\underline{62.90}_{\pm2.46}$ |
| | MVP | $67.74_{\pm4.96}$ | $63.22_{\pm0.69}$ | $39.50_{\pm1.41}$ | $32.63_{\pm3.95}$ | $54.69_{\pm3.14}$ | $50.07_{\pm3.86}$ |
| | MISA | $\underline{80.35}_{\pm2.39}$ | $80.75_{\pm1.24}$ | $51.52_{\pm2.09}$ | $45.08_{\pm1.43}$ | $65.40_{\pm3.01}$ | $60.20_{\pm1.82}$ |
| | FlyPrompt (Ours) | $\mathbf{83.24}_{\pm2.23}$ | $\mathbf{86.76}_{\pm0.73}$ | $\mathbf{56.58}_{\pm1.47}$ | $\mathbf{55.27}_{\pm0.91}$ | $\mathbf{70.64}_{\pm2.85}$ | $\mathbf{73.40}_{\pm1.88}$ |
| Sup-21K/1K | L2P | $63.88_{\pm7.79}$ | $68.96_{\pm7.63}$ | $47.10_{\pm1.21}$ | $42.22_{\pm1.94}$ | $42.96_{\pm4.13}$ | $45.00_{\pm3.83}$ |
| | DualPrompt | $68.02_{\pm2.08}$ | $67.04_{\pm5.84}$ | $\underline{52.80}_{\pm1.21}$ | $47.39_{\pm1.60}$ | $\underline{46.80}_{\pm2.89}$ | $\underline{46.39}_{\pm2.76}$ |
| | CODA-P | $\underline{69.29}_{\pm2.52}$ | $\underline{69.47}_{\pm7.19}$ | $51.20_{\pm1.76}$ | $44.30_{\pm1.50}$ | $44.66_{\pm2.73}$ | $45.18_{\pm4.50}$ |
| | MVP | $64.69_{\pm3.77}$ | $51.29_{\pm7.56}$ | $48.99_{\pm2.01}$ | $38.12_{\pm5.20}$ | $44.10_{\pm2.81}$ | $33.97_{\pm9.62}$ |
| | MISA | $62.91_{\pm7.96}$ | $67.99_{\pm7.41}$ | $50.87_{\pm1.69}$ | $\underline{47.75}_{\pm2.87}$ | $42.76_{\pm2.33}$ | $44.05_{\pm1.94}$ |
| | FlyPrompt (Ours) | $\mathbf{78.48}_{\pm1.31}$ | $\mathbf{80.39}_{\pm3.54}$ | $\mathbf{62.01}_{\pm2.32}$ | $\mathbf{56.55}_{\pm3.94}$ | $\mathbf{54.42}_{\pm4.67}$ | $\mathbf{55.50}_{\pm3.55}$ |
| iBOT-21K | L2P | $56.82_{\pm8.42}$ | $\underline{67.61}_{\pm8.76}$ | $35.97_{\pm1.62}$ | $36.95_{\pm2.44}$ | $14.76_{\pm1.53}$ | $\underline{24.51}_{\pm4.82}$ |
| | DualPrompt | $\underline{66.06}_{\pm4.52}$ | $67.14_{\pm8.60}$ | $42.48_{\pm1.62}$ | $35.91_{\pm0.88}$ | $19.90_{\pm3.68}$ | $21.84_{\pm2.35}$ |
| | CODA-P | $62.13_{\pm7.17}$ | $63.38_{\pm7.98}$ | $\underline{45.50}_{\pm1.66}$ | $\underline{39.44}_{\pm1.35}$ | $17.72_{\pm5.33}$ | $20.82_{\pm7.66}$ |
| | MVP | $62.33_{\pm3.06}$ | $48.32_{\pm11.42}$ | $41.55_{\pm1.98}$ | $29.29_{\pm5.03}$ | $\underline{28.73}_{\pm3.18}$ | $23.62_{\pm9.51}$ |
| | MISA | $65.30_{\pm2.28}$ | $67.43_{\pm6.75}$ | $40.94_{\pm1.22}$ | $36.16_{\pm1.58}$ | $18.62_{\pm3.36}$ | $23.66_{\pm2.21}$ |
| | FlyPrompt (Ours) | $\mathbf{75.58}_{\pm1.70}$ | $\mathbf{79.36}_{\pm3.47}$ | $\mathbf{57.75}_{\pm2.12}$ | $\mathbf{54.39}_{\pm1.29}$ | $\mathbf{28.86}_{\pm5.84}$ | $\mathbf{36.79}_{\pm7.58}$ |
| iBOT-1K | L2P | $53.17_{\pm7.08}$ | $\underline{62.28}_{\pm8.19}$ | $38.29_{\pm2.65}$ | $39.86_{\pm0.95}$ | $19.20_{\pm2.21}$ | $31.21_{\pm5.24}$ |
| | DualPrompt | $52.39_{\pm3.21}$ | $53.56_{\pm6.10}$ | $45.76_{\pm1.63}$ | $39.19_{\pm0.65}$ | $29.32_{\pm3.15}$ | $30.53_{\pm5.33}$ |
| | CODA-P | $\underline{59.29}_{\pm4.03}$ | $61.30_{\pm6.73}$ | $\underline{49.56}_{\pm1.57}$ | $\underline{42.64}_{\pm2.78}$ | $27.57_{\pm2.83}$ | $33.61_{\pm4.52}$ |
| | MVP | $57.52_{\pm3.62}$ | $44.08_{\pm12.42}$ | $44.76_{\pm2.23}$ | $34.93_{\pm4.48}$ | $\underline{33.81}_{\pm3.50}$ | $26.32_{\pm9.97}$ |
| | MISA | $54.31_{\pm2.91}$ | $55.89_{\pm5.10}$ | $43.91_{\pm3.95}$ | $40.09_{\pm1.24}$ | $27.76_{\pm2.69}$ | $\underline{33.74}_{\pm2.11}$ |
| | FlyPrompt (Ours) | $\mathbf{70.14}_{\pm1.76}$ | $\mathbf{74.84}_{\pm4.26}$ | $\mathbf{61.50}_{\pm1.66}$ | $\mathbf{57.18}_{\pm1.36}$ | $\mathbf{38.75}_{\pm5.72}$ | $\mathbf{45.00}_{\pm4.19}$ |
| DINO-1K | L2P | $47.98_{\pm7.38}$ | $\underline{59.13}_{\pm6.32}$ | $35.81_{\pm1.37}$ | $36.58_{\pm1.31}$ | $21.18_{\pm2.01}$ | $32.47_{\pm6.10}$ |
| | DualPrompt | $52.12_{\pm4.01}$ | $55.71_{\pm6.11}$ | $43.03_{\pm1.12}$ | $35.40_{\pm1.40}$ | $27.80_{\pm4.21}$ | $29.49_{\pm4.24}$ |
| | CODA-P | $\underline{54.69}_{\pm4.49}$ | $58.91_{\pm5.43}$ | $\underline{45.16}_{\pm2.05}$ | $\underline{38.23}_{\pm2.02}$ | $29.22_{\pm2.97}$ | $31.85_{\pm7.47}$ |
| | MVP | $53.64_{\pm3.91}$ | $41.02_{\pm12.09}$ | $41.78_{\pm2.15}$ | $32.00_{\pm4.22}$ | $\underline{33.44}_{\pm3.43}$ | $26.02_{\pm10.29}$ |
| | MISA | $52.03_{\pm3.07}$ | $55.98_{\pm4.26}$ | $41.26_{\pm3.25}$ | $37.50_{\pm1.62}$ | $27.13_{\pm3.31}$ | $\underline{33.08}_{\pm4.10}$ |
| | FlyPrompt (Ours) | $\mathbf{65.92}_{\pm2.74}$ | $\mathbf{72.66}_{\pm4.52}$ | $\mathbf{57.29}_{\pm2.40}$ | $\mathbf{54.72}_{\pm1.89}$ | $\mathbf{37.38}_{\pm5.86}$ | $\mathbf{44.66}_{\pm2.35}$ |
| MoCo v3-1K | L2P | $28.17_{\pm7.08}$ | $39.07_{\pm11.31}$ | $17.43_{\pm1.71}$ | $16.27_{\pm5.43}$ | $12.42_{\pm2.31}$ | $20.00_{\pm7.36}$ |
| | DualPrompt | $53.33_{\pm4.65}$ | $58.20_{\pm7.73}$ | $36.69_{\pm1.74}$ | $30.24_{\pm1.94}$ | $19.88_{\pm3.35}$ | $21.93_{\pm4.30}$ |
| | CODA-P | $53.47_{\pm3.42}$ | $58.55_{\pm7.19}$ | $\underline{39.89}_{\pm2.71}$ | $31.72_{\pm4.86}$ | $20.09_{\pm2.52}$ | $24.10_{\pm6.48}$ |
| | MVP | $54.33_{\pm4.56}$ | $40.84_{\pm14.21}$ | $36.45_{\pm2.35}$ | $26.37_{\pm6.04}$ | $\mathbf{28.48}_{\pm3.34}$ | $23.56_{\pm9.78}$ |
| | MISA | $\underline{57.00}_{\pm6.06}$ | $\underline{62.18}_{\pm3.94}$ | $38.85_{\pm4.27}$ | $\underline{33.47}_{\pm0.95}$ | $25.02_{\pm4.39}$ | $\underline{27.68}_{\pm4.35}$ |
| | FlyPrompt (Ours) | $\mathbf{64.12}_{\pm5.18}$ | $\mathbf{71.51}_{\pm8.48}$ | $\mathbf{52.32}_{\pm1.50}$ | $\mathbf{49.06}_{\pm1.35}$ | $\underline{27.92}_{\pm4.53}$ | $\mathbf{33.32}_{\pm3.58}$ |

**Theorem 2** (TE$^2$, informal). *For an EMA head with decay $\alpha$ and window $L = 1/(1 - \alpha)$, the parameter error at time $t$ satisfies*

$$\mathbb{E}\left\|\widetilde{\boldsymbol{W}}_t^{(\alpha)} - \boldsymbol{W}_t^\star\right\|^2 \lesssim \zeta^2/L + (L\,P_t)^2,$$

*where $\zeta^2$ bounds the online noise and $P_t$ measures drift. A geometric EMA bank contains, at every time, a head that achieves a near-optimal bias-variance trade-off up to a constant factor.*

**Interpretation.** The bound decomposes the parameter error into a variance term $O(\zeta^2/L)$, controlled by the effective window size, and a drift-induced bias term $O((LP_t)^2)$, which increases with nonstationarity. Larger $L$ reduces variance but increases bias, creating a bias-variance trade-off. A geometric bank of EMA windows ensures that, at any time, one head is near the optimal trade-off for the current drift level. Intuitively, when the input stream contains segments with varying temporal dynamics, such as sudden shifts at session transitions or gradual changes within each task, different EMA heads can align better with different segments, leading to more adaptive predictions. In practice, two EMA heads with windows of 10 and 100 ($\alpha = 0.9, 0.99$) are sufficient (see Sec. 4.2).

## 4 EXPERIMENT

In this section, we first introduce the experiment setups and then present the experiment results.

Table 2: Comparison with prominent offline methods on three GCL benchmarks under Sup-21K.

| Method | CIFAR-100 | | ImageNet-R | | CUB-200 | |
|---|---|---|---|---|---|---|
| | $A_{\mathrm{auc}}(\%, \uparrow)$ | $A_{\mathrm{last}}(\%, \uparrow)$ | $A_{\mathrm{auc}}(\%, \uparrow)$ | $A_{\mathrm{last}}(\%, \uparrow)$ | $A_{\mathrm{auc}}(\%, \uparrow)$ | $A_{\mathrm{last}}(\%, \uparrow)$ |
| S-Prompt++ | $\underline{80.21}_{\pm 2.55}$ | $\underline{83.48}_{\pm 1.20}$ | $52.14_{\pm 1.65}$ | $49.13_{\pm 1.60}$ | $66.61_{\pm 2.21}$ | $64.73_{\pm 2.25}$ |
| HiDe-Prompt | $77.10_{\pm 3.81}$ | $81.77_{\pm 2.00}$ | $53.77_{\pm 1.09}$ | $49.87_{\pm 3.01}$ | $67.05_{\pm 2.37}$ | $67.12_{\pm 0.50}$ |
| HiDe-LoRA | $80.07_{\pm 2.41}$ | $82.00_{\pm 1.25}$ | $55.09_{\pm 1.45}$ | $51.29_{\pm 6.29}$ | $\underline{67.26}_{\pm 1.76}$ | $\underline{67.28}_{\pm 1.45}$ |
| HiDe-Adapter | $79.52_{\pm 2.81}$ | $81.41_{\pm 0.95}$ | $53.92_{\pm 1.32}$ | $50.86_{\pm 5.08}$ | $66.09_{\pm 1.41}$ | $64.53_{\pm 1.78}$ |
| NoRGa | $78.89_{\pm 3.33}$ | $83.03_{\pm 1.20}$ | $54.12_{\pm 1.37}$ | $50.09_{\pm 3.66}$ | $67.16_{\pm 2.44}$ | $67.06_{\pm 0.58}$ |
| SD-LoRA | $79.26_{\pm 2.21}$ | $78.91_{\pm 2.48}$ | $\underline{55.51}_{\pm 1.30}$ | $\underline{51.97}_{\pm 3.09}$ | $64.12_{\pm 2.02}$ | $60.57_{\pm 0.77}$ |
| FlyPrompt (Ours) | $\mathbf{83.24}_{\pm 2.23}$ | $\mathbf{86.76}_{\pm 0.73}$ | $\mathbf{56.58}_{\pm 1.47}$ | $\mathbf{55.27}_{\pm 0.91}$ | $\mathbf{70.64}_{\pm 2.85}$ | $\mathbf{73.40}_{\pm 1.88}$ |

## 4.1 EXPERIMENT SETUP

**Benchmarks.** We evaluate FlyPrompt under the Si-Blurry GCL setting (Moon et al., 2023; Kang et al., 2025) using three representative benchmarks: CIFAR-100 (Krizhevsky et al., 2009) (60K images, 100 classes), ImageNet-R (Hendrycks et al., 2021) (30K images, 200 classes), and CUB-200 (Wah et al., 2011) (12K images, 200 fine-grained classes). Unless specified otherwise, we adopt the default Si-Blurry configuration with disjoint class ratio $r_{\mathrm{D}} = 50\%$ and blurry sample ratio $r_{\mathrm{B}} = 10\%$, trained over five sessions. We report two widely used metrics: average anytime accuracy $A_{\mathrm{auc}}$ (evaluated every 1000 batches) and final accuracy $A_{\mathrm{last}}$ (measured after all sessions) (Koh et al., 2021). Additional experiments of different $(r_{\mathrm{D}}, r_{\mathrm{B}})$ and online CL are provided in Appendix F.1. Unless specified, all results are averaged over five runs ($\pm$ standard deviation) with different seeds.

**Baselines.** We compare FlyPrompt against a diverse set of CL and GCL methods: (1) lower-bound baselines such as sequential fine-tuning (Seq FT, including the version with a slow learning rate, SL) (Zhang et al., 2023) and linear probing; (2) prompt-based CL baselines including L2P (Wang et al., 2022d), DualPrompt (Wang et al., 2022c) and CODA-P (Smith et al., 2023); (3) state-of-the-art GCL methods such as MVP (Moon et al., 2023) and MISA (Kang et al., 2025); (4) prominent offline CL methods S-Prompt++ (Wang et al., 2022b), HiDe (Wang et al., 2023a), NoRGa (Le et al., 2024) and SD-LoRA (Wu et al., 2025) in Tabs. 2 and 12. We also implement the online version of the analytic baseline RanPAC (McDonnell et al., 2024) in Tab. 3 and other variants in Tab. 13.

**Implementation.** We adopt the ViT-B/16 backbone pretrained on ImageNet-21K and ImageNet-1K, including strong supervised paradigms Sup-21K, Sup-21K/1K (Sup-21K fine-tuned on ImageNet-1K) (Ridnik et al., 2021; Dosovitskiy et al., 2020), and self-supervised paradigms iBOT-21K, iBOT-1K (Zhou et al., 2021), DINO-1K (Caron et al., 2021), and MoCo v3-1K (Chen et al., 2021). We set the projection dimension $M = 10^4$, and $\lambda$ based on checkpoints: $10^4$ (Sup-21K), $10^6$ (Sup-21K/1K, MoCo), and $10^7$ (iBOT, DINO). We use $n = 2$ EMA heads with decay rates $0.9, 0.99$. More implementation details of baseline methods and GCL benchmark setup can be found in Appendix C.

## 4.2 EXPERIMENT RESULTS

**Overall Performance.** Tab. 1 summarizes GCL performance across all benchmarks. Prompt-based CL methods perform well with supervised backbones (e.g., Sup-21K, Sup-21K/1K), but degrade significantly under self-supervised ones (e.g., DINO, MoCo v3-1K), particularly on fine-grained benchmarks CUB-200. This highlights the challenge of extracting discriminative features without strong pretraining priors. MVP, which incorporates contrastive learning for improved expert selection, outperforms others under the fine-grained benchmark and self-supervised PTMs, reinforcing the importance of prompt routing. However, the contrastive loss yields limited performance gains in other cases due to the absence of a replay buffer. Fig. 1 presents the anytime accuracy during GCL. MISA benefits from stronger prompt initialization and achieves relatively higher performance at the early stage, but steadily declines due to parameter overwriting, eventually matching weaker baselines like CODA-P. This suggests that while good initialization helps, it alone is insufficient for sustained GCL performance. In contrast, **FlyPrompt** consistently outperforms all baselines across datasets and PTMs. It achieves up to 11.23%, 12.43%, and 7.62% improvements in $A_{\mathrm{auc}}$; 13.53%, 16.49%, and 12.28% in $A_{\mathrm{last}}$ on CIFAR-100, ImageNet-R, and CUB-200, respectively. As shown in Fig. 1, FlyPrompt maintains stable, high accuracy throughout GCL, with minimal drops during session transitions. This results confirm FlyPrompt as a new state-of-the-art for GCL.

Table 3: Ablation study of different components in FlyPrompt. "−" indicates not applicable.

| PTM | FlyPrompt Components | | | CIFAR-100 | | ImageNet-R | |
|---|---|---|---|---|---|---|---|
| | REAR | Prompt Expert | EMA head | $A_{auc}(\%,\uparrow)$ | $A_{last}(\%,\uparrow)$ | $A_{auc}(\%,\uparrow)$ | $A_{last}(\%,\uparrow)$ |
| Sup-21K | RP-Based Analytic Classifier (RanPAC†) | | | $69.91_{\pm3.88}$ | $79.92_{\pm0.07}$ | $47.29_{\pm0.70}$ | $47.33_{\pm0.12}$ |
| | − | × | × | $71.33_{\pm2.17}$ | $73.22_{\pm1.63}$ | $41.73_{\pm0.97}$ | $37.33_{\pm1.71}$ |
| | − | × | ✓ | $71.69_{\pm2.27}$ | $73.30_{\pm1.55}$ | $42.50_{\pm1.01}$ | $38.35_{\pm1.01}$ |
| | × | ✓ | × | $80.75_{\pm1.98}$ | $83.65_{\pm1.94}$ | $54.91_{\pm1.32}$ | $52.58_{\pm1.36}$ |
| | × | ✓ | ✓ | $82.17_{\pm2.07}$ | $83.75_{\pm1.86}$ | $55.90_{\pm1.37}$ | $53.65_{\pm0.92}$ |
| | ✓ | ✓ | × | $81.90_{\pm2.20}$ | $84.23_{\pm1.32}$ | $55.76_{\pm1.32}$ | $52.76_{\pm1.30}$ |
| | ✓ | ✓ | ✓ | $\mathbf{83.24}_{\pm2.23}$ | $\mathbf{86.76}_{\pm0.73}$ | $\mathbf{56.58}_{\pm1.47}$ | $\mathbf{55.27}_{\pm0.91}$ |
| Sup-21K/1K | RP-Based Analytic Classifier (RanPAC†) | | | $69.76_{\pm3.33}$ | $79.49_{\pm0.16}$ | $52.91_{\pm1.07}$ | $54.79_{\pm0.22}$ |
| | − | × | × | $57.82_{\pm6.94}$ | $62.67_{\pm8.55}$ | $46.24_{\pm0.72}$ | $39.06_{\pm2.81}$ |
| | − | × | ✓ | $60.76_{\pm6.77}$ | $63.39_{\pm8.81}$ | $49.88_{\pm0.93}$ | $41.65_{\pm1.38}$ |
| | × | ✓ | × | $70.09_{\pm3.52}$ | $67.62_{\pm5.80}$ | $52.07_{\pm1.35}$ | $44.13_{\pm3.61}$ |
| | × | ✓ | ✓ | $72.51_{\pm3.15}$ | $68.66_{\pm5.96}$ | $55.55_{\pm1.51}$ | $45.90_{\pm3.53}$ |
| | ✓ | ✓ | × | $71.28_{\pm2.58}$ | $69.73_{\pm5.78}$ | $53.12_{\pm2.19}$ | $44.69_{\pm3.65}$ |
| | ✓ | ✓ | ✓ | $\mathbf{78.48}_{\pm1.31}$ | $\mathbf{80.39}_{\pm3.54}$ | $\mathbf{62.01}_{\pm2.32}$ | $\mathbf{56.55}_{\pm3.94}$ |

Table 4: Effect of REAR and $TE^2$ on $A_{auc}(\%)$ performance for PTM-based CL methods using CIFAR-100 under Sup-21K. The numbers in parentheses indicate the difference from the baseline, and the arrow direction indicates an increase (↑) or decrease (↓). See Tab. 14 for complete results.

| Method | Baseline | w/ REAR | w/ $TE^2$ | w/ Both |
|---|---|---|---|---|
| DualPrompt | $76.04_{\pm3.32}$ | $80.63_{\pm2.25}$ (↑ 4.59) | $76.83_{\pm3.44}$ (↑ 0.79) | $82.33_{\pm2.17}$ (↑ 6.29) |
| MVP | $67.74_{\pm4.96}$ | $67.44_{\pm4.89}$ (↓ 0.30) | $68.91_{\pm4.86}$ (↑ 1.17) | $68.93_{\pm4.60}$ (↑ 1.19) |
| MISA | $\mathbf{80.35}_{\pm2.39}$ | $82.03_{\pm1.97}$ (↑ 1.68) | $\underline{81.65}_{\pm2.24}$ (↑ 1.30) | $\mathbf{83.60}_{\pm2.08}$ (↑ 3.25) |
| S-Prompt++ | $\underline{80.21}_{\pm2.55}$ | $81.43_{\pm2.45}$ (↑ 1.21) | $\mathbf{81.93}_{\pm2.21}$ (↑ 1.72) | $\underline{83.11}_{\pm2.30}$ (↑ 2.90) |
| HiDe-Prompt | $77.10_{\pm3.81}$ | $78.41_{\pm2.64}$ (↑ 1.31) | $77.46_{\pm3.56}$ (↑ 0.36) | $78.60_{\pm2.53}$ (↑ 1.51) |
| NoRGa | $78.89_{\pm3.33}$ | $79.37_{\pm2.71}$ (↑ 0.48) | $79.16_{\pm3.28}$ (↑ 0.27) | $79.37_{\pm2.74}$ (↑ 0.48) |

**Ablation Study.** To assess the contribution of each FlyPrompt component, we conduct a comprehensive ablation study of REAR for prompt selection, multi-prompt across tasks, and $TE^2$ for EMA head ensemble. Results in Tab. 3 show that each module provides consistent gains, with the full FlyPrompt achieving the best performance. We additionally include RanPAC†, an analytic learner using random projections but no expert modularity, to simulate REAR without multi-expert routing. While this performs competitively under limited training, it falls short without expert specialization, underscoring the importance of both routing and competence. Notably, the gain from EMA heads alone is modest unless combined with REAR and prompt modularization, highlighting the synergy among bio-inspired components rather than simple additive effects. We further integrate our REAR and $TE^2$ components into a range of strong baseline models by replacing their routing and output head modules correspondingly. Results in Tab. 4 further demonstrate the consistent improvements when either component is added (more results across datasets and metrics are presented in Tab. 14).

**Hyperparameter Sensitivity.** Fig. 5 explores key hyperparameters in REAR. Increasing the projection dimension $M$ improves performance, consistent with the theory that higher-dimensional spaces enable better feature separability and router performance (in Theorem 1), mirroring sparse expansion in the fruit fly mushroom body. However, since the memory cost grows linearly with $M$, we set $M = 10,000$ as a practical trade-off. The regularization parameter $\lambda$ has smaller impact, with performance stable across several orders of magnitude. Full results across other PTMs are provided in Appendix F.5. We further analyze EMA decay rates with temporal ensemble. Tab. 6 shows that two EMA heads of $0.9, 0.99$, combined with the online head, achieve the best trade-off across datasets. This aligns with neurobiological findings that the mushroom body maintains short-, mid-, and long-term memory modules in parallel. Among various ensemble strategies (Tab. 5), the "SoftMax + element-wise maximum" method is most effective and used by default. Detailed evaluations across other PTMs and configurations are provided in Appendices F.8 and F.10.

**Detailed Analysis.** Returning to the core challenges identified in Sec. 2.2, we revisit the roles of expert routing and expert competence improvement. As shown in Fig. 2, methods (e.g., FlyPrompt) that improve in these two areas correlate strongly with better overall GCL performance. In particular, Fig. 2b demonstrates the impact of random projection in boosting routing accuracy, while Fig. 2c highlights remaining headroom for improving expert competence. Despite introducing RP layer and tracking feature statistics, FlyPrompt adds minimal parameter overhead, i.e., just 0.83% more

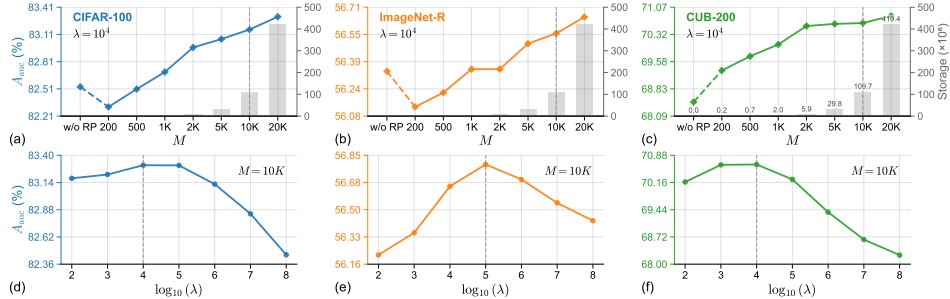

Figure 5: Analysis of hyperparameters in REAR. (a-c) Different projection dimension $M$ with fixed $\lambda = 10^4$: we report $A_{\text{auc}}$ and extra storage cost (bar) given $M$. (d-f) Different regularization parameter $\lambda$ with fixed $M = 10^4$. Dashed lines indicate the optimal choice of each hyperparameter.

Table 5: Performance comparison of ensemble aggregation choices for TE$^2$ under Sup-21K.

| Ensemble Method | CIFAR-100 | | ImageNet-R | | CUB-200 | |
|---|---|---|---|---|---|---|
| | $A_{\text{auc}}(\%,\uparrow)$ | $A_{\text{last}}(\%,\uparrow)$ | $A_{\text{auc}}(\%,\uparrow)$ | $A_{\text{last}}(\%,\uparrow)$ | $A_{\text{auc}}(\%,\uparrow)$ | $A_{\text{last}}(\%,\uparrow)$ |
| Mean | $81.34_{\pm1.64}$ | $85.11_{\pm1.03}$ | $52.71_{\pm1.36}$ | $53.24_{\pm1.22}$ | $68.49_{\pm2.57}$ | $\mathbf{73.95}_{\pm1.90}$ |
| Max Prob. | $82.29_{\pm2.25}$ | $84.95_{\pm1.20}$ | $55.56_{\pm1.38}$ | $53.53_{\pm1.40}$ | $68.00_{\pm2.50}$ | $66.56_{\pm1.60}$ |
| Min Entropy | $81.92_{\pm2.19}$ | $84.23_{\pm1.32}$ | $55.05_{\pm1.31}$ | $52.88_{\pm1.38}$ | $66.78_{\pm2.53}$ | $64.73_{\pm1.36}$ |
| SoftMax+Mean | $82.30_{\pm1.82}$ | $85.98_{\pm0.80}$ | $\underline{56.16}_{\pm1.56}$ | $\mathbf{55.53}_{\pm0.89}$ | $\mathbf{70.77}_{\pm3.00}$ | $\underline{74.86}_{\pm1.54}$ |
| SoftMax+Max Prob. | $\mathbf{83.24}_{\pm2.23}$ | $\mathbf{86.76}_{\pm0.73}$ | $\mathbf{56.58}_{\pm1.47}$ | $\underline{55.27}_{\pm0.91}$ | $\underline{70.64}_{\pm2.85}$ | $73.40_{\pm1.88}$ |
| SoftMax+Min Entropy | $\underline{83.11}_{\pm2.34}$ | $\underline{86.50}_{\pm0.64}$ | $55.94_{\pm1.41}$ | $54.24_{\pm1.34}$ | $69.86_{\pm2.80}$ | $71.51_{\pm1.79}$ |

Table 6: Performance comparison of different EMA decay rates for TE$^2$ under Sup-21K.

Table 7: Computational cost and overall performance using CIFAR-100 under Sup-21K.

| EMA Decay Rate | CIFAR-100 | | ImageNet-R | |
|---|---|---|---|---|
| | $A_{\text{auc}}(\%,\uparrow)$ | $A_{\text{last}}(\%,\uparrow)$ | $A_{\text{auc}}(\%,\uparrow)$ | $A_{\text{last}}(\%,\uparrow)$ |
| Online head only | $81.90_{\pm2.20}$ | $84.23_{\pm1.32}$ | $54.91_{\pm1.32}$ | $52.58_{\pm1.36}$ |
| +0.9 | $82.81_{\pm2.28}$ | $86.36_{\pm0.54}$ | $\underline{56.36}_{\pm1.52}$ | $55.09_{\pm0.89}$ |
| +0.99 | $82.84_{\pm2.51}$ | $\underline{86.41}_{\pm0.39}$ | $55.94_{\pm1.65}$ | $54.67_{\pm0.89}$ |
| +0.999 | $81.80_{\pm2.37}$ | $84.39_{\pm0.83}$ | $55.15_{\pm1.39}$ | $53.52_{\pm0.80}$ |
| +0.9,0.99 | $\mathbf{83.24}_{\pm2.23}$ | $\mathbf{86.76}_{\pm0.73}$ | $\mathbf{56.58}_{\pm1.47}$ | $\underline{55.27}_{\pm0.91}$ |
| +0.9,0.99,0.999 | $\underline{82.99}_{\pm2.22}$ | $86.24_{\pm0.79}$ | $56.35_{\pm1.72}$ | $\mathbf{55.50}_{\pm0.77}$ |

| Method | Total Param. (M,↓) | Trainable Param. (M,↓) | Time Delay (s/batch,↓) | $A_{\text{auc}}$ (%,↑) |
|---|---|---|---|---|
| L2P | 86.01 | 0.22 | 5.17 | 76.23 |
| DualPrompt | 86.35 | 0.55 | 4.78 | 76.04 |
| CODA-P | 86.72 | 0.92 | 4.75 | 79.13 |
| MVP | 86.12 | 0.32 | 5.35 | 67.74 |
| MISA | 86.37 | 0.58 | 4.78 | 80.35 |
| FlyPrompt (ours) | 87.08 | 0.46 | 4.96 | $\mathbf{83.24}$ |

parameters than MISA on ViT-B/16, and incurs negligible increase in computational cost (see Tab. 7, more comprehensive comparison in Tab. 12 and detailed cost breakdown of components in Tab. 19). Together, these findings validate FlyPrompt's effectiveness in resolving the GCL challenges.

## 5 CONCLUSION

We presented **FlyPrompt**, a biologically inspired framework for GCL, which addresses the core challenges of expert routing and expert competence improvement under blurred task boundaries and single-pass constraints. Grounded in the neurobiological principles of the fruit fly's mushroom body, known for its sparse expansion, random connectivity, and multiscale modularity, FlyPrompt integrates a randomly expanded analytic router for non-iterative expert selection and a temporal ensemble of expert heads for robust adaptation over time. Theoretical analysis and empirical results across multiple GCL benchmarks demonstrate its strong performance and scalability.

While these results are encouraging, several limitations of the current work point to promising future directions. For instance, the temporal ensemble relies on a fixed composition of EMA decay rates, and adapting these dynamically to data drift could enhance robustness. Additionally, performance under extreme long-tailed distributions warrants further study. Looking forward, GCL is essential for deploying real-world learning systems, such as embodied agents, user-facing AI, and resource-constrained devices, where data is dynamic and supervision is limited. As continual adaptation is a natural strength of biological systems, the underlying principles they offer will continue to inspire future advances in GCL and beyond.

**Acknowledgment.** This work was supported by the STI2030-Major Projects 2022ZD0204900, the National Natural Science Foundation of China (NO. 62406160, 32530042, 32021002), the Beijing Natural Science Foundation L247011, and the Beijing Major Science and Technology Project No. Z251100008425003.

**Ethics statement.** I acknowledge that I and all co-authors of this work have read and commit to adhering to the ICLR Code of Ethics.

**Reproducibility statement.** We have included the source code with clear instructions, and will release them upon acceptance.

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

## A  Large language models assistance

Large language models were used to polish the manuscript. The authors have thoroughly reviewed and edited all content and take full responsibility for the published work.

## B  Related Work

**Continual Learning (CL)** aims to train models on task streams with evolving distributions (Wang et al., 2024b; 2021; 2022a; Parisi et al., 2019). Canonical CL settings are categorized into task-incremental learning (TIL), class-incremental learning (CIL), and domain-incremental learning (DIL) (Van de Ven & Tolias, 2019; Yan et al., 2024; 2025), depending on the structure of input and label spaces. TIL and CIL assume disjoint label spaces, with task identity provided only in TIL, while DIL shares label space but varies input domains. Theoretically, CL has been formalized by decoupling task identity prediction (TIP) and within-task prediction (WTP), which remain orthogonal under clearly segmented tasks and from-scratch training (Kim et al., 2022; Wang et al., 2023a).

The rise of pretrained models (PTMs) has shifted CL towards adapting frozen backbones via lightweight modules, known as parameter-efficient tuning (PET) (Lester et al., 2021; Li & Liang, 2021; Rebuffi et al., 2017; Hu et al., 2021). PET-based CL methods often employ either task-shared modules (Zhang et al., 2023; McDonnell et al., 2024; Zhou et al., 2024a; 2025b;a; 2024b) that require gradual updates, or task-specific experts (Wang et al., 2022d; 2023a) that demand effective expert selection (implicitly via external queries (Wang et al., 2022d) or explicitly via routing functions (Wang et al., 2023a)). Importantly, the strong priors embedded in PTMs blur the TIP–WTP decomposition, making classical CL theory less applicable (Wang et al., 2023a; 2024a).

**General Continual Learning (GCL)** extends CL to more practical scenarios by removing assumptions of clear task segmentation and offline data access (Buzzega et al., 2020; De Lange et al., 2021; Mi et al., 2020). Specifically, GCL emphasizes *online learning*, where each data point is seen only once; and *blurry or unknown task boundaries*, where task identities are absent or ill-defined (Aljundi et al., 2019a; Prabhu et al., 2020; Bang et al., 2021; Moon et al., 2023). These properties introduce unique challenges in expert selection, knowledge retention, and fast adaptation, without task identities or replay buffers. Additional constraints, such as constant memory budgets and anytime inference, further distinguish GCL from traditional CL (De Lange et al., 2021).

To implement the GCL challenges, benchmarks such as Task-Free CL (Aljundi et al., 2019a; Prabhu et al., 2020) and Si-Blurry (Moon et al., 2023) have been proposed, progressively relaxing task-awareness and enforcing stream-based learning. Correspondingly, GCL methods adapt replay-based sampling (Aljundi et al., 2019b; Bang et al., 2021), memory management (Koh et al., 2021), or PET-based designs (Moon et al., 2023; Kang et al., 2025). However, replay methods raise privacy and scalability concerns, while recent PET-based methods (e.g., MVP (Moon et al., 2023) and MISA (Kang et al., 2025)) still suffer from limited representation capacity and lack principled mechanisms for prompt expert selection under non-stationary inputs. Consequently, their improvements over naive PTM-based baselines remain modest.

## C  Implementation Details

### C.1  Training Setup

We follow the previous GCL studies (Moon et al., 2023; Kang et al., 2025) for a fair comparison. The standard ViT-B/16 transformer backbone has an embedding dimension of $d = 768$. For prompt-based methods, we unify the prompt length to 5 and the position to insert the prompt as the first five layers of ViT. All methods are trained with an Adam optimizer with a learning rate $0.005$ and zero weight decay. We set the batch size to 64, the epoch number to 1 (online learning), and the online iteration of each batch to 3. All images are cropped and resized to $224 \times 224$ to fit the ViT format using standard data transformation operations (resize, random crop, random horizontal flip and normalization). Moreover, the logit mask $m$ trick in Sec. 3.2 is generally applied to all methods to enhance training stability. All experiment jobs are performed on the same Linux server with Intel Xeon Silver 4316 2.3GHz CPUs (20 cores), 1 NVIDIA RTX 4090 GPU. For random seeds, we use the fixed values 1, 2, 3, 4, and 5 for all parallel runs.

## C.2 BASELINES

Unless otherwise specified, all baselines share the common ViT-B/16 backbone, single-pass Si-Blurry streams, optimizer, and data preprocessing described at the beginning of this section; below, we highlight method-specific architectures and the GCL-specific adaptations.

**Sequential fine-tuning (Seq FT / SL) and Linear probing.** Seq FT fine-tunes all parameters of the ViT-B/16 backbone and classifier on the Si-Blurry streams without replay buffers or task-specific heads; SL is an otherwise identical variant with a smaller learning rate (i.e., $5 \times 10^{-5}$, 10 times smaller than 0.005 used by other baselines) to provide a more optimistic lower bound (Zhang et al., 2023). Linear probing instead freezes the backbone and trains only a linear classifier, with no prompt modules or expert structures; together, these methods serve as simple PTM-based lower bounds.

**Prompt-based CL baselines (L2P, DualPrompt, CODA-Prompt).** For L2P (Wang et al., 2022d), DualPrompt (Wang et al., 2022c), and CODA-Prompt (Smith et al., 2023), we keep their original prompt-controller designs (key-based prompt pool in L2P, global+task prompts in DualPrompt, and attention-based prompts in CODA-Prompt), but adapt them to GCL by freezing the ViT-B/16 backbone and unifying prompt length and insertion position as in Sec. C.1. All three methods are trained in a single online pass over the Si-Blurry streams with the same update schedule as FlyPrompt, without extra replay or offline fine-tuning, so that differences in performance come from their prompt mechanisms rather than from additional data passes.

**GCL baselines (MVP, MISA).** MVP (Moon et al., 2023) and MISA (Kang et al., 2025) are implemented on top of the same ViT-B/16 backbone and Si-Blurry streams. We follow their official configurations for expert/prompt structures and initialization, while enforcing the unified prompt configuration and online training protocol of Sec. C.1. As in prior work, they maintain session-wise experts or prompts, but do not use any privileged task oracle beyond the evolving data stream; their routing structures can be interpreted in the same way as FlyPrompt's experts discussed in Sec. D. For fairness, MVP and MISA also use the batch-seen class logit mask in Sec. 3.2, whose effect is ablated alongside other mask types in Tab. 15.

**Offline PTM-based CL methods (S-Prompt++, HiDe-Prompt/LoRA/Adapter, NoRGa, SD-LoRA).** S-Prompt++ (Wang et al., 2022b) introduces prompt experts with a mixture-of-experts (MoE) structure with linear gating, while HiDe-Prompt/LoRA/Adapter (Wang et al., 2023a) and NoRGa (Le et al., 2024) build hierarchical decompositions and stronger MoE-based routing on top of S-Prompt++, and SD-LoRA Wu et al. (2025) leverages structured low-rank adapters by decomposing expert LoRA into learnable amplitude and fixed direction; all of these are originally designed for offline or task/class-incremental CL. Specifically, HiDe and NoRGa consist of a two-stage TIP+WTP (task-ID prediction then within-task prediction) pipeline. To make them compatible with GCL and Si-Blurry, we adapt their TIP step as follows: when the method predicts a class, it is allowed to activate *all* prompts corresponding to the candidate task IDs associated with that class, and we count the prediction as correct if *any* activated prompt outputs the true label. This is an intentionally favorable modification for these baselines. In addition, any feature statistics required by their alignment modules (e.g., for HiDe or NoRGa) are accumulated online from the stream, rather than being computed from stored per-task datasets. Quantitative comparisons of these adapted offline PTM-based methods with FlyPrompt are reported in Tab. 12.

**Analytic random-projection baselines (RanPAC variants).** RanPAC (McDonnell et al., 2024) was originally proposed for offline class-incremental learning, where the PTM is fine-tuned on the first task, frozen, and all task-1 features are recomputed and stored to form a stable Gram matrix for closed-form ridge regression; this protocol is incompatible with single-pass GCL and blurry task boundaries. To ensure a fair analytic baseline, we adapt RanPAC into three GCL-compliant variants: **RanPAC**[†] fine-tunes the PTM on the first Si-Blurry session without storing features and then solves a closed-form ridge classifier on the resulting random-feature representations, serving as the main analytic random-projection baseline in our ablations; **RanPAC**[‡] freezes the PTM during the first session and stores all features to approximate the original offline setting while still respecting the single-pass constraint on labels; **RanPAC**[*] simultaneously fine-tunes the PTM and collects features during the first session, which yields an ill-conditioned Gram matrix due to representation drift but offers an optimistic upper bound on analytic-classifier performance under our setting. As summarized in Tab. 13, FlyPrompt consistently outperforms all RanPAC variants across datasets.

# D  DISCUSSION OF TASK BOUNDARY IN SI-BLURRY

Table 8: Performance comparison of different numbers of tasks and experts for GCL methods on CIFAR-100 dataset over Sup-21K. "task-correlated" indicates that the initialization and training of expert parameters are aligned with task/sessions. $w$ denotes the sample budget (window size) of each expert when methods adopt a self-triggered expert allocation mechanism. All results are reported as an average of five parallel runs ($\pm$ standard deviation) with different random seeds.

| # of Tasks | # of Experts | MVP | | MISA | | FlyPrompt | |
|---|---|---|---|---|---|---|---|
| | | $A_{auc}(\%,\uparrow)$ | $A_{last}(\%,\uparrow)$ | $A_{auc}(\%,\uparrow)$ | $A_{last}(\%,\uparrow)$ | $A_{auc}(\%,\uparrow)$ | $A_{last}(\%,\uparrow)$ |
| 5 | 5 (task-correlated) | $\underline{67.74}_{\pm4.96}$ | $63.22_{\pm0.69}$ | $\underline{80.35}_{\pm2.39}$ | $80.75_{\pm1.24}$ | $83.24_{\pm2.23}$ | $\mathbf{86.76}_{\pm0.73}$ |
| | 5 ($w=10000$) | $\mathbf{67.75}_{\pm4.96}$ | $63.22_{\pm0.76}$ | $\mathbf{80.60}_{\pm2.06}$ | $\mathbf{81.73}_{\pm1.17}$ | $\mathbf{83.67}_{\pm2.23}$ | $\underline{85.78}_{\pm0.62}$ |
| | 10 ($w=5000$) | $67.23_{\pm5.06}$ | $\underline{63.47}_{\pm0.78}$ | $79.95_{\pm1.86}$ | $\underline{81.32}_{\pm1.15}$ | $\underline{83.40}_{\pm2.28}$ | $85.43_{\pm0.32}$ |
| | 20 ($w=2500$) | $67.19_{\pm4.80}$ | $\mathbf{64.06}_{\pm1.48}$ | $79.94_{\pm1.75}$ | $81.03_{\pm0.92}$ | $82.26_{\pm1.94}$ | $84.48_{\pm0.64}$ |
| 10 | 10 (task-correlated) | $58.23_{\pm3.42}$ | $61.13_{\pm6.00}$ | $75.71_{\pm3.10}$ | $80.22_{\pm0.47}$ | $\mathbf{77.65}_{\pm3.05}$ | $\mathbf{84.87}_{\pm0.50}$ |
| | 5 ($w=10000$) | $\mathbf{58.49}_{\pm3.57}$ | $\underline{61.98}_{\pm6.19}$ | $\mathbf{76.25}_{\pm3.10}$ | $\underline{80.62}_{\pm0.62}$ | $\underline{77.28}_{\pm2.79}$ | $\underline{84.63}_{\pm0.50}$ |
| | 10 ($w=5000$) | $58.19_{\pm3.42}$ | $60.80_{\pm6.48}$ | $75.69_{\pm3.11}$ | $80.18_{\pm0.41}$ | $76.96_{\pm3.23}$ | $84.03_{\pm0.39}$ |
| | 20 ($w=2500$) | $\underline{58.46}_{\pm3.34}$ | $\mathbf{62.15}_{\pm5.28}$ | $\underline{75.73}_{\pm2.80}$ | $\mathbf{80.66}_{\pm0.61}$ | $76.47_{\pm3.38}$ | $83.39_{\pm0.55}$ |
| 20 | 20 (task-correlated) | $\underline{56.52}_{\pm3.20}$ | $\mathbf{56.87}_{\pm5.18}$ | $\underline{73.96}_{\pm0.72}$ | $\mathbf{77.98}_{\pm1.17}$ | $75.87_{\pm1.93}$ | $\underline{81.98}_{\pm1.12}$ |
| | 5 ($w=10000$) | $\mathbf{56.59}_{\pm3.34}$ | $56.58_{\pm6.76}$ | $\mathbf{74.27}_{\pm0.97}$ | $\underline{77.89}_{\pm1.60}$ | $\mathbf{76.12}_{\pm1.57}$ | $81.90_{\pm0.32}$ |
| | 10 ($w=5000$) | $56.35_{\pm3.38}$ | $56.80_{\pm6.57}$ | $73.86_{\pm0.82}$ | $77.59_{\pm1.42}$ | $\mathbf{76.12}_{\pm1.21}$ | $\mathbf{82.21}_{\pm0.80}$ |
| | 20 ($w=2500$) | $56.48_{\pm3.15}$ | $\underline{56.86}_{\pm5.39}$ | $73.66_{\pm0.85}$ | $77.70_{\pm1.24}$ | $75.36_{\pm1.65}$ | $81.13_{\pm0.55}$ |

GCL (Buzzega et al., 2020) is defined by a single-pass, non-stationary data stream without task boundaries during training and without a task oracle at test time. The Si-Blurry benchmark (Moon et al., 2023) that we adopt has been carefully analyzed in subsequent work (Kang et al., 2025): by controlling the disjoint-class ratio $r_D$ and blurry-sample ratio $r_B$, it generates streams where (i) the number of active classes can vary across sessions, (ii) classes may reoccur across sessions, and (iii) the number of samples per class and per session is randomized (Mi et al., 2020) In particular, when $r_D$ approaching 0, the nominal "task" or "session" index becomes decorrelated from distributional changes. These properties ensure that Si-Blurry conforms to the core GCL assumptions, rather than reducing to standard task-incremental CIL.

Within this setting, FlyPrompt does not assume any privileged boundary information beyond what is already used by prior GCL methods such as MVP and MISA. The "task" or "session" index provided by Si-Blurry is treated as a conceptual device to describe how the benchmark constructs streams, not as a supervision signal for the model. In our implementation, expert indices are aligned with nominal session identities purely for convenience: the same behavior can be reproduced by starting a new expert after a fixed number of observed samples or when a user-defined computational/storage budget is reached, without accessing the task index. Moreover, the total number of experts $T$ is not a hard-coded prior; matrices such as $Q \in \mathbb{R}^{M \times T}$ and the router head can be dynamically extended from $T$ to $T+1$ via zero-padding, analogous to adding classes in a standard classifier. Implementation details for how both GCL methods and offline PTM-based baselines are instantiated under this regime are summarized in Appendix C.2.

To empirically validate that FlyPrompt and comparable GCL baselines do not gain an advantage from Si-Blurry's session structure, we further compare task-aligned expert management with a self-triggered expert allocation mechanism. In the self-triggered setup, each method maintains a fixed sample budget and freezes the current expert while initializing a new one whenever the number of observed samples reaches a predefined threshold, fully decoupling expert updates from external task segmentation. We evaluate multiple combinations of nominal task counts (# of Tasks $= 5, 10, 20$) and expert budgets (# of Experts $= 5, 10, 20$) on CIFAR-100, with corresponding sample budgets chosen to cover the 50K training examples. As summarized in Tab. 8, self-triggered expert initialization achieves performance on par with, or slightly better than, session-aligned setups for MVP, MISA, and FlyPrompt across all tested budgets. This confirms that (i) task-switching signals offer no measurable benefit in this benchmark, and (ii) our expert management mechanism does not exploit any extra boundary information.

# E  THEORETICAL PROOFS

## E.1  PROOFS FOR REAR (THEOREM 1)

### NOTATION AND ASSUMPTIONS

We repeat and fix the notation used throughout the proof of Theorem 1:

1. $f_{\boldsymbol{\theta}} : \mathcal{X} \to \mathbb{R}^d$ is the given pretrained backbone. For an input $\boldsymbol{x}$ we write $\boldsymbol{h} = f_{\boldsymbol{\theta}}(\boldsymbol{x}) \in \mathbb{R}^d$.

2. $\boldsymbol{R} \in \mathbb{R}^{d \times M}$ is a random matrix with i.i.d. $\mathcal{N}(0,1)$ entries; the $j$-th column is $\boldsymbol{r}_j \in \mathbb{R}^d$.

3. The random-expanded feature map is
$$\boldsymbol{\varphi}(\boldsymbol{x}) = \big(\varphi_1(\boldsymbol{x}), \dots, \varphi_M(\boldsymbol{x})\big)^\top, \qquad \varphi_j(\boldsymbol{x}) = \sigma(\boldsymbol{h}^\top \boldsymbol{r}_j).$$

4. Assume embedding-boundedness: $\|\boldsymbol{h}\|_2 \leq H$ for all $\boldsymbol{x}$. (This can be enforced in practice by layer-norm or clipping.)

5. Activation $\sigma : \mathbb{R} \to \mathbb{R}$ is $L_\sigma$-Lipschitz and has linear growth $|\sigma(z)| \leq C(1 + |z|)$. ReLU satisfies these with $L_\sigma = 1$ and linear growth $C = 1$.

6. For training, we accumulate batches (or singletons) to form
$$\boldsymbol{G} = \sum_{i=1}^{N} \boldsymbol{\varphi}(\boldsymbol{x}_i) \boldsymbol{\varphi}(\boldsymbol{x}_i)^\top \in \mathbb{R}^{M \times M}, \qquad \boldsymbol{Q} = \sum_{i=1}^{N} \boldsymbol{\varphi}(\boldsymbol{x}_i) \boldsymbol{c}_i^\top \in \mathbb{R}^{M \times T},$$
where $\boldsymbol{c}_i \in \{0,1\}^T$ is the one-hot indicator of the target expert; see Eq. (2)).

7. Ridge solution (router):
$$\widehat{\boldsymbol{U}}^\top = (\boldsymbol{G} + \lambda \boldsymbol{I})^{-1} \boldsymbol{Q}, \qquad \lambda > 0,$$
with regularization parameter $\lambda$ as in Eq. (4).

8. For the theoretical analysis, we treat the training pairs $(\boldsymbol{x}_i, y_i)$ as i.i.d. draws from an underlying distribution over $\mathcal{X} \times \{1, \dots, T\}$, with a fixed and finite number of experts $T$.

9. For the margin-based routing-accuracy corollary below, we additionally assume that there exists a margin $\gamma > 0$ such that, for the population minimizer $U^\star$ and almost every input $\boldsymbol{x}$, the score of the correct expert $t^\star(\boldsymbol{x})$ satisfies $s_{U^\star}(\boldsymbol{x})_{t^\star(\boldsymbol{x})} \geq s_{U^\star}(\boldsymbol{x})_t + \gamma$ for all $t \neq t^\star(\boldsymbol{x})$, where $s_U(\boldsymbol{x}) := \boldsymbol{\varphi}(\boldsymbol{x}) U^\top$.[2]

At the population level we consider the regularized squared risk
$$\mathcal{R}(U) := \mathbb{E} \big\| s_U(X) - C \big\|_2^2 + \lambda \|U\|_F^2,$$

where $(X, C)$ denotes a random variable pair drawn from the same distribution as the training examples $(\boldsymbol{x}_i, \boldsymbol{c}_i)$, with $X \in \mathcal{X}$ and $C \in \{0,1\}^T$ is the one-hot indicator of the target expert. We write $s_U(\boldsymbol{x}) := \boldsymbol{\varphi}(\boldsymbol{x}) U^\top$ for the router scores as in Eq. (5). The minimizer of $\mathcal{R}$ in the kernel-induced feature space is precisely the $U^\star$ appearing below.

We then state the complete Theorem 1 here based on the above assumptions:

**Theorem** (REAR, full). *Under the standing assumptions above, form the online statistics $\boldsymbol{G}, \boldsymbol{Q}$ as in Eq. (2) and the ridge solution $\widehat{\boldsymbol{U}}$ as in Eq. (4). Let $N$ be the total number of samples used to form $\boldsymbol{G}, \boldsymbol{Q}$ and let $U^\star$ denote the population regularized minimizer in the feature space induced by the kernel $k(\boldsymbol{h}, \boldsymbol{h}') = \mathbb{E}_{\boldsymbol{r}}[\sigma(\boldsymbol{h}^\top \boldsymbol{r}) \sigma(\boldsymbol{h}'^\top \boldsymbol{r})]$. Then for any $\delta \in (0, 1)$, with probability at least $1 - \delta$ (over $\boldsymbol{R}$ and the training samples), the excess (population) squared risk decomposes as*
$$\mathcal{R}(\widehat{\boldsymbol{U}}) - \mathcal{R}(U^\star) \leq \mathcal{E}_{\text{feat}}(M, \delta) + \mathcal{E}_{\text{estim}}(N, \lambda, \delta) + \mathcal{E}_{\text{reg}}(\lambda),$$

*where, for universal constants $C_i$ (depending on $H, L_\sigma, C$),*

$$\mathcal{E}_{\text{feat}}(M, \delta) \leq C_1 \sqrt{\frac{\log(N/\delta)}{M}}, \quad \mathcal{E}_{\text{estim}}(N, \lambda, \delta) \leq C_2 \frac{1}{\sqrt{N}} \cdot \frac{1}{\lambda}, \quad \mathcal{E}_{\text{reg}}(\lambda) \leq C_3 \lambda \|U^\star\|_F^2.$$

---

[2]This assumption is not needed for the excess-risk decomposition in Theorem 1 itself, but acts as a bridge.

### E.1.1 Lemma: random-feature concentration

**Lemma 1.** *Let $S = \{\boldsymbol{x}_1, \ldots, \boldsymbol{x}_N\}$ be the finite training set and write $\boldsymbol{h}_i = f_{\boldsymbol{\theta}}(\boldsymbol{x}_i)$. Define the kernel*

$$k(\boldsymbol{h}, \boldsymbol{h}') := \mathbb{E}_{\boldsymbol{r} \sim \mathcal{N}(0, \boldsymbol{I}_d)}\big[\sigma(\boldsymbol{h}^\top \boldsymbol{r})\sigma(\boldsymbol{h}'^\top \boldsymbol{r})\big].$$

*Then, under the standing assumptions, for any $\delta \in (0, 1)$ and any $\varepsilon \in (0, 1)$, if*

$$M \geq C_{\mathrm{rf}} \cdot \frac{\log(N^2/\delta)}{\varepsilon^2}$$

*(with $C_{\mathrm{rf}}$ depending only on $H, L_\sigma, C$ above), then with probability at least $1 - \delta$ over $\boldsymbol{R}$,*

$$\max_{1 \leq i,j \leq N} \Big| \frac{1}{M} \boldsymbol{\varphi}(\boldsymbol{x}_i)^\top \boldsymbol{\varphi}(\boldsymbol{x}_j) - k(\boldsymbol{h}_i, \boldsymbol{h}_j) \Big| \leq \varepsilon.$$

*Proof.* Fix a pair $(i, j)$. Write

$$Z_\ell := \varphi_\ell(\boldsymbol{x}_i)\varphi_\ell(\boldsymbol{x}_j) - \mathbb{E}[\varphi_\ell(\boldsymbol{x}_i)\varphi_\ell(\boldsymbol{x}_j)], \qquad \ell = 1, \ldots, M.$$

The $Z_\ell$ are independent (across $\ell$) mean-zero random variables because columns $\boldsymbol{r}_\ell$ are independent. We will apply Bernstein's inequality for sums of independent sub-exponential variables; to do so, we need a variance proxy and a uniform tail bound.

From the growth assumption $|\sigma(z)| \leq C(1 + |z|)$ and $\boldsymbol{r}_\ell \sim \mathcal{N}(0, \boldsymbol{I}_d)$, the marginal $\varphi_\ell(\boldsymbol{x}) = \sigma(\boldsymbol{h}_i^\top \boldsymbol{r}_\ell)$ is sub-Gaussian / sub-exponential: more precisely, since $\boldsymbol{h}_i^\top \boldsymbol{r}_\ell \sim \mathcal{N}(0, \|\boldsymbol{h}_i\|^2) \leq \mathcal{N}(0, H^2)$, we have for some constants $v, b$ (depending on $H, L_\sigma, C$) that $\mathbb{P}(|\varphi_\ell(\boldsymbol{x})| \geq t) \leq 2\exp(-ct)$ for large $t$; thus $\varphi_\ell(\boldsymbol{x})\varphi_\ell(\boldsymbol{x}')$ is sub-exponential with parameters bounded by functions of $H, L_\sigma, C$. Concretely, one can verify

$$\|Z_\ell\|_{\psi_1} \leq \tilde{b}$$

for a finite constant $\tilde{b}$ depending only on $H, L_\sigma, C$, where $\| \cdot \|_{\psi_1}$ denotes the standard sub-exponential Orlicz norm. Hence, applying Bernstein's inequality for sub-exponential variables yields, for any $\tau > 0$,

$$\mathbb{P}\Big( \Big| \sum_{\ell=1}^{M} Z_\ell \Big| \geq \tau \Big) \leq 2\exp\Big( -c\min\Big( \frac{\tau^2}{M\tilde{v}^2}, \frac{\tau}{\tilde{b}} \Big) \Big),$$

with constants $\tilde{v}, \tilde{b}, c > 0$ determined by the sub-exponential parameters.

Choose $\tau = M\varepsilon$. Plugging $\tau = M\varepsilon$ and requiring the RHS to be $\leq \delta/N^2$ (to union bound over all $\leq N^2$ pairs) yields the condition

$$M \geq C_{\mathrm{rf}} \frac{\log(N^2/\delta)}{\varepsilon^2}$$

for some $C_{\mathrm{rf}}$ (combining cases of Bernstein). This gives, for fixed pair $(i, j)$,

$$\mathbb{P}\Big( \Big| \frac{1}{M} \sum_{\ell=1}^{M} Z_\ell \Big| \geq \varepsilon \Big) \leq \frac{\delta}{N^2}.$$

Apply union bound over all $\leq N^2$ ordered pairs $(i, j)$. This yields the claimed uniform bound with probability at least $1 - \delta$. $\square$

**Remarks on applicability of Bernstein.** We used Bernstein for independent sub-exponential summands. The summands are independent across random-feature index $\ell$; sub-exponentiality follows from (i) Gaussianity of $\boldsymbol{r}_\ell$ and (ii) Lipschitz + linear-growth of $\sigma$. For ReLU (which is Lipschitz with linear growth) the same argument applies (moments of Gaussian tails control tails of $\sigma(\cdot)$).

### E.1.2 Lemma: ridge perturbation

We next show that when the empirical feature covariance concentrates around its population counterpart and the empirical cross-covariance concentrates, then the finite-sample ridge solution is close to the population ridge solution.

**Lemma 2.** *Let $\Phi \in \mathbb{R}^{N \times M}$ be the feature matrix with rows $\varphi(x_i)^\top$, define empirical covariance*

$$\widehat{\Sigma} = \frac{1}{N}\Phi^\top\Phi \in \mathbb{R}^{M \times M}, \qquad \widehat{b} = \frac{1}{N}\Phi^\top Y \in \mathbb{R}^{M \times T},$$

*where $Y \in \mathbb{R}^{N \times T}$ is the one-hot label matrix (or soft labels). Let the population quantities be $\Sigma = \mathbb{E}[\varphi(x)\varphi(x)^\top]$ and $b = \mathbb{E}[\varphi(x)c^\top]$. Denote population ridge solution*

$$U_\lambda^\star := (\Sigma + \lambda I)^{-1}b, \qquad \widehat{U}_\lambda := (\widehat{\Sigma} + \lambda I)^{-1}\widehat{b}.$$

*If $\|\widehat{\Sigma} - \Sigma\|_{\mathrm{op}} \leq \frac{\lambda}{2}$ and $\|\widehat{b} - b\|_F \leq \epsilon_b$, then*

$$\|\widehat{U}_\lambda - U_\lambda^\star\|_F \leq \frac{2}{\lambda}\epsilon_b + \frac{2}{\lambda^2}\|b\|_F\|\widehat{\Sigma} - \Sigma\|_{\mathrm{op}}.$$

*Proof.* We write

$$\widehat{U}_\lambda - U_\lambda^\star = (\widehat{\Sigma} + \lambda I)^{-1}(\widehat{b} - b) + \big[(\widehat{\Sigma} + \lambda I)^{-1} - (\Sigma + \lambda I)^{-1}\big]b.$$

For the first term use operator norm bound $\|(\widehat{\Sigma} + \lambda I)^{-1}\|_{\mathrm{op}} \leq 1/\lambda$ to get

$$\|(\widehat{\Sigma} + \lambda I)^{-1}(\widehat{b} - b)\|_F \leq \frac{1}{\lambda}\|\widehat{b} - b\|_F.$$

For the second term use the identity $A^{-1} - B^{-1} = A^{-1}(B - A)B^{-1}$ with $A = \widehat{\Sigma} + \lambda I$, $B = \Sigma + \lambda I$. Hence

$$\|A^{-1} - B^{-1}\|_{\mathrm{op}} \leq \|A^{-1}\|_{\mathrm{op}}\|A - B\|_{\mathrm{op}}\|B^{-1}\|_{\mathrm{op}} \leq \frac{1}{\lambda}\|\widehat{\Sigma} - \Sigma\|_{\mathrm{op}}\frac{1}{\lambda}.$$

Thus

$$\big\|\big[(\widehat{\Sigma} + \lambda I)^{-1} - (\Sigma + \lambda I)^{-1}\big]b\big\|_F \leq \frac{1}{\lambda^2}\|\widehat{\Sigma} - \Sigma\|_{\mathrm{op}}\|b\|_F.$$

Combining the two terms and tightening constants when $\|\widehat{\Sigma} - \Sigma\|_{\mathrm{op}} \leq \lambda/2$ gives the stated bound. $\qquad\square$

**Concentration of $\widehat{\Sigma}$ and $\widehat{b}$.** We next control the empirical covariance and cross-covariance. Recall

$$\widehat{\Sigma} = \frac{1}{N}\sum_{i=1}^N \varphi(x_i)\varphi(x_i)^\top, \qquad \Sigma = \mathbb{E}[\varphi(x)\varphi(x)^\top].$$

It is convenient to write

$$\widehat{\Sigma} - \Sigma = \sum_{i=1}^N X_i, \qquad X_i := \frac{1}{N}\Big(\varphi(x_i)\varphi(x_i)^\top - \Sigma\Big).$$

Each $X_i$ is self-adjoint and satisfies $\mathbb{E}[X_i] = 0$. Under the bounded-embedding and activation assumptions (cf. Lemma 1), there exists a constant $C_\varphi > 0$ (depending only on $(H, L_\sigma, C)$) such that $\|\varphi(x)\|_2 \leq C_\varphi$ almost surely. Hence, for every $i$,

$$\|X_i\|_{\mathrm{op}} \leq \frac{1}{N}\Big(\|\varphi(x_i)\varphi(x_i)^\top\|_{\mathrm{op}} + \|\Sigma\|_{\mathrm{op}}\Big) \leq \frac{1}{N}\big(C_\varphi^2 + \|\Sigma\|_{\mathrm{op}}\big) =: \frac{L_0}{N}.$$

Similarly, we can bound the "matrix variance" term

$$v^2 := \Big\|\sum_{i=1}^N \mathbb{E}[X_i^2]\Big\|_{\mathrm{op}} = N\Big\|\mathbb{E}[X_1^2]\Big\|_{\mathrm{op}} \leq \frac{N}{N^2}\Big\|\mathbb{E}\big[(\varphi(x)\varphi(x)^\top - \Sigma)^2\big]\Big\|_{\mathrm{op}} \leq \frac{V_0}{N},$$

for some constant $V_0 > 0$ depending only on $(H, L_\sigma, C)$. For completeness, we recall a standard matrix Bernstein inequality (Theorem 6.1 in Tropp (2012)): if $\{X_i\}_{i=1}^N$ are independent, mean-zero, self-adjoint matrices with $\|X_i\|_{\mathrm{op}} \leq L$ almost surely and

$$v^2 := \Big\| \sum_{i=1}^N \mathbb{E}[X_i^2] \Big\|_{\mathrm{op}},$$

then for all $t > 0$:

$$\mathbb{P}\Big( \Big\| \sum_{i=1}^N X_i \Big\|_{\mathrm{op}} \geq t \Big) \leq 2D \exp\Big( -\frac{t^2/2}{v^2 + Lt/3} \Big),$$

where $D$ denotes the matrix dimension (in our case $D = M$, the feature dimension). Applying this with $t = \varepsilon$ and our bounds $L \leq L_0/N$ and $v^2 \leq V_0/N$ gives

$$\mathbb{P}\big(\|\widehat{\Sigma} - \Sigma\|_{\mathrm{op}} \geq \varepsilon\big) = \mathbb{P}\Big( \Big\| \sum_{i=1}^N X_i \Big\|_{\mathrm{op}} \geq \varepsilon \Big) \leq 2M \exp\Big( -\frac{N\varepsilon^2/2}{V_0 + L_0\varepsilon/3} \Big).$$

For $\varepsilon \in (0, 1)$ the denominator in the exponent is bounded above by a constant multiple of $V_0$, so there exists $C' > 0$ (depending only on $(H, L_\sigma, C)$) such that, for all $\delta \in (0, 1)$, taking

$$\varepsilon = C' \sqrt{\frac{\log(M/\delta)}{N}}$$

ensures

$$\mathbb{P}\big(\|\widehat{\Sigma} - \Sigma\|_{\mathrm{op}} \geq \varepsilon\big) \leq \delta.$$

Equivalently, with probability at least $1 - \delta$,

$$\|\widehat{\Sigma} - \Sigma\|_{\mathrm{op}} \leq C' \sqrt{\frac{\log(M/\delta)}{N}}.$$

For the empirical cross-covariance $\widehat{b} = \frac{1}{N} \sum_{i=1}^N \boldsymbol{\varphi}(\boldsymbol{x}_i) \boldsymbol{c}_i^\top$ and its population counterpart $b = \mathbb{E}[\boldsymbol{\varphi}(\boldsymbol{x}) \boldsymbol{c}^\top]$ we apply the same argument column-wise (each column is an average of bounded sub-exponential vectors of length $M$) and obtain

$$\|\widehat{b} - b\|_F \leq C'' \sqrt{\frac{\log(T/\delta)}{N}},$$

for some constant $C'' > 0$ depending only on the same problem parameters. Adjusting constants to account for the two events and taking a union bound, we may assume that both inequalities hold simultaneously with probability at least $1 - \delta$. Choosing $N$ large enough to make $\|\widehat{\Sigma} - \Sigma\|_{\mathrm{op}} \leq \lambda/2$ and to make $\|\widehat{b} - b\|_F$ (denoted $\epsilon_b$ in Lemma 2) small then yields the desired estimation error term in Lemma 2.

### E.1.3 LEMMA: ONLINE STATISTICS IMPLEMENT BATCH RIDGE

**Lemma 3.** *If $\boldsymbol{G}$ and $\boldsymbol{Q}$ are formed by accumulating per-example contributions*

$$\boldsymbol{G} = \sum_{i=1}^N \boldsymbol{\varphi}(\boldsymbol{x}_i) \boldsymbol{\varphi}(\boldsymbol{x}_i)^\top, \qquad \boldsymbol{Q} = \sum_{i=1}^N \boldsymbol{\varphi}(\boldsymbol{x}_i) \boldsymbol{c}_i^\top,$$

*then the closed-form solution $\widehat{\boldsymbol{U}}^\top = (\boldsymbol{G} + \lambda \boldsymbol{I})^{-1} \boldsymbol{Q}$ equals the ridge regression solution computed in batch on features $\boldsymbol{\varphi}(\boldsymbol{x}_i)$ and labels $\boldsymbol{c}_i$. Moreover, if the online implementation maintains $(\boldsymbol{G} + \lambda \boldsymbol{I})^{-1}$ via rank-1 updates, numerical equivalence holds up to floating-point precision.*

*Proof.* This is algebraic: batch ridge with design matrix $\Phi$ and labels $Y$ solves $\widehat{U}^\top = (\Phi^\top \Phi + \lambda I)^{-1} \Phi^\top Y$. But $\Phi^\top \Phi = \sum_i \boldsymbol{\varphi}(\boldsymbol{x}_i) \boldsymbol{\varphi}(\boldsymbol{x}_i)^\top = \boldsymbol{G}$ and $\Phi^\top Y = \boldsymbol{Q}$. The equality follows. For incremental numerical maintenance of the inverse, standard Sherman–Morrison or Woodbury updates apply; also numerically stable Cholesky-updates are recommended when $M$ is large. $\square$

### E.1.4 COMBINING LEMMAS: PROOF OF THEOREM 1

*Proof.* The excess population risk decomposes as

$$\mathcal{R}(\widehat{U}) - \mathcal{R}(U^\star) = \underbrace{\mathcal{R}(\widehat{U}) - \mathcal{R}(U_\lambda^\star)}_{\text{estimation error}} + \underbrace{\mathcal{R}(U_\lambda^\star) - \mathcal{R}(U^\star)}_{\text{reg. bias}}.$$

The regularization bias is the usual ridge bias and yields the $\mathcal{E}_{\text{reg}}(\lambda)$ term; standard calculus shows it is bounded by $\lambda\|U^\star\|_F^2$ up to constant factors.

For the estimation error, apply Lemma 2 to relate the empirical ridge solution (which equals the online $\widehat{U}$ by Lemma 3) to the population ridge solution $U_\lambda^\star$. The two perturbation terms are controlled by $\|\widehat{\Sigma}-\Sigma\|_{\text{op}}$ and $\|\widehat{b}-b\|_F$, which in turn are bounded by the matrix Bernstein concentration bounds for $\widehat{\Sigma}$ and $\widehat{b}$ derived above. This yields the stated $O\left(\frac{1}{\sqrt{N}} \cdot \frac{1}{\lambda}\right)$ behavior for $\mathcal{E}_{\text{estim}}$ (explicit constants follow from the above bounds).

Finally, the approximation error due to random features is precisely Lemma 1: replacing the kernel $k$ by its Monte Carlo approximation using $M$ independent features introduces a uniform $O\left(\sqrt{\log(N/\delta)/M}\right)$ perturbation in inner products, which propagates to the excess risk as the term $\mathcal{E}_{\text{feat}}(M,\delta)$ displayed in Theorem 1. $\square$

**Margin-based routing accuracy.** Under the additional margin assumption in the REAR standing assumptions and a uniform bound $\|\boldsymbol{\varphi}(\boldsymbol{x})\|_2 \leq C_\varphi$ (for some constant $C_\varphi$ depending only on $(H, L_\sigma, C)$), the excess risk bound above can be converted into a bound on misrouting probability. Let $\hat{t}(\boldsymbol{x}) := \arg\max_t s_{\widehat{U}}(\boldsymbol{x})_t$ and $t^\star(\boldsymbol{x}) := \arg\max_t s_{U^\star}(\boldsymbol{x})_t$ denote the experts selected by $\widehat{U}$ and $U^\star$, respectively. Then

$$\mathbb{P}\left(\hat{t}(X) \neq t^\star(X)\right) \leq \frac{8C_\varphi^2}{\lambda\gamma^2}\left(\mathcal{R}(\widehat{U}) - \mathcal{R}(U^\star)\right).$$

*Proof.* On the event $\{\hat{t}(\boldsymbol{x}) \neq t^\star(\boldsymbol{x})\}$, the margin condition implies

$$\gamma \leq s_{U^\star}(\boldsymbol{x})_{t^\star(\boldsymbol{x})} - s_{U^\star}(\boldsymbol{x})_{\hat{t}(\boldsymbol{x})} \leq 2\max_t \left|s_{U^\star}(\boldsymbol{x})_t - s_{\widehat{U}}(\boldsymbol{x})_t\right|.$$

$$\text{Hence,} \quad \mathbf{1}\{\hat{t}(\boldsymbol{x}) \neq t^\star(\boldsymbol{x})\} \leq (2/\gamma)^2 \left\|s_{\widehat{U}}(\boldsymbol{x}) - s_{U^\star}(\boldsymbol{x})\right\|_2^2.$$

Given $s_U(\boldsymbol{x}) = \boldsymbol{\varphi}(\boldsymbol{x})U^\top$ and Cauchy–Schwarz yields:

$$\max_t |s_{U^\star}(\boldsymbol{x})_t - s_{\widehat{U}}(\boldsymbol{x})_t| \leq \|\boldsymbol{\varphi}(\boldsymbol{x})\|_2 \|\widehat{U} - U^\star\|_F \leq C_\varphi\|\widehat{U} - U^\star\|_F,$$

$$\Rightarrow \mathbf{1}\{\hat{t}(\boldsymbol{x}) \neq t^\star(\boldsymbol{x})\} \leq (2C_\varphi/\gamma)^2\|\widehat{U} - U^\star\|_F^2.$$

Since the regularized risk is defined as $\mathcal{R}(U) := \mathbb{E}\|s_U(X) - C\|_2^2 + \lambda\|U\|_F^2$, the quadratic ridge term $\lambda\|U\|_F^2$ makes $\mathcal{R}$ (at least) $\lambda$-strongly convex in $U$. In particular, strong convexity implies:

$$\mathcal{R}(\widehat{U}) - \mathcal{R}(U^\star) \geq (\lambda/2)\|\widehat{U} - U^\star\|_F^2,$$

$$\Rightarrow \|\widehat{U} - U^\star\|_F^2 \leq (2/\lambda)\left(\mathcal{R}(\widehat{U}) - \mathcal{R}(U^\star)\right).$$

Combining these inequalities and taking expectations over $X$ gives the stated bound. $\square$

### E.2 PROOFS FOR TE$^2$ (THEOREM 2)

### E.2.1 NOTATION AND ASSUMPTIONS

We reuse notation from the main text. For a fixed expert (prompt/head), we denote:

- $\boldsymbol{W}_t^\star \in \mathbb{R}^{|\mathcal{Y}|\times D}$: the (population) time-$t$ optimal linear parameter (in the chosen feature space) for that expert.

- $W_t$: the instantaneous (online) estimator after observing the $t$-th update; we model $W_t = W_t^\star + \boldsymbol{\xi}_t$.

- EMA head with decay $\alpha \in (0, 1)$:

$$\widetilde{W}_t^{(\alpha)} = (1 - \alpha) \sum_{k \geq 0} \alpha^k \, W_{t-k}, \qquad L(\alpha) = \frac{1}{1-\alpha}.$$

- Discounted path length (drift measure):

$$P_t := \sum_{j \geq 1} \gamma^{j-1} \big\| W_{t-j+1}^\star - W_{t-j}^\star \big\|,$$

where we define the discount factor to match the EMA decay, i.e., $\gamma = \alpha$ or comparable. This is a discounted analogue of the standard path length / variation measure used in dynamic regret (Zinkevich, 2003; Besbes et al., 2015).

- Standing conditions: finite $P_t$, zero-mean noise with bounded variance:

$$\mathbb{E}[\boldsymbol{\xi}_t] = \mathbf{0}, \qquad \mathbb{E}\|\boldsymbol{\xi}_t\|^2 \leq \zeta^2.$$

- (Optional, for classification calibration) A margin $\Delta > 0$ and a Lipschitz map-to-logits with constant $C_f$ imply a bound on 0/1 error from parameter MSE.

We then state the complete Theorem 2 here based on the above assumptions:

**Theorem** ($\text{TE}^2$, full). *Under the standing conditions above, fix $t$ and an EMA decay $\alpha$ with $L = 1/(1-\alpha)$. There exist constants $C_1, C_2 > 0$ such that*

$$\mathbb{E}\big\|\widetilde{W}_t^{(\alpha)} - W_t^\star\big\|^2 \; \leq \; C_1 \frac{\zeta^2}{L} \; + \; C_2 \, (L \, P_t)^2.$$

*Moreover, if we keep a geometric grid of windows $\{L_i\}_{i=1}^m$ with ratio $r > 1$ (e.g., $L_i = r^{i-1}$), then for every $t$ there exists an index $i_t$ with*

$$\mathbb{E}\big\|\widetilde{W}_t^{(\alpha_{i_t})} - W_t^\star\big\|^2 \; \leq \; c(r) \min_{L \geq 1} \Big\{ C_1 \frac{\zeta^2}{L} + C_2 \, (L \, P_t)^2 \Big\},$$

*where $c(r)$ depends only on the grid ratio $r$ (one can take, for example, $c(r) = \max(r^2, r)$ by the argument below). Additionally, if a margin $\Delta > 0$ holds and the logits map is Lipschitz with constant $C_f$, then the above parameter MSE implies a classification error bound $O\big((C_f \varepsilon / \Delta)^2\big)$ whenever $\mathbb{E}\|\widetilde{W}_t^{(\alpha)} - W_t^\star\|^2 \leq \varepsilon^2$.*

### E.2.2   Proof of Theorem 2

*Proof.* The proof proceeds via a variance–bias decomposition and a geometric grid selection argument, followed by a calibration from parameter MSE to classification error.

We start by decomposing the error into the variance term. Define the difference between the EMA and the population optimum

$$\widetilde{W}_t^{(\alpha)} - \overline{W}_t^\star = (1 - \alpha) \sum_{k \geq 0} \alpha^k \, \boldsymbol{\xi}_{t-k},$$

$$\overline{W}_t^\star := (1 - \alpha) \sum_{k \geq 0} \alpha^k \, W_{t-k}^\star,$$

where $\overline{W}_t^\star$ is the EMA of the population optima.

Since we have $\mathbb{E}[\boldsymbol{\xi}_t] = \mathbf{0}$ and $\mathbb{E}\|\boldsymbol{\xi}_t\|^2 \leq \zeta^2$, and the EMA weights are $a_k = (1 - \alpha)\alpha^k$, we can compute their squared sum explicitly:

$$\sum_{k \geq 0} a_k^2 = (1 - \alpha)^2 \sum_{k \geq 0} \alpha^{2k} = \frac{(1-\alpha)^2}{1 - \alpha^2} = \frac{1 - \alpha}{1 + \alpha} \leq \frac{1}{L},$$

Therefore,

$$\mathbb{E}\big\|\widetilde{\boldsymbol{W}}_t^{(\alpha)} - \overline{\boldsymbol{W}}_t^\star\big\|^2 = \mathbb{E}\Big\|\sum_{k\geq 0} a_k \boldsymbol{\xi}_{t-k}\Big\|^2 \leq \zeta^2 \sum_{k\geq 0} a_k^2 \leq \frac{\zeta^2}{L},$$

which matches the variance term $C_1 \zeta^2 / L$ in Theorem 2 (with $C_1$ absorbing the constant factor). Next, we address the bias term. The difference between the EMA of the population optima and the actual population optimum is given by:

$$\overline{\boldsymbol{W}}_t^\star - \boldsymbol{W}_t^\star = \sum_{k\geq 0} a_k (\boldsymbol{W}_{t-k}^\star - \boldsymbol{W}_t^\star).$$

Reordering sums yields:

$$\big\|\overline{\boldsymbol{W}}_t^\star - \boldsymbol{W}_t^\star\big\| \leq \sum_{j\geq 1}\Big(\sum_{k\geq j} a_k\Big) \|\boldsymbol{W}_{t-j+1}^\star - \boldsymbol{W}_{t-j}^\star\| = \sum_{j\geq 1} \alpha^j \Delta_{t-j+1}, \tag{11}$$
$$\text{where} \quad \Delta_u = \|\boldsymbol{W}_u^\star - \boldsymbol{W}_{u-1}^\star\|.$$

Using the discounted path length $P_t$ with the same discount factor $\gamma = \alpha$ as in the EMA definition, we can simply rewrite the above bound as

$$\big\|\overline{\boldsymbol{W}}_t^\star - \boldsymbol{W}_t^\star\big\| = \sum_{j\geq 1} \alpha^j \Delta_{t-j+1} = \alpha \sum_{j\geq 1} \alpha^{j-1}\Delta_{t-j+1} = \alpha P_t \leq L P_t,$$

where we used $L = 1/(1-\alpha) \geq \alpha$. Consequently, the squared bias admits the explicit bound

$$\big\|\overline{\boldsymbol{W}}_t^\star - \boldsymbol{W}_t^\star\big\|^2 \leq (L P_t)^2,$$

which corresponds to the term $C_2 (L P_t)^2$ in Theorem 2.

Then, by combining the variance and bias terms, we apply the inequality $\|a+b\|^2 \leq 2\|a\|^2 + 2\|b\|^2$, which yields the claimed MSE bound:

$$\mathbb{E}\left[\big\|\widetilde{\boldsymbol{W}}_t^{(\alpha)} - \overline{\boldsymbol{W}}_t^\star\big\|^2\right] \leq 2\frac{\zeta^2}{L} + 2(L P_t)^2.$$

Therefore, in Theorem 2 we may take the explicit choice $C_1 = C_2 = 2$. If we allow $\gamma$ to differ slightly from $\alpha$, the constant $C_2$ becomes $(\alpha/\gamma)^2$ (bounded if we restrict $\gamma \in [(1-\epsilon)\alpha, (1+\epsilon)\alpha]$).

For the geometric bank, it is convenient to write the bound in the generic form

$$f(L) := \frac{A}{L} + B(L P_t)^2, \qquad A \asymp \zeta^2, \ B \asymp 1.$$

A direct derivative calculation of $f'(L)$ shows that the minimizer over $L > 0$ is

$$f'(L) = -\frac{A}{L^2} + 2B P_t^2 L \Rightarrow L^\star = \Big(\frac{A}{2B P_t^2}\Big)^{1/3}.$$

If we maintain a geometric grid $L_i = r^{i-1}$ with ratio $r > 1$, then for any $L^\star$ there exists an index $i_t$ such that $L_{i_t} \in [L^\star/r, r L^\star]$. Writing $L = L^\star \eta$ with $\eta \in [1/r, r]$ and using the optimality condition $A/L^\star = 2B(L^\star P_t)^2$, we obtain

$$\frac{f(L)}{f(L^\star)} = \frac{2B(L^\star P_t)^2/\eta + \eta^2 B(L^\star P_t)^2}{3B(L^\star P_t)^2} = \frac{2/\eta + \eta^2}{3}.$$

The right-hand side is maximized over $\eta \in [1/r, r]$ at one of the endpoints; a simple bound yields

$$\sup_{\eta\in[1/r,r]} \frac{2/\eta + \eta^2}{3} \leq \max\Big(\frac{2}{3r} + \frac{r^2}{3}, \frac{2r}{3} + \frac{1}{3r^2}\Big) \leq \max(r^2, r).$$

Therefore $f(L_{i_t}) \leq c(r) f(L^\star)$ with the explicit choice $c(r) = \max(r^2, r)$ used in Theorem 2 (obviously $c(r) = r^2$, given $r > 1$ in our case.)

From the general classification error's view, by the Lipschitz-to-logit condition, the induced logit error is at most:

$$C_f \|\widetilde{\boldsymbol{W}} - \boldsymbol{W}^\star\|.$$

Under the margin condition ($\Delta > 0$), the standard margin-to-0/1 calibration gives an error bound $O((C_f \varepsilon/\Delta)^2)$ when $\mathbb{E}\|\widetilde{\boldsymbol{W}} - \boldsymbol{W}^\star\|^2 \leq \varepsilon^2$. $\qquad\square$

# F ADDITIONAL EXPERIMENT RESULTS

## F.1 EXPERIMENT WITH DISJOINT AND BLURRY SETUP VARIANTS

Table 9: Performance comparison of $r_D$ variants in FlyPrompt. We fixed $r_B = 10\%$ and use Sup-21K backbone PTM. All results are reported as an average of five parallel runs ($\pm$ standard deviation) with different random seeds.

| $r_D$(%) | Method | CIFAR-100 | | ImageNet-R | | CUB-200 | |
|---|---|---|---|---|---|---|---|
| | | $A_{auc}(\%,\uparrow)$ | $A_{last}(\%,\uparrow)$ | $A_{auc}(\%,\uparrow)$ | $A_{last}(\%,\uparrow)$ | $A_{auc}(\%,\uparrow)$ | $A_{last}(\%,\uparrow)$ |
| 0 (pure blurry) | L2P | $73.64_{\pm1.69}$ | $81.71_{\pm1.28}$ | $42.97_{\pm1.91}$ | $42.53_{\pm0.79}$ | $64.62_{\pm3.37}$ | $62.72_{\pm3.32}$ |
| | DualPrompt | $72.16_{\pm2.40}$ | $77.52_{\pm1.43}$ | $45.08_{\pm3.48}$ | $42.34_{\pm1.55}$ | $65.42_{\pm3.53}$ | $62.84_{\pm2.21}$ |
| | CODA-P | $72.33_{\pm5.84}$ | $78.56_{\pm4.46}$ | $50.31_{\pm4.42}$ | $48.88_{\pm3.46}$ | $66.64_{\pm3.42}$ | $63.70_{\pm2.48}$ |
| | MVP | $67.10_{\pm2.33}$ | $74.94_{\pm7.45}$ | $39.04_{\pm1.49}$ | $32.28_{\pm3.12}$ | $56.86_{\pm3.71}$ | $55.50_{\pm6.71}$ |
| | MISA | $78.38_{\pm1.49}$ | $83.04_{\pm1.44}$ | $51.78_{\pm3.22}$ | $47.64_{\pm0.64}$ | $67.38_{\pm3.25}$ | $64.49_{\pm2.55}$ |
| | FlyPrompt (ours) | $\mathbf{80.12}_{\pm1.38}$ | $\mathbf{87.11}_{\pm0.52}$ | $\mathbf{55.44}_{\pm1.82}$ | $\mathbf{55.89}_{\pm0.86}$ | $\mathbf{71.60}_{\pm3.49}$ | $\mathbf{74.72}_{\pm1.69}$ |
| 50 (mixed) | L2P | $76.23_{\pm2.73}$ | $79.11_{\pm1.43}$ | $44.40_{\pm1.03}$ | $42.03_{\pm1.72}$ | $64.30_{\pm2.18}$ | $61.42_{\pm2.13}$ |
| | DualPrompt | $76.04_{\pm3.32}$ | $76.62_{\pm0.74}$ | $46.13_{\pm1.94}$ | $40.80_{\pm1.04}$ | $65.03_{\pm2.24}$ | $62.43_{\pm1.78}$ |
| | CODA-P | $79.13_{\pm3.06}$ | $80.91_{\pm0.70}$ | $51.87_{\pm2.81}$ | $48.09_{\pm2.75}$ | $66.01_{\pm2.20}$ | $62.90_{\pm2.46}$ |
| | MVP | $67.74_{\pm4.96}$ | $63.22_{\pm0.69}$ | $39.50_{\pm1.41}$ | $32.63_{\pm3.95}$ | $54.69_{\pm3.14}$ | $50.07_{\pm3.86}$ |
| | MISA | $80.35_{\pm2.39}$ | $80.75_{\pm1.24}$ | $51.52_{\pm2.09}$ | $45.08_{\pm1.43}$ | $65.40_{\pm3.01}$ | $60.20_{\pm1.82}$ |
| | FlyPrompt (ours) | $\mathbf{83.24}_{\pm2.23}$ | $\mathbf{86.76}_{\pm0.73}$ | $\mathbf{56.58}_{\pm1.47}$ | $\mathbf{55.27}_{\pm0.91}$ | $\mathbf{70.64}_{\pm2.85}$ | $\mathbf{73.40}_{\pm1.88}$ |
| 100 (disjoint) | L2P | $82.98_{\pm0.72}$ | $78.79_{\pm0.95}$ | $45.48_{\pm0.71}$ | $43.12_{\pm0.75}$ | $71.74_{\pm3.58}$ | $61.52_{\pm1.82}$ |
| | DualPrompt | $81.12_{\pm2.10}$ | $75.94_{\pm0.37}$ | $46.79_{\pm2.00}$ | $41.42_{\pm0.63}$ | $72.81_{\pm3.80}$ | $62.46_{\pm0.97}$ |
| | CODA-P | $82.68_{\pm3.99}$ | $77.97_{\pm3.19}$ | $53.01_{\pm3.01}$ | $49.31_{\pm2.87}$ | $73.95_{\pm4.10}$ | $63.90_{\pm0.94}$ |
| | MVP | $74.92_{\pm1.10}$ | $56.17_{\pm2.98}$ | $40.43_{\pm0.51}$ | $28.32_{\pm5.55}$ | $63.41_{\pm2.52}$ | $43.80_{\pm2.60}$ |
| | MISA | $85.67_{\pm1.01}$ | $81.04_{\pm1.02}$ | $53.88_{\pm1.99}$ | $47.63_{\pm0.78}$ | $74.27_{\pm3.56}$ | $62.97_{\pm1.44}$ |
| | FlyPrompt (ours) | $\mathbf{88.25}_{\pm0.90}$ | $\mathbf{85.51}_{\pm0.64}$ | $\mathbf{57.41}_{\pm0.95}$ | $\mathbf{55.69}_{\pm0.33}$ | $\mathbf{78.14}_{\pm3.62}$ | $\mathbf{74.34}_{\pm0.74}$ |

Table 10: Performance comparison of $r_B$ variants in FlyPrompt. We fixed $r_D = 50\%$ and use Sup-21K PTM backbone. All results are reported as an average of five parallel runs ($\pm$ standard deviation) with different random seeds.

| $r_B$(%) | Method | CIFAR-100 | | ImageNet-R | | CUB-200 | |
|---|---|---|---|---|---|---|---|
| | | $A_{auc}(\%,\uparrow)$ | $A_{last}(\%,\uparrow)$ | $A_{auc}(\%,\uparrow)$ | $A_{last}(\%,\uparrow)$ | $A_{auc}(\%,\uparrow)$ | $A_{last}(\%,\uparrow)$ |
| 10 | L2P | $76.23_{\pm2.73}$ | $79.11_{\pm1.43}$ | $44.40_{\pm1.03}$ | $42.03_{\pm1.72}$ | $64.30_{\pm2.18}$ | $61.42_{\pm2.13}$ |
| | DualPrompt | $76.04_{\pm3.32}$ | $76.62_{\pm0.74}$ | $46.13_{\pm1.94}$ | $40.80_{\pm1.04}$ | $65.03_{\pm2.24}$ | $62.43_{\pm1.78}$ |
| | CODA-P | $79.13_{\pm3.06}$ | $80.91_{\pm0.70}$ | $51.87_{\pm2.81}$ | $48.09_{\pm2.75}$ | $66.01_{\pm2.20}$ | $62.90_{\pm2.46}$ |
| | MVP | $67.74_{\pm4.96}$ | $63.22_{\pm0.69}$ | $39.50_{\pm1.41}$ | $32.63_{\pm3.95}$ | $54.69_{\pm3.14}$ | $50.07_{\pm3.86}$ |
| | MISA | $80.35_{\pm2.39}$ | $80.75_{\pm1.24}$ | $51.52_{\pm2.09}$ | $45.08_{\pm1.43}$ | $65.40_{\pm3.01}$ | $60.20_{\pm1.82}$ |
| | FlyPrompt (ours) | $\mathbf{83.24}_{\pm2.23}$ | $\mathbf{86.76}_{\pm0.73}$ | $\mathbf{56.58}_{\pm1.47}$ | $\mathbf{55.27}_{\pm0.91}$ | $\mathbf{70.64}_{\pm2.85}$ | $\mathbf{73.40}_{\pm1.88}$ |
| 30 | L2P | $78.48_{\pm0.92}$ | $80.13_{\pm0.87}$ | $43.32_{\pm0.79}$ | $42.27_{\pm1.40}$ | $63.67_{\pm2.03}$ | $63.60_{\pm3.09}$ |
| | DualPrompt | $77.76_{\pm1.65}$ | $77.50_{\pm0.49}$ | $45.11_{\pm1.09}$ | $41.01_{\pm0.70}$ | $64.36_{\pm1.98}$ | $63.72_{\pm2.22}$ |
| | CODA-P | $81.50_{\pm1.20}$ | $82.65_{\pm0.72}$ | $50.55_{\pm2.24}$ | $47.58_{\pm2.81}$ | $65.89_{\pm1.42}$ | $64.97_{\pm2.83}$ |
| | MVP | $71.01_{\pm1.70}$ | $65.71_{\pm4.60}$ | $38.62_{\pm1.35}$ | $32.43_{\pm4.75}$ | $54.16_{\pm4.56}$ | $51.04_{\pm6.89}$ |
| | MISA | $82.54_{\pm1.08}$ | $82.50_{\pm0.68}$ | $51.69_{\pm0.94}$ | $47.09_{\pm1.16}$ | $67.13_{\pm1.72}$ | $66.53_{\pm2.39}$ |
| | FlyPrompt (ours) | $\mathbf{84.61}_{\pm1.25}$ | $\mathbf{86.89}_{\pm0.38}$ | $\mathbf{55.30}_{\pm0.86}$ | $\mathbf{55.43}_{\pm1.04}$ | $\mathbf{70.19}_{\pm2.01}$ | $\mathbf{74.25}_{\pm1.27}$ |
| 50 | L2P | $77.44_{\pm2.42}$ | $80.31_{\pm0.69}$ | $44.39_{\pm1.72}$ | $43.66_{\pm1.04}$ | $65.42_{\pm2.71}$ | $64.62_{\pm1.63}$ |
| | DualPrompt | $77.44_{\pm2.64}$ | $77.13_{\pm1.08}$ | $46.23_{\pm1.83}$ | $41.99_{\pm0.72}$ | $66.40_{\pm2.72}$ | $65.34_{\pm3.05}$ |
| | CODA-P | $81.39_{\pm2.18}$ | $83.10_{\pm0.97}$ | $53.05_{\pm1.60}$ | $50.20_{\pm1.74}$ | $67.88_{\pm2.26}$ | $65.44_{\pm2.32}$ |
| | MVP | $67.97_{\pm4.78}$ | $58.11_{\pm1.26}$ | $40.68_{\pm1.59}$ | $31.87_{\pm6.56}$ | $57.25_{\pm3.76}$ | $53.86_{\pm3.41}$ |
| | MISA | $81.81_{\pm2.29}$ | $82.51_{\pm0.38}$ | $53.27_{\pm1.71}$ | $48.32_{\pm0.84}$ | $68.68_{\pm2.47}$ | $66.84_{\pm2.27}$ |
| | FlyPrompt (ours) | $\mathbf{83.69}_{\pm1.81}$ | $\mathbf{86.31}_{\pm0.73}$ | $\mathbf{56.50}_{\pm1.87}$ | $\mathbf{55.59}_{\pm0.83}$ | $\mathbf{71.95}_{\pm1.92}$ | $\mathbf{74.03}_{\pm1.02}$ |

As detailed in Tabs. 9 and 10, FlyPrompt consistently outperforms baselines from the purely blurry regime ($r_D = 0$) to fully disjoint tasks ($r_D = 1$, namely, online CIL) and across different blurry ratios $r_B$, demonstrating robustness under extreme GCL configurations and superior versatility.

## F.2 GCL EVALUATION METRICS

Table 11: Average accuracy of 5 sessions and forgetting of different GCL methods over three datasets. All results are reported as an average of five runs (± standard deviation) with different random seeds.

| PTM | Method | CIFAR-100 | | ImageNet-R | | CUB-200 | |
|---|---|---|---|---|---|---|---|
| | | $A_{avg}(\%,\uparrow)$ | $F_{last}(\%,\downarrow)$ | $A_{avg}(\%,\uparrow)$ | $F_{last}(\%,\downarrow)$ | $A_{avg}(\%,\uparrow)$ | $F_{last}(\%,\downarrow)$ |
| Sup-21K | L2P | $75.41_{\pm2.75}$ | $11.53_{\pm1.44}$ | $47.82_{\pm0.95}$ | $19.16_{\pm1.92}$ | $65.44_{\pm3.30}$ | $29.15_{\pm3.19}$ |
| | DualPrompt | $75.96_{\pm2.56}$ | $11.42_{\pm0.91}$ | $49.90_{\pm2.34}$ | $\underline{18.24}_{\pm4.34}$ | $66.54_{\pm3.05}$ | $\underline{27.18}_{\pm2.91}$ |
| | CODA-P | $78.73_{\pm3.09}$ | $9.95_{\pm1.30}$ | $55.75_{\pm2.83}$ | $18.49_{\pm2.44}$ | $\underline{67.19}_{\pm2.98}$ | $27.43_{\pm3.42}$ |
| | MVP | $64.70_{\pm4.14}$ | $33.19_{\pm2.33}$ | $40.14_{\pm1.39}$ | $43.39_{\pm4.23}$ | $52.42_{\pm5.06}$ | $47.61_{\pm5.73}$ |
| | MISA | $\underline{79.67}_{\pm1.78}$ | $\underline{9.67}_{\pm1.39}$ | $\underline{55.78}_{\pm1.41}$ | $21.46_{\pm4.25}$ | $66.92_{\pm3.12}$ | $30.06_{\pm3.70}$ |
| | FlyPrompt (ours) | $\mathbf{82.72}_{\pm2.69}$ | $\mathbf{5.03}_{\pm1.16}$ | $\mathbf{60.38}_{\pm1.76}$ | $\mathbf{13.42}_{\pm1.74}$ | $\mathbf{72.22}_{\pm4.35}$ | $\mathbf{10.97}_{\pm1.52}$ |
| Sup-21K/1K | L2P | $60.98_{\pm10.04}$ | $14.88_{\pm6.27}$ | $50.21_{\pm3.09}$ | $35.88_{\pm4.51}$ | $43.72_{\pm5.19}$ | $35.05_{\pm9.31}$ |
| | DualPrompt | $66.13_{\pm4.34}$ | $19.18_{\pm5.13}$ | $\underline{56.53}_{\pm1.22}$ | $30.73_{\pm7.45}$ | $\underline{47.37}_{\pm5.07}$ | $35.40_{\pm8.16}$ |
| | CODA-P | $\underline{66.81}_{\pm4.87}$ | $19.82_{\pm6.85}$ | $55.01_{\pm1.66}$ | $35.58_{\pm5.52}$ | $45.38_{\pm4.99}$ | $35.73_{\pm8.95}$ |
| | MVP | $63.22_{\pm1.67}$ | $46.98_{\pm8.15}$ | $50.91_{\pm3.09}$ | $51.11_{\pm2.86}$ | $46.13_{\pm1.76}$ | $62.93_{\pm8.05}$ |
| | MISA | $60.11_{\pm8.35}$ | $\underline{11.38}_{\pm2.76}$ | $54.17_{\pm2.92}$ | $\underline{28.66}_{\pm8.19}$ | $43.33_{\pm5.30}$ | $\underline{33.95}_{\pm8.63}$ |
| | FlyPrompt (ours) | $\mathbf{76.86}_{\pm2.29}$ | $\mathbf{9.50}_{\pm2.33}$ | $\mathbf{64.41}_{\pm1.80}$ | $\mathbf{21.59}_{\pm4.28}$ | $\mathbf{55.12}_{\pm5.49}$ | $\mathbf{21.71}_{\pm3.56}$ |
| iBOT-21K | L2P | $52.73_{\pm7.78}$ | $\underline{12.58}_{\pm6.24}$ | $38.41_{\pm6.38}$ | $\underline{34.17}_{\pm7.54}$ | $14.93_{\pm2.30}$ | $\underline{19.09}_{\pm8.05}$ |
| | DualPrompt | $62.86_{\pm7.45}$ | $19.22_{\pm4.22}$ | $46.07_{\pm2.16}$ | $37.96_{\pm9.50}$ | $21.96_{\pm5.20}$ | $30.18_{\pm8.79}$ |
| | CODA-P | $59.59_{\pm7.94}$ | $22.20_{\pm5.61}$ | $\underline{49.47}_{\pm2.79}$ | $41.21_{\pm6.19}$ | $18.77_{\pm6.28}$ | $28.98_{\pm9.54}$ |
| | MVP | $61.48_{\pm2.55}$ | $50.23_{\pm11.27}$ | $44.01_{\pm4.18}$ | $62.17_{\pm6.10}$ | $\mathbf{30.92}_{\pm2.26}$ | $62.20_{\pm5.81}$ |
| | MISA | $\underline{62.87}_{\pm3.76}$ | $17.34_{\pm6.20}$ | $44.62_{\pm3.36}$ | $35.62_{\pm11.69}$ | $19.48_{\pm4.88}$ | $21.18_{\pm10.00}$ |
| | FlyPrompt (ours) | $\mathbf{73.08}_{\pm2.18}$ | $\mathbf{8.38}_{\pm1.52}$ | $\mathbf{59.40}_{\pm1.84}$ | $\mathbf{23.40}_{\pm3.20}$ | $\underline{30.78}_{\pm8.41}$ | $\mathbf{18.85}_{\pm4.15}$ |
| iBOT-1K | L2P | $48.97_{\pm6.18}$ | $\underline{16.47}_{\pm7.80}$ | $41.23_{\pm7.31}$ | $\underline{33.01}_{\pm8.96}$ | $19.75_{\pm3.57}$ | $\mathbf{21.26}_{\pm11.09}$ |
| | DualPrompt | $50.16_{\pm4.91}$ | $20.45_{\pm3.88}$ | $49.51_{\pm1.57}$ | $35.14_{\pm7.72}$ | $30.29_{\pm3.97}$ | $32.70_{\pm9.00}$ |
| | CODA-P | $55.84_{\pm5.28}$ | $22.72_{\pm6.32}$ | $\underline{53.49}_{\pm1.21}$ | $40.23_{\pm6.19}$ | $28.85_{\pm3.98}$ | $27.71_{\pm9.65}$ |
| | MVP | $\underline{56.41}_{\pm2.00}$ | $53.47_{\pm12.87}$ | $46.96_{\pm3.97}$ | $56.26_{\pm5.49}$ | $\underline{34.97}_{\pm1.44}$ | $62.61_{\pm6.88}$ |
| | MISA | $52.58_{\pm4.34}$ | $19.16_{\pm4.04}$ | $48.89_{\pm2.92}$ | $33.52_{\pm10.12}$ | $27.98_{\pm5.32}$ | $29.35_{\pm6.60}$ |
| | FlyPrompt (ours) | $\mathbf{64.94}_{\pm2.57}$ | $\mathbf{11.52}_{\pm3.71}$ | $\mathbf{63.77}_{\pm1.42}$ | $\mathbf{20.89}_{\pm3.75}$ | $\mathbf{38.05}_{\pm6.98}$ | $\underline{21.62}_{\pm3.82}$ |
| DINO-1K | L2P | $45.45_{\pm6.86}$ | $\underline{14.82}_{\pm6.81}$ | $38.98_{\pm7.34}$ | $\underline{33.00}_{\pm7.22}$ | $23.21_{\pm3.68}$ | $\underline{24.23}_{\pm9.93}$ |
| | DualPrompt | $49.65_{\pm5.46}$ | $18.69_{\pm5.27}$ | $47.16_{\pm1.05}$ | $37.89_{\pm7.65}$ | $29.48_{\pm5.88}$ | $31.35_{\pm10.61}$ |
| | CODA-P | $50.76_{\pm5.65}$ | $19.43_{\pm5.83}$ | $\underline{48.68}_{\pm2.79}$ | $41.26_{\pm6.20}$ | $29.43_{\pm4.89}$ | $32.69_{\pm10.65}$ |
| | MVP | $\underline{52.41}_{\pm1.48}$ | $54.70_{\pm12.47}$ | $44.45_{\pm4.11}$ | $56.54_{\pm6.00}$ | $\underline{34.71}_{\pm1.92}$ | $64.21_{\pm7.82}$ |
| | MISA | $49.81_{\pm3.67}$ | $17.76_{\pm4.28}$ | $46.61_{\pm2.24}$ | $34.38_{\pm10.60}$ | $27.46_{\pm5.11}$ | $25.77_{\pm10.77}$ |
| | FlyPrompt (ours) | $\mathbf{63.59}_{\pm4.56}$ | $\mathbf{9.04}_{\pm1.73}$ | $\mathbf{60.83}_{\pm1.57}$ | $\mathbf{20.65}_{\pm3.08}$ | $\mathbf{37.50}_{\pm8.33}$ | $\mathbf{19.47}_{\pm4.84}$ |
| MoCo-1K | L2P | $26.75_{\pm6.56}$ | $19.68_{\pm15.19}$ | $19.02_{\pm2.75}$ | $43.59_{\pm4.64}$ | $13.42_{\pm3.89}$ | $\underline{26.38}_{\pm11.63}$ |
| | DualPrompt | $49.61_{\pm7.76}$ | $\underline{14.79}_{\pm4.09}$ | $41.37_{\pm1.80}$ | $35.07_{\pm5.93}$ | $21.87_{\pm4.80}$ | $28.42_{\pm9.47}$ |
| | CODA-P | $48.26_{\pm5.59}$ | $16.97_{\pm6.15}$ | $44.71_{\pm2.43}$ | $43.27_{\pm2.27}$ | $22.33_{\pm3.94}$ | $30.90_{\pm10.77}$ |
| | MVP | $52.82_{\pm2.35}$ | $55.42_{\pm15.00}$ | $38.68_{\pm3.83}$ | $55.87_{\pm4.26}$ | $\mathbf{30.74}_{\pm2.46}$ | $62.97_{\pm5.67}$ |
| | MISA | $\underline{53.46}_{\pm6.41}$ | $16.19_{\pm5.26}$ | $\underline{45.41}_{\pm4.62}$ | $\underline{34.42}_{\pm8.60}$ | $\underline{26.92}_{\pm4.88}$ | $35.50_{\pm7.97}$ |
| | FlyPrompt (ours) | $\mathbf{60.16}_{\pm6.88}$ | $\mathbf{7.69}_{\pm2.42}$ | $\mathbf{55.95}_{\pm1.60}$ | $\mathbf{21.27}_{\pm2.89}$ | $26.52_{\pm6.31}$ | $\mathbf{20.69}_{\pm3.58}$ |

Following Moon et al. (2023); Kang et al. (2025), we consider the standard evaluation metrics $A_{auc}$, $A_{last}$, $A_{avg}$ and $F_{last}$ for GCL performance. We denote $R_{i,j}$ as the accuracy recorded right after session $i$ with respect to the data in the session $j$. We then maintain a matrix $\boldsymbol{R}$ whose $j$th column is the history of evaluation after each session with respect to the data in session $j$. Firstly we calculate the final average accuracy $A_{last}$ as:

$$A_{last} = \frac{1}{T}\sum_{i=1}^{T} R_{T,i}, \tag{12}$$

average running accuracy $A_{avg}$:

$$A_{avg} = \frac{1}{T}\sum_{i=1}^{T} R_{i,i}, \tag{13}$$

and the final average forgetting $F_{last}$:

$$F_{last} = \frac{1}{T}\sum_{i=1}^{T}(\max(R_j) - R_{T,i}), \tag{14}$$

where $A_{last}$ and $A_{avg}$ are the higher the better and $F_{last}$ is the lower the better. They are all conventional metrics used in classic CL research.

Moreover, $A_{\text{auc}}$ (higher the better) is the anytime inference metric proposed by Koh et al. (2021) to better evaluate the GCL performance during the learning process. Specifically, the evaluation of GCL accuracy is performed every $b$ batches. Throughout this work, we fix $b = 1000$. Denote the total number of the assessments performed as $S$, then given the history $\boldsymbol{a} \in \mathbb{R}^S$ (where $a_s$ shows the accuracy of the model at time stamp $s$ over the data it has observed so far), we calculate the area under curve of any-time accuracy $A_{\text{auc}}$ as :

$$A_{\text{auc}} = \frac{1}{S} \sum_{s=1}^{S} a_s. \tag{15}$$

In addition to these metrics, we also report the backward transfer (BWT) metric (Lin et al., 2022) to quantify how learning later sessions influences performance on earlier ones. Using the same notation $R_{i,j}$ and $T$ as above, we define

$$\text{BWT} = \frac{1}{T-1} \sum_{i=1}^{T-1} \left( R_{T,i} - R_{i,i} \right). \tag{16}$$

A positive BWT indicates that subsequent learning improves performance on previous sessions (positive backward transfer), while a negative value implies net forgetting on earlier sessions. BWT results of GCL methods are presented in Tab. 12.

### F.3 COMPARISON WITH PROMINENT OFFLINE PTM-BASED METHODS

Table 12: Extended Comparison of performance, backward transfer, and computational cost across PTM-based CL methods. All performance results are reported as an average of five parallel runs ($\pm$ standard deviation) with different random seeds, over the CIFAR-100 dataset and Sup-21K backbone. Parameter counts are measured in millions. Time cost is reported in seconds per batch.

| Method | $A_{\text{auc}}(\%, \uparrow)$ | BWT $(\%, \uparrow)$ | Total Param. | Trainable Param. | Training Time | Inference Time |
|---|---|---|---|---|---|---|
| L2P | $76.23_{\pm 2.73}$ | $0.10_{\pm 2.65}$ | 86.01 | 0.22 | 5.57 | 0.95 |
| DualPrompt | $76.04_{\pm 3.32}$ | $-2.93_{\pm 2.42}$ | 86.35 | 0.55 | 4.78 | 0.90 |
| CODA-P | $79.13_{\pm 3.06}$ | $-0.83_{\pm 2.17}$ | 86.72 | 0.92 | 4.75 | 0.94 |
| MVP | $67.74_{\pm 4.96}$ | $-18.09_{\pm 3.24}$ | 86.12 | 0.32 | 5.35 | 1.27 |
| MISA | $\underline{80.35}_{\pm 2.39}$ | $-1.76_{\pm 2.28}$ | 86.37 | 0.58 | 4.78 | 0.90 |
| S-Prompt++ | $80.21_{\pm 2.55}$ | $0.81_{\pm 1.86}$ | 86.26 | 0.46 | 6.03 | 1.18 |
| HiDe-Prompt | $77.10_{\pm 3.81}$ | $\underline{3.35}_{\pm 2.71}$ | 86.81 | 0.94 | 6.13 | 1.27 |
| HiDe-LoRA | $80.07_{\pm 2.41}$ | $0.36_{\pm 0.94}$ | 87.39 | 1.51 | 6.80 | 1.17 |
| HiDe-Adapter | $79.52_{\pm 2.81}$ | $-2.05_{\pm 1.95}$ | 87.41 | 1.53 | 6.68 | 1.04 |
| NoRGa | $78.89_{\pm 3.33}$ | $2.72_{\pm 1.98}$ | 86.81 | 0.94 | 6.69 | 1.05 |
| SD-LoRA | $79.26_{\pm 2.21}$ | $-6.66_{\pm 3.22}$ | 87.72 | 1.92 | 7.24 | 0.82 |
| FlyPrompt (ours) | $\mathbf{83.24}_{\pm 2.23}$ | $\mathbf{4.35}_{\pm 1.19}$ | 87.08 | 0.46 | 4.96 | 0.92 |

Table 13: Comparison between RanPAC variants and FlyPrompt over three GCL benchmarks. All results are reported as an average of five parallel runs ($\pm$ standard deviation) with different seeds.

| Method | CIFAR-100 | | ImageNet-R | | CUB-200 | |
|---|---|---|---|---|---|---|
| | $A_{\text{auc}}(\%, \uparrow)$ | $A_{\text{last}}(\%, \uparrow)$ | $A_{\text{auc}}(\%, \uparrow)$ | $A_{\text{last}}(\%, \uparrow)$ | $A_{\text{auc}}(\%, \uparrow)$ | $A_{\text{last}}(\%, \uparrow)$ |
| RanPAC$^\dagger$ | $69.91_{\pm 3.88}$ | $79.92_{\pm 0.07}$ | $47.14_{\pm 2.18}$ | $50.75_{\pm 2.15}$ | $60.18_{\pm 5.52}$ | $66.21_{\pm 6.15}$ |
| RanPAC$^\ddagger$ | $57.35_{\pm 8.23}$ | $77.65_{\pm 0.21}$ | $36.90_{\pm 4.17}$ | $44.39_{\pm 0.11}$ | $64.52_{\pm 8.23}$ | $71.65_{\pm 0.17}$ |
| RanPAC$^*$ | $\underline{77.88}_{\pm 4.28}$ | $\underline{86.52}_{\pm 1.15}$ | $\underline{53.18}_{\pm 2.22}$ | $\underline{54.71}_{\pm 2.48}$ | $\underline{69.64}_{\pm 3.89}$ | $\underline{72.30}_{\pm 1.09}$ |
| FlyPrompt (ours) | $\mathbf{83.24}_{\pm 2.23}$ | $\mathbf{86.76}_{\pm 0.73}$ | $\mathbf{56.58}_{\pm 1.47}$ | $\mathbf{55.27}_{\pm 0.91}$ | $\mathbf{70.64}_{\pm 2.85}$ | $\mathbf{73.40}_{\pm 1.88}$ |

### F.4 CKA ANALYSIS OF EXPERTS' REPRESENTATION SIMILARITY

We use centered kernel alignment (CKA) (Kornblith et al., 2019) to quantify the similarity between expert-specific representations while factoring out the shared contribution of the frozen PTM backbone. For a given method and dataset, we first fix the backbone $f_{\boldsymbol{\theta}}$ and, for each expert $E_t$, apply its prompt (or expert-specific encoder parameters) to obtain a matrix of CLS features $\boldsymbol{Z}_t \in \mathbb{R}^{n \times d}$

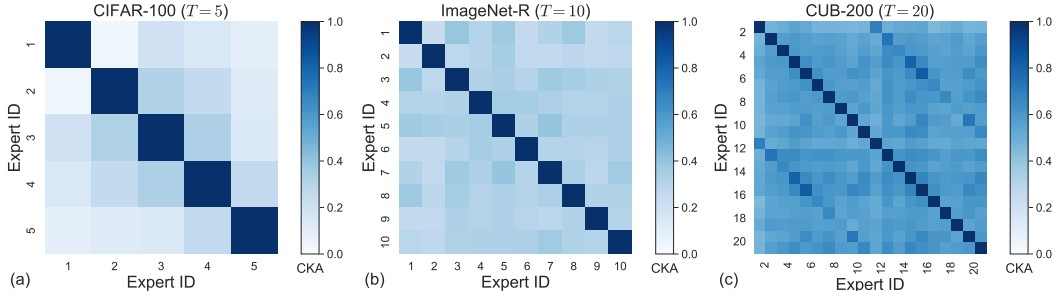

Figure 6: CKA similarity of feature representations between experts of MISA on three datasets.

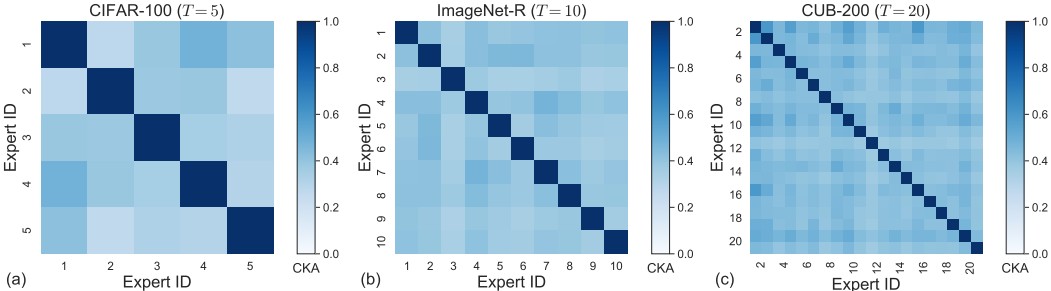

Figure 7: CKA similarity of feature representations between FlyPrompt experts on three datasets.

over a common set of $n$ samples. Using the same backbone together with the expert-shared parameters (e.g., the global prompt in DualPrompt), we also compute a common representation matrix $\boldsymbol{Z}^{\text{com}} \in \mathbb{R}^{n \times d}$. We then define the residual features of expert $E_t$ as $\widetilde{\boldsymbol{Z}}_t = \boldsymbol{Z}_t - \boldsymbol{Z}^{\text{com}}$, which remove the largely stable PTM-driven component and highlight the expert-specific modulation that emerges during online GCL. Based on these residual features, we measure the (linear) CKA between experts $E_t$ and $E_{t'}$ as

$$\text{CKA}(\widetilde{\boldsymbol{Z}}_t, \widetilde{\boldsymbol{Z}}_{t'}) = \frac{\|\widetilde{\boldsymbol{Z}}_t^\top \widetilde{\boldsymbol{Z}}_{t'}\|_{\text{F}}^2}{\|\widetilde{\boldsymbol{Z}}_t^\top \widetilde{\boldsymbol{Z}}_t\|_{\text{F}} \|\widetilde{\boldsymbol{Z}}_{t'}^\top \widetilde{\boldsymbol{Z}}_{t'}\|_{\text{F}}}. \tag{17}$$

This similarity measure is invariant to isotropic rescaling and orthogonal transformations of the features, and is well-suited for comparing representations across experts. We report the pairwise CKA scores between all experts as a heatmap: diagonal entries capture self-similarity, whereas off-diagonal values reveal the degree of specialization or redundancy among experts after removing the common PTM-induced component.

## F.5 EXTRA HYPERPARAMETER SENSITIVITY TEST RESULTS

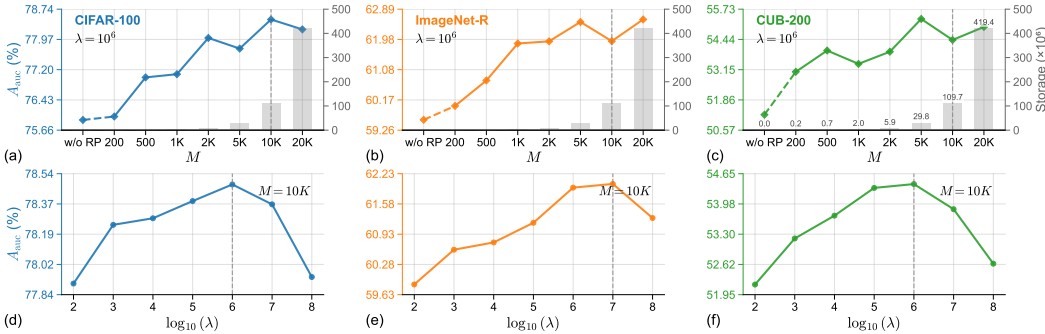

Figure 8: Analysis of hyperparameters in REAR [**Backbone: Sup-21K/1K**]. (a-c) Different random projection dimension $M$ with fixed $\lambda = 10^6$: we report $A_{\text{auc}}$ and extra storage cost (bar) given $M$. (d-f) Different regularization parameter $\lambda$ with fixed $M = 10^4$.

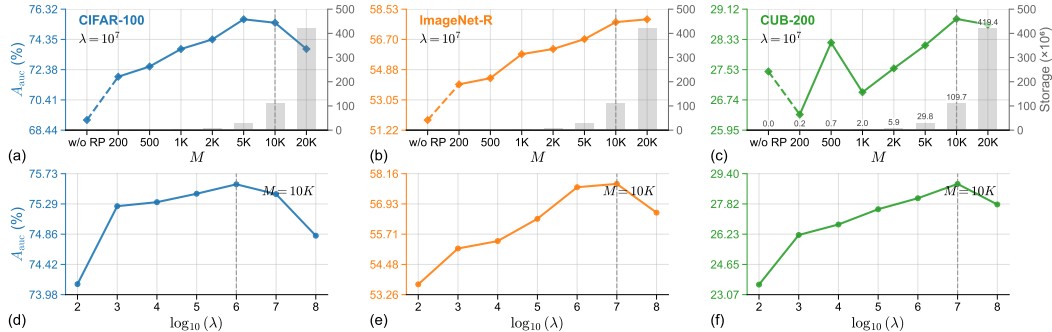

Figure 9: Analysis of hyperparameters in REAR **[Backbone: iBOT-21K]**. (a-c) Different random projection dimension $M$ with fixed $\lambda = 10^7$: we report $A_{\text{auc}}$ and extra storage cost (bar) given $M$. (d-f) Different regularization parameter $\lambda$ with fixed $M = 10^4$.

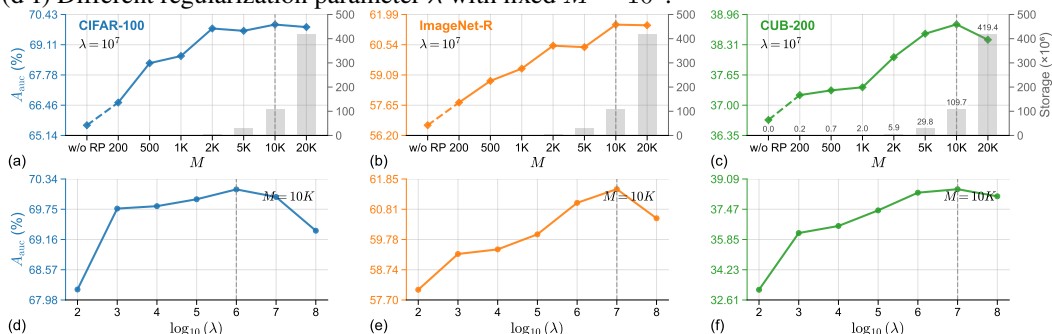

Figure 10: Analysis of hyperparameters in REAR **[Backbone: iBOT-1K]**. (a-c) Different random projection dimension $M$ with fixed $\lambda = 10^7$: we report $A_{\text{auc}}$ and extra storage cost (bar) given $M$. (d-f) Different regularization parameter $\lambda$ with fixed $M = 10^4$.

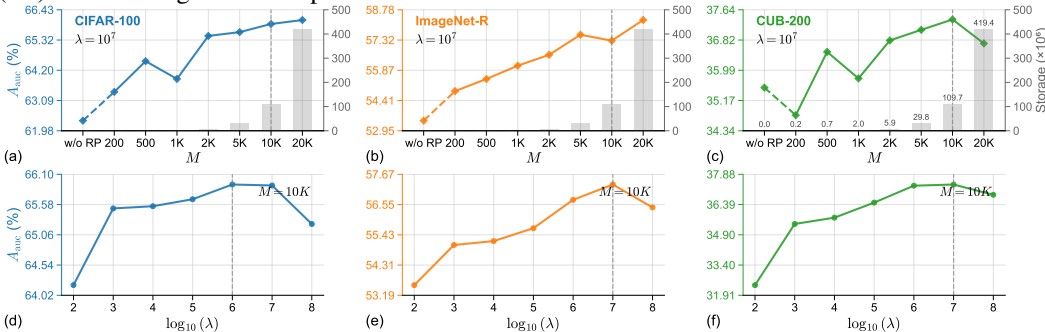

Figure 11: Analysis of hyperparameters in REAR **[Backbone: DINO-1K]**. (a-c) Different random projection dimension $M$ with fixed $\lambda = 10^7$: we report $A_{\text{auc}}$ and extra storage cost (bar) given $M$. (d-f) Different regularization parameter $\lambda$ with fixed $M = 10^4$.

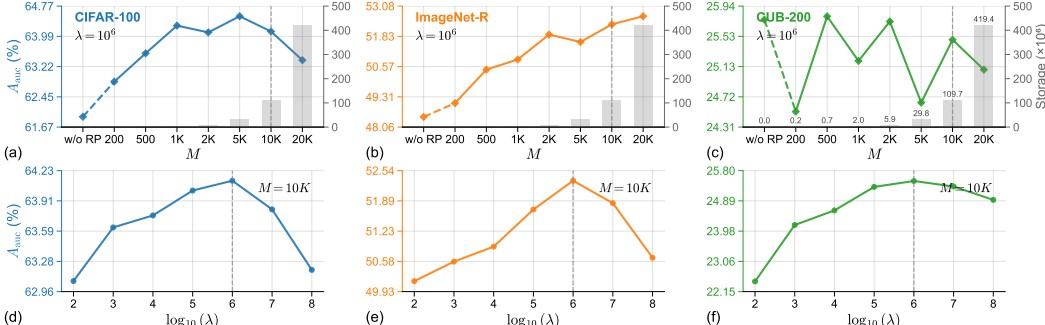

Figure 12: Analysis of hyperparameters in REAR **[Backbone: MoCo v3-1K]**. (a-c) Different random projection dimension $M$ with fixed $\lambda = 10^6$: we report $A_{\text{auc}}$ and extra storage cost (bar) given $M$. (d-f) Different regularization parameter $\lambda$ with fixed $M = 10^4$.

## F.6 ADDITIONAL ABLATION STUDY

Table 14: Effect of REAR and TE$^2$ on various PTM-based CL methods over three GCL benchmarks. All results are reported as an average of five runs ($\pm$ standard deviation) over Sup-21K backbone.

| Setup | Method | CIFAR-100 | | ImageNet-R | | CUB-200 | |
|---|---|---|---|---|---|---|---|
| | | $A_{\text{auc}}(\%,\uparrow)$ | $A_{\text{last}}(\%,\uparrow)$ | $A_{\text{auc}}(\%,\uparrow)$ | $A_{\text{last}}(\%,\uparrow)$ | $A_{\text{auc}}(\%,\uparrow)$ | $A_{\text{last}}(\%,\uparrow)$ |
| Baseline | DualPrompt | $76.04_{\pm3.32}$ | $76.62_{\pm0.74}$ | $46.13_{\pm1.94}$ | $40.80_{\pm1.04}$ | $65.03_{\pm2.24}$ | $62.43_{\pm1.78}$ |
| | MVP | $67.74_{\pm4.96}$ | $63.22_{\pm0.69}$ | $39.50_{\pm1.41}$ | $32.63_{\pm3.95}$ | $54.69_{\pm3.14}$ | $50.07_{\pm3.86}$ |
| | MISA | $80.35_{\pm2.39}$ | $80.75_{\pm1.24}$ | $51.52_{\pm2.09}$ | $45.08_{\pm1.43}$ | $65.40_{\pm3.01}$ | $60.20_{\pm1.82}$ |
| | S-Prompt++ | $80.21_{\pm2.55}$ | $83.48_{\pm1.20}$ | $52.14_{\pm1.65}$ | $49.13_{\pm1.60}$ | $66.61_{\pm2.21}$ | $64.73_{\pm2.25}$ |
| | HiDe-Prompt | $77.10_{\pm3.81}$ | $81.77_{\pm2.00}$ | $53.77_{\pm1.09}$ | $49.87_{\pm3.01}$ | $67.05_{\pm2.37}$ | $67.12_{\pm0.50}$ |
| | NoRGa | $78.89_{\pm3.33}$ | $83.03_{\pm1.20}$ | $54.12_{\pm1.37}$ | $50.09_{\pm3.66}$ | $67.16_{\pm2.44}$ | $67.06_{\pm0.58}$ |
| w/ REAR | DualPrompt | $80.63_{\pm2.25}$ | $83.65_{\pm1.25}$ | $53.16_{\pm1.26}$ | $51.11_{\pm0.91}$ | $65.96_{\pm2.50}$ | $63.81_{\pm1.74}$ |
| | MVP | $67.44_{\pm4.89}$ | $62.33_{\pm1.62}$ | $38.87_{\pm1.27}$ | $31.59_{\pm4.19}$ | $53.65_{\pm3.17}$ | $48.25_{\pm3.38}$ |
| | MISA | $82.03_{\pm1.97}$ | $83.82_{\pm1.04}$ | $\underline{57.30}_{\pm1.12}$ | $54.02_{\pm0.66}$ | $68.04_{\pm2.34}$ | $65.55_{\pm2.69}$ |
| | S-Prompt++ | $81.43_{\pm2.45}$ | $83.93_{\pm0.84}$ | $54.74_{\pm1.53}$ | $52.30_{\pm1.05}$ | $66.80_{\pm2.48}$ | $64.72_{\pm1.97}$ |
| | HiDe-Prompt | $78.41_{\pm2.64}$ | $83.46_{\pm1.14}$ | $53.61_{\pm1.16}$ | $49.26_{\pm2.56}$ | $67.05_{\pm2.36}$ | $66.99_{\pm1.01}$ |
| | NoRGa | $79.37_{\pm2.71}$ | $83.79_{\pm1.28}$ | $54.78_{\pm1.05}$ | $50.24_{\pm3.39}$ | $67.26_{\pm2.50}$ | $67.20_{\pm0.85}$ |
| w/ TE$^2$ | DualPrompt | $76.83_{\pm3.44}$ | $78.00_{\pm0.61}$ | $47.11_{\pm2.19}$ | $42.15_{\pm0.81}$ | $66.47_{\pm2.72}$ | $65.42_{\pm1.55}$ |
| | MVP | $68.91_{\pm4.86}$ | $64.28_{\pm1.00}$ | $42.06_{\pm1.11}$ | $35.94_{\pm0.92}$ | $56.98_{\pm2.79}$ | $54.14_{\pm2.89}$ |
| | MISA | $81.65_{\pm2.24}$ | $82.80_{\pm1.06}$ | $54.05_{\pm1.70}$ | $48.46_{\pm1.15}$ | $69.30_{\pm2.43}$ | $67.29_{\pm2.44}$ |
| | S-Prompt++ | $81.93_{\pm2.21}$ | $83.98_{\pm0.65}$ | $55.37_{\pm1.64}$ | $52.91_{\pm1.53}$ | $67.97_{\pm2.51}$ | $67.68_{\pm1.49}$ |
| | HiDe-Prompt | $77.46_{\pm3.56}$ | $82.09_{\pm1.92}$ | $54.83_{\pm1.08}$ | $50.26_{\pm2.78}$ | $67.77_{\pm2.60}$ | $69.64_{\pm0.76}$ |
| | NoRGa | $79.16_{\pm3.28}$ | $83.01_{\pm1.45}$ | $54.08_{\pm1.58}$ | $51.81_{\pm3.51}$ | $67.97_{\pm2.70}$ | $69.58_{\pm0.90}$ |
| w/ both | DualPrompt | $82.33_{\pm2.17}$ | $86.14_{\pm0.90}$ | $54.72_{\pm1.29}$ | $54.00_{\pm0.76}$ | $69.65_{\pm2.93}$ | $72.75_{\pm2.03}$ |
| | MVP | $68.93_{\pm4.60}$ | $64.52_{\pm1.45}$ | $41.59_{\pm1.34}$ | $35.45_{\pm1.44}$ | $55.65_{\pm2.79}$ | $51.84_{\pm2.44}$ |
| | MISA | $\mathbf{83.60}_{\pm2.08}$ | $86.66_{\pm0.55}$ | $\mathbf{59.12}_{\pm1.02}$ | $\mathbf{56.62}_{\pm0.82}$ | $\mathbf{72.38}_{\pm2.98}$ | $\mathbf{74.68}_{\pm2.11}$ |
| | S-Prompt++ | $83.11_{\pm2.30}$ | $\underline{86.67}_{\pm0.45}$ | $56.57_{\pm1.48}$ | $55.24_{\pm1.24}$ | $\underline{70.65}_{\pm2.84}$ | $\underline{73.42}_{\pm1.88}$ |
| | HiDe-Prompt | $78.60_{\pm2.53}$ | $83.14_{\pm1.12}$ | $54.79_{\pm1.13}$ | $51.30_{\pm2.81}$ | $68.27_{\pm2.57}$ | $69.61_{\pm0.84}$ |
| | NoRGa | $79.37_{\pm2.74}$ | $83.79_{\pm0.96}$ | $55.75_{\pm1.31}$ | $52.50_{\pm3.33}$ | $68.32_{\pm2.64}$ | $70.03_{\pm0.67}$ |
| | FlyPrompt (ours) | $\underline{83.24}_{\pm2.23}$ | $\mathbf{86.76}_{\pm0.73}$ | $56.58_{\pm1.47}$ | $\underline{55.27}_{\pm0.91}$ | $70.64_{\pm2.85}$ | $73.40_{\pm1.88}$ |

## F.7 DIFFERENT LOGIT MASK STRATEGIES FOR GCL MASKS

Table 15: Performance of different logit mask strategies for GCL methods. All results are reported as an average of five parallel runs ($\pm$ standard deviation) with different random seeds, over Sup-21K.

| Mask Type | Method | CIFAR-100 | | ImageNet-R | | CUB-200 | |
|---|---|---|---|---|---|---|---|
| | | $A_{\text{auc}}(\%,\uparrow)$ | $A_{\text{last}}(\%,\uparrow)$ | $A_{\text{auc}}(\%,\uparrow)$ | $A_{\text{last}}(\%,\uparrow)$ | $A_{\text{auc}}(\%,\uparrow)$ | $A_{\text{last}}(\%,\uparrow)$ |
| No Mask | L2P | $62.74_{\pm4.39}$ | $56.08_{\pm2.10}$ | $34.58_{\pm1.23}$ | $26.18_{\pm4.69}$ | $54.83_{\pm2.65}$ | $47.94_{\pm4.64}$ |
| | DualPrompt | $66.68_{\pm5.25}$ | $61.98_{\pm4.09}$ | $41.62_{\pm1.43}$ | $\underline{36.08}_{\pm1.39}$ | $56.68_{\pm2.57}$ | $50.71_{\pm4.21}$ |
| | CODA-P | $66.15_{\pm5.22}$ | $58.42_{\pm1.39}$ | $40.71_{\pm3.50}$ | $30.56_{\pm5.15}$ | $56.44_{\pm2.84}$ | $49.50_{\pm6.04}$ |
| | MVP | $68.18_{\pm4.85}$ | $63.96_{\pm1.48}$ | $38.79_{\pm1.12}$ | $32.01_{\pm2.80}$ | $54.74_{\pm2.01}$ | $\underline{52.88}_{\pm3.08}$ |
| | MISA | $\underline{69.85}_{\pm3.73}$ | $\underline{65.13}_{\pm1.70}$ | $\underline{45.15}_{\pm1.75}$ | $35.91_{\pm3.48}$ | $\underline{57.87}_{\pm2.49}$ | $52.15_{\pm4.27}$ |
| | FlyPrompt (ours) | $\mathbf{78.73}_{\pm3.55}$ | $\mathbf{83.62}_{\pm0.50}$ | $\mathbf{51.39}_{\pm1.80}$ | $\mathbf{48.72}_{\pm1.06}$ | $\mathbf{69.22}_{\pm3.04}$ | $\mathbf{73.07}_{\pm2.34}$ |
| Random Mask | L2P | $62.46_{\pm4.54}$ | $54.67_{\pm1.39}$ | $33.10_{\pm1.34}$ | $24.73_{\pm4.52}$ | $52.92_{\pm2.51}$ | $45.05_{\pm4.89}$ |
| | DualPrompt | $65.48_{\pm4.45}$ | $59.66_{\pm1.70}$ | $36.94_{\pm1.77}$ | $29.22_{\pm1.66}$ | $55.07_{\pm2.21}$ | $47.83_{\pm5.00}$ |
| | CODA-P | $65.58_{\pm5.23}$ | $57.05_{\pm1.97}$ | $39.23_{\pm2.80}$ | $28.91_{\pm5.22}$ | $55.64_{\pm2.27}$ | $48.00_{\pm5.50}$ |
| | MVP | $67.75_{\pm4.95}$ | $\underline{63.21}_{\pm0.78}$ | $39.45_{\pm1.42}$ | $\underline{32.70}_{\pm3.96}$ | $54.72_{\pm3.14}$ | $\underline{50.01}_{\pm3.81}$ |
| | MISA | $\underline{68.23}_{\pm3.82}$ | $61.55_{\pm2.02}$ | $\underline{40.67}_{\pm1.93}$ | $29.48_{\pm4.32}$ | $\underline{56.06}_{\pm2.19}$ | $48.12_{\pm4.86}$ |
| | FlyPrompt (ours) | $\mathbf{78.32}_{\pm3.48}$ | $\mathbf{81.88}_{\pm0.88}$ | $\mathbf{51.67}_{\pm1.30}$ | $\mathbf{47.26}_{\pm1.25}$ | $\mathbf{68.63}_{\pm2.82}$ | $\mathbf{69.52}_{\pm4.03}$ |
| Seen-Class Mask | L2P | $62.22_{\pm4.39}$ | $53.36_{\pm2.03}$ | $33.80_{\pm1.22}$ | $25.20_{\pm4.65}$ | $53.10_{\pm2.60}$ | $45.55_{\pm4.96}$ |
| | DualPrompt | $65.29_{\pm4.62}$ | $57.74_{\pm2.53}$ | $37.31_{\pm1.80}$ | $29.72_{\pm1.49}$ | $55.25_{\pm2.43}$ | $47.67_{\pm4.65}$ |
| | CODA-P | $65.63_{\pm5.40}$ | $56.75_{\pm1.51}$ | $40.13_{\pm2.46}$ | $29.35_{\pm5.01}$ | $55.80_{\pm2.58}$ | $47.71_{\pm5.55}$ |
| | MVP | $67.72_{\pm4.87}$ | $\underline{62.99}_{\pm0.95}$ | $39.57_{\pm1.44}$ | $\underline{32.72}_{\pm4.00}$ | $54.72_{\pm3.14}$ | $\underline{50.14}_{\pm3.80}$ |
| | MISA | $\underline{68.34}_{\pm3.90}$ | $61.44_{\pm2.54}$ | $\underline{41.17}_{\pm1.85}$ | $29.97_{\pm4.14}$ | $\underline{56.44}_{\pm2.42}$ | $48.58_{\pm4.74}$ |
| | FlyPrompt (ours) | $\mathbf{78.75}_{\pm3.52}$ | $\mathbf{82.87}_{\pm0.82}$ | $\mathbf{52.39}_{\pm1.46}$ | $\mathbf{48.02}_{\pm1.00}$ | $\mathbf{69.28}_{\pm2.95}$ | $\mathbf{70.91}_{\pm2.83}$ |
| Batch Seen-Class Mask | L2P | $76.23_{\pm2.73}$ | $79.11_{\pm1.43}$ | $44.40_{\pm1.03}$ | $42.03_{\pm1.72}$ | $64.30_{\pm2.18}$ | $61.42_{\pm2.13}$ |
| | DualPrompt | $76.04_{\pm3.32}$ | $76.62_{\pm0.74}$ | $46.13_{\pm1.94}$ | $40.80_{\pm1.04}$ | $65.03_{\pm2.24}$ | $62.43_{\pm1.78}$ |
| | CODA-P | $79.13_{\pm3.06}$ | $\underline{80.91}_{\pm0.70}$ | $\underline{51.87}_{\pm2.81}$ | $\underline{48.09}_{\pm2.75}$ | $\underline{66.01}_{\pm2.20}$ | $\underline{62.90}_{\pm2.46}$ |
| | MVP | $67.74_{\pm4.96}$ | $63.22_{\pm0.69}$ | $39.50_{\pm1.41}$ | $32.63_{\pm3.95}$ | $54.69_{\pm3.14}$ | $50.07_{\pm3.86}$ |
| | MISA | $\underline{80.35}_{\pm2.39}$ | $80.75_{\pm1.24}$ | $51.52_{\pm2.09}$ | $45.08_{\pm1.43}$ | $65.40_{\pm3.01}$ | $60.20_{\pm1.82}$ |
| | FlyPrompt (ours) | $\mathbf{83.24}_{\pm2.23}$ | $\mathbf{86.76}_{\pm0.73}$ | $\mathbf{56.58}_{\pm1.47}$ | $\mathbf{55.27}_{\pm0.91}$ | $\mathbf{70.64}_{\pm2.85}$ | $\mathbf{73.40}_{\pm1.88}$ |

(1) **No Mask**, standard softmax over all output classes. (2) **Random Mask**, for each sample $(\boldsymbol{x}, y)$, set $m_y = 0$ and assign $m_c = 0$ or $-\infty$ randomly with 0.5 probability for each previously seen class $c \neq y$. (3) **Seen-Class Mask**, setting $m_c = 0$ for all previously seen classes, $-\infty$ otherwise. (4) **Batch Seen-Class Mask** (used), setting $m_c = 0$ only for the classes present in the current batch $\boldsymbol{y}$.

### F.8 COMPLETE RESULTS OF DIFFERENT EMA DECAY RATES FOR TEMPORAL ENSEMBLE

Table 16: Performance comparison of different EMA decay rates for TE$^2$ across all PTMs. All results are reported as an average of five parallel runs ($\pm$ standard deviation) with different seeds.

| PTM | EMA Decay Rate | CIFAR-100 | | ImageNet-R | |
|---|---|---|---|---|---|
| | | $A_{auc}(\%,\uparrow)$ | $A_{last}(\%,\uparrow)$ | $A_{auc}(\%,\uparrow)$ | $A_{last}(\%,\uparrow)$ |
| Sup-21K | online | $81.90_{\pm2.20}$ | $84.23_{\pm1.32}$ | $54.91_{\pm1.32}$ | $52.58_{\pm1.36}$ |
| | 0.9 | $82.81_{\pm2.28}$ | $86.36_{\pm0.54}$ | $\underline{56.36}_{\pm1.52}$ | $55.09_{\pm0.89}$ |
| | 0.99 | $82.84_{\pm2.51}$ | $\underline{86.41}_{\pm0.39}$ | $55.94_{\pm1.65}$ | $54.67_{\pm0.89}$ |
| | 0.999 | $81.80_{\pm2.37}$ | $84.39_{\pm0.83}$ | $55.15_{\pm1.39}$ | $53.52_{\pm0.80}$ |
| | 0.9,0.99 | $\mathbf{83.24}_{\pm2.23}$ | $\mathbf{86.76}_{\pm0.73}$ | $\mathbf{56.58}_{\pm1.47}$ | $\underline{55.27}_{\pm0.91}$ |
| | 0.9,0.99,0.999 | $\underline{82.99}_{\pm2.22}$ | $86.24_{\pm0.79}$ | $56.35_{\pm1.72}$ | $\mathbf{55.50}_{\pm0.77}$ |
| Sup-21K/1K | online | $71.28_{\pm2.58}$ | $69.73_{\pm5.78}$ | $53.12_{\pm2.19}$ | $44.69_{\pm3.65}$ |
| | 0.9 | $75.59_{\pm2.93}$ | $77.39_{\pm6.14}$ | $61.56_{\pm1.48}$ | $\underline{57.35}_{\pm1.63}$ |
| | 0.99 | $77.96_{\pm2.15}$ | $79.71_{\pm3.67}$ | $60.96_{\pm1.94}$ | $56.32_{\pm1.87}$ |
| | 0.999 | $74.13_{\pm1.64}$ | $74.68_{\pm2.93}$ | $53.96_{\pm1.30}$ | $46.55_{\pm1.17}$ |
| | 0.9,0.99 | $\mathbf{78.48}_{\pm1.31}$ | $\underline{80.39}_{\pm3.54}$ | $\underline{62.01}_{\pm2.32}$ | $56.55_{\pm3.94}$ |
| | 0.9,0.99,0.999 | $\underline{78.44}_{\pm1.38}$ | $\mathbf{80.55}_{\pm4.12}$ | $\mathbf{62.59}_{\pm2.22}$ | $\mathbf{58.00}_{\pm1.79}$ |
| iBOT-21K | online | $67.01_{\pm3.85}$ | $65.38_{\pm7.24}$ | $45.32_{\pm1.46}$ | $35.45_{\pm4.60}$ |
| | 0.9 | $72.43_{\pm1.56}$ | $75.46_{\pm4.96}$ | $\underline{56.37}_{\pm2.06}$ | $\underline{52.37}_{\pm0.87}$ |
| | 0.99 | $\underline{75.03}_{\pm0.78}$ | $\underline{78.08}_{\pm3.35}$ | $55.93_{\pm2.35}$ | $51.64_{\pm0.53}$ |
| | 0.999 | $68.96_{\pm3.22}$ | $69.30_{\pm2.94}$ | $43.78_{\pm2.21}$ | $34.64_{\pm2.90}$ |
| | 0.9,0.99 | $\mathbf{75.58}_{\pm1.70}$ | $\mathbf{79.36}_{\pm3.47}$ | $\mathbf{57.75}_{\pm2.12}$ | $\mathbf{54.39}_{\pm1.29}$ |
| | 0.9,0.99,0.999 | $74.16_{\pm2.47}$ | $76.61_{\pm2.26}$ | $55.94_{\pm3.12}$ | $52.12_{\pm0.79}$ |
| iBOT-1K | online | $61.38_{\pm2.32}$ | $60.58_{\pm7.91}$ | $50.79_{\pm1.43}$ | $41.92_{\pm1.76}$ |
| | 0.9 | $68.01_{\pm1.18}$ | $71.99_{\pm4.62}$ | $60.52_{\pm1.51}$ | $\underline{56.98}_{\pm1.17}$ |
| | 0.99 | $\mathbf{71.50}_{\pm1.00}$ | $\mathbf{75.20}_{\pm3.10}$ | $\underline{60.66}_{\pm1.56}$ | $56.57_{\pm0.70}$ |
| | 0.999 | $65.88_{\pm3.51}$ | $67.07_{\pm1.95}$ | $50.27_{\pm1.40}$ | $42.71_{\pm2.03}$ |
| | 0.9,0.99 | $\underline{70.14}_{\pm1.76}$ | $\underline{74.84}_{\pm4.26}$ | $\mathbf{61.50}_{\pm1.66}$ | $\mathbf{57.18}_{\pm1.36}$ |
| | 0.9,0.99,0.999 | $67.93_{\pm3.07}$ | $70.69_{\pm3.50}$ | $59.60_{\pm1.88}$ | $55.35_{\pm1.34}$ |
| DINO-1K | online | $58.61_{\pm3.26}$ | $60.76_{\pm6.77}$ | $47.35_{\pm2.24}$ | $41.33_{\pm2.04}$ |
| | 0.9 | $62.95_{\pm4.13}$ | $69.65_{\pm6.54}$ | $\underline{56.83}_{\pm1.47}$ | $\underline{53.68}_{\pm0.83}$ |
| | 0.99 | $\mathbf{66.42}_{\pm2.51}$ | $\mathbf{73.03}_{\pm3.61}$ | $56.67_{\pm1.74}$ | $53.23_{\pm0.77}$ |
| | 0.999 | $60.37_{\pm4.38}$ | $64.06_{\pm2.37}$ | $46.38_{\pm1.51}$ | $39.69_{\pm2.07}$ |
| | 0.9,0.99 | $\underline{65.92}_{\pm2.74}$ | $\underline{72.66}_{\pm4.52}$ | $\mathbf{57.29}_{\pm2.40}$ | $\mathbf{54.72}_{\pm1.89}$ |
| | 0.9,0.99,0.999 | $65.27_{\pm2.98}$ | $70.83_{\pm4.32}$ | $55.66_{\pm1.57}$ | $51.91_{\pm1.73}$ |
| MoCo-1K | online | $57.90_{\pm5.29}$ | $62.20_{\pm10.03}$ | $42.81_{\pm0.83}$ | $35.46_{\pm3.32}$ |
| | 0.9 | $61.96_{\pm6.50}$ | $69.83_{\pm10.05}$ | $51.47_{\pm1.64}$ | $47.88_{\pm1.49}$ |
| | 0.99 | $\mathbf{65.95}_{\pm4.40}$ | $\mathbf{73.28}_{\pm6.96}$ | $50.75_{\pm1.89}$ | $47.42_{\pm1.29}$ |
| | 0.999 | $61.26_{\pm3.27}$ | $66.42_{\pm5.07}$ | $42.33_{\pm1.54}$ | $36.82_{\pm2.14}$ |
| | 0.9,0.99 | $\underline{64.12}_{\pm5.18}$ | $\underline{71.51}_{\pm8.48}$ | $\mathbf{52.32}_{\pm1.50}$ | $\mathbf{49.06}_{\pm1.35}$ |
| | 0.9,0.99,0.999 | $64.03_{\pm4.47}$ | $69.92_{\pm6.13}$ | $\underline{51.64}_{\pm2.59}$ | $\underline{48.69}_{\pm0.68}$ |

### F.9 COMPARISON OF DIFFERENT ROUTING ALGORITHMS ON RANDOM EXPANDED FEATURES

Table 17: Comparison of routing algorithms based on random expanded features in FlyPrompt. $M$: expansion dimension (default 10,000); $T$: number of experts (default 5); $H$: hidden dimension of MLP ($H = 512$ is used); $K$: number of nearest neighbors ($K = 10$ is used). Time cost is reported in seconds per batch on CIFAR-100. All performance results are reported as an average of five parallel runs ($\pm$ standard deviation) with different random seeds over Sup-21K.

| Routing Algorithm | Train Time | Inference Time | Inference Complexity | CIFAR-100 | | ImageNet-R | | CUB-200 | |
|---|---|---|---|---|---|---|---|---|---|
| | | | | $A_{auc}(\%,\uparrow)$ | $A_{last}(\%,\uparrow)$ | $A_{auc}(\%,\uparrow)$ | $A_{last}(\%,\uparrow)$ | $A_{auc}(\%,\uparrow)$ | $A_{last}(\%,\uparrow)$ |
| Prototype Similarity | 5.58 | 0.90 | $O(MT)$ | $80.67_{\pm2.48}$ | $83.80_{\pm1.15}$ | $54.29_{\pm1.72}$ | $52.36_{\pm1.12}$ | $67.00_{\pm2.77}$ | $66.66_{\pm1.56}$ |
| Naïve Bayes | 5.30 | 0.93 | $O(MT)$ | $\underline{82.73}_{\pm2.17}$ | $\underline{85.51}_{\pm0.97}$ | $55.85_{\pm1.83}$ | $\underline{53.84}_{\pm1.40}$ | $\underline{69.08}_{\pm2.91}$ | $\underline{69.63}_{\pm1.18}$ |
| MLP | 7.03 | 1.00 | $O(MH + HT)$ | $81.75_{\pm2.09}$ | $82.76_{\pm1.98}$ | $\underline{56.31}_{\pm1.29}$ | $53.70_{\pm0.96}$ | $68.53_{\pm2.39}$ | $67.92_{\pm1.91}$ |
| K-Means | 6.11 | 1.49 | $O(KMT)$ | $82.22_{\pm2.04}$ | $85.27_{\pm0.83}$ | $54.93_{\pm1.94}$ | $53.08_{\pm1.43}$ | $68.33_{\pm2.71}$ | $68.24_{\pm2.59}$ |
| Ridge Regression (ours) | 4.96 | 0.92 | $O(MT)$ | $\mathbf{83.24}_{\pm2.23}$ | $\mathbf{86.76}_{\pm0.73}$ | $\mathbf{56.58}_{\pm1.47}$ | $\mathbf{55.27}_{\pm0.91}$ | $\mathbf{70.64}_{\pm2.85}$ | $\mathbf{73.40}_{\pm1.88}$ |

(1) **Prototype Similarity**, cosine similarity to each expert's mean feature. (2) **Naïve Bayes**, assuming Gaussian-distributed features per expert. (3) **MLP**, a two-layer MLP router, as in HiDe (Wang et al., 2023a). (4) **K-Means**, clusters each expert's features and routes based on the nearest center. (5) **Ridge Regression (Ours)**, the REAR analytic router trained once over accumulated statistics.

## F.10 COMPLETE RESULTS OF DIFFERENT AGGREGATION METHODS FOR TEMPORAL ENSEMBLE

Table 18: Performance comparison of different aggregation choices for TE$^2$ across all PTMs. All results are reported as an average of five parallel runs ($\pm$ standard deviation) with different random seeds.

| PTM | Ensemble Method | CIFAR-100 | | ImageNet-R | | CUB-200 | |
|---|---|---|---|---|---|---|---|
| | | $A_{\text{auc}}(\%,\uparrow)$ | $A_{\text{last}}(\%,\uparrow)$ | $A_{\text{auc}}(\%,\uparrow)$ | $A_{\text{last}}(\%,\uparrow)$ | $A_{\text{auc}}(\%,\uparrow)$ | $A_{\text{last}}(\%,\uparrow)$ |
| Sup-21K | Mean | $81.34_{\pm1.64}$ | $85.11_{\pm1.03}$ | $52.71_{\pm1.36}$ | $53.24_{\pm1.22}$ | $68.49_{\pm2.57}$ | $\underline{73.95}_{\pm1.90}$ |
| | Max Prob | $82.29_{\pm2.25}$ | $84.95_{\pm1.20}$ | $55.56_{\pm1.38}$ | $53.53_{\pm1.40}$ | $68.00_{\pm2.50}$ | $66.56_{\pm1.60}$ |
| | Min Entropy | $81.92_{\pm2.19}$ | $84.23_{\pm1.32}$ | $55.05_{\pm1.31}$ | $52.88_{\pm1.38}$ | $66.78_{\pm2.53}$ | $64.73_{\pm1.36}$ |
| | SoftMax+Mean | $82.30_{\pm1.82}$ | $85.98_{\pm0.80}$ | $\underline{56.16}_{\pm1.56}$ | $\mathbf{55.53}_{\pm0.89}$ | $\mathbf{70.77}_{\pm3.00}$ | $\mathbf{74.86}_{\pm1.54}$ |
| | SoftMax+Max | $\mathbf{83.24}_{\pm2.23}$ | $\mathbf{86.76}_{\pm0.73}$ | $\mathbf{56.58}_{\pm1.47}$ | $\underline{55.27}_{\pm0.91}$ | $\underline{70.64}_{\pm2.85}$ | $73.40_{\pm1.88}$ |
| | SoftMax+Min | $\underline{83.11}_{\pm2.34}$ | $86.50_{\pm0.64}$ | $55.94_{\pm1.41}$ | $54.24_{\pm1.34}$ | $69.86_{\pm2.80}$ | $71.51_{\pm1.79}$ |
| Sup-21K/1K | Mean | $\mathbf{79.44}_{\pm1.14}$ | $\mathbf{82.82}_{\pm1.63}$ | $60.84_{\pm2.09}$ | $\mathbf{56.97}_{\pm2.72}$ | $56.64_{\pm4.59}$ | $\underline{60.39}_{\pm3.95}$ |
| | Max Prob | $72.58_{\pm2.99}$ | $71.31_{\pm7.52}$ | $54.28_{\pm2.06}$ | $46.44_{\pm3.23}$ | $47.05_{\pm3.41}$ | $44.42_{\pm3.59}$ |
| | Min Entropy | $71.27_{\pm2.58}$ | $69.73_{\pm5.82}$ | $53.05_{\pm2.11}$ | $44.80_{\pm3.42}$ | $45.36_{\pm3.15}$ | $42.82_{\pm4.31}$ |
| | SoftMax+Mean | $77.72_{\pm1.48}$ | $\underline{81.77}_{\pm3.15}$ | $60.35_{\pm1.86}$ | $\underline{56.63}_{\pm2.05}$ | $\mathbf{57.68}_{\pm5.02}$ | $\mathbf{62.60}_{\pm4.05}$ |
| | SoftMax+Max | $\underline{78.48}_{\pm1.31}$ | $80.39_{\pm3.54}$ | $\mathbf{62.01}_{\pm2.32}$ | $56.55_{\pm3.94}$ | $54.42_{\pm4.67}$ | $55.50_{\pm3.55}$ |
| | SoftMax+Min | $78.30_{\pm1.25}$ | $80.07_{\pm3.68}$ | $\underline{60.89}_{\pm2.15}$ | $55.59_{\pm3.37}$ | $54.12_{\pm4.61}$ | $54.75_{\pm3.57}$ |
| iBOT-21K | Mean | $\mathbf{76.59}_{\pm1.33}$ | $\mathbf{81.40}_{\pm2.48}$ | $54.92_{\pm1.86}$ | $51.82_{\pm1.97}$ | $\mathbf{29.31}_{\pm5.17}$ | $\mathbf{37.60}_{\pm4.77}$ |
| | Max Prob | $68.51_{\pm4.08}$ | $67.60_{\pm8.11}$ | $46.74_{\pm1.39}$ | $37.64_{\pm4.20}$ | $24.14_{\pm3.53}$ | $28.53_{\pm3.46}$ |
| | Min Entropy | $67.03_{\pm3.85}$ | $65.44_{\pm7.22}$ | $45.33_{\pm1.36}$ | $35.86_{\pm4.55}$ | $23.44_{\pm3.39}$ | $27.40_{\pm3.60}$ |
| | SoftMax+Mean | $74.69_{\pm1.77}$ | $\underline{80.58}_{\pm2.75}$ | $54.41_{\pm1.95}$ | $\underline{52.58}_{\pm1.10}$ | $\underline{29.09}_{\pm6.33}$ | $36.28_{\pm7.54}$ |
| | SoftMax+Max | $\underline{75.58}_{\pm1.70}$ | $79.36_{\pm3.47}$ | $\mathbf{57.75}_{\pm2.12}$ | $\mathbf{54.39}_{\pm1.29}$ | $28.86_{\pm5.84}$ | $\underline{36.79}_{\pm7.58}$ |
| | SoftMax+Min | $74.87_{\pm1.89}$ | $77.60_{\pm4.71}$ | $\underline{55.98}_{\pm1.90}$ | $52.03_{\pm1.61}$ | $27.47_{\pm5.43}$ | $34.57_{\pm5.12}$ |
| iBOT-1K | Mean | $\mathbf{70.96}_{\pm1.13}$ | $\mathbf{76.38}_{\pm4.03}$ | $58.15_{\pm1.53}$ | $54.24_{\pm1.39}$ | $37.64_{\pm5.03}$ | $\underline{44.46}_{\pm3.01}$ |
| | Max Prob | $62.92_{\pm2.41}$ | $63.09_{\pm8.29}$ | $52.02_{\pm1.32}$ | $44.15_{\pm1.01}$ | $31.53_{\pm3.46}$ | $34.27_{\pm4.92}$ |
| | Min Entropy | $61.36_{\pm2.30}$ | $60.73_{\pm7.76}$ | $50.74_{\pm1.30}$ | $42.27_{\pm1.46}$ | $30.53_{\pm3.62}$ | $32.82_{\pm4.98}$ |
| | SoftMax+Mean | $67.24_{\pm1.92}$ | $72.80_{\pm4.63}$ | $56.14_{\pm1.33}$ | $53.35_{\pm0.95}$ | $\underline{38.08}_{\pm5.57}$ | $44.13_{\pm4.27}$ |
| | SoftMax+Max | $\underline{70.14}_{\pm1.76}$ | $\underline{74.84}_{\pm4.26}$ | $\mathbf{61.50}_{\pm1.66}$ | $\mathbf{57.18}_{\pm1.36}$ | $\mathbf{38.54}_{\pm5.72}$ | $\mathbf{45.00}_{\pm4.19}$ |
| | SoftMax+Min | $69.72_{\pm1.64}$ | $73.76_{\pm5.07}$ | $\underline{59.87}_{\pm1.52}$ | $\underline{55.39}_{\pm0.90}$ | $37.31_{\pm5.53}$ | $41.86_{\pm4.01}$ |
| DINO-1K | Mean | $\mathbf{66.72}_{\pm1.89}$ | $\mathbf{73.94}_{\pm3.66}$ | $54.00_{\pm2.34}$ | $51.60_{\pm2.21}$ | $\mathbf{37.91}_{\pm6.38}$ | $\underline{44.02}_{\pm2.34}$ |
| | Max Prob | $59.76_{\pm3.26}$ | $62.61_{\pm7.14}$ | $48.12_{\pm2.33}$ | $42.17_{\pm2.21}$ | $31.36_{\pm3.40}$ | $34.50_{\pm4.80}$ |
| | Min Entropy | $58.51_{\pm3.22}$ | $60.79_{\pm6.67}$ | $47.14_{\pm2.31}$ | $40.95_{\pm2.13}$ | $30.15_{\pm3.34}$ | $32.83_{\pm4.91}$ |
| | SoftMax+Mean | $64.79_{\pm3.29}$ | $\underline{72.87}_{\pm4.57}$ | $52.98_{\pm1.49}$ | $51.38_{\pm0.95}$ | $37.22_{\pm7.26}$ | $43.74_{\pm5.34}$ |
| | SoftMax+Max | $\underline{65.92}_{\pm2.74}$ | $72.66_{\pm4.52}$ | $\mathbf{57.29}_{\pm2.40}$ | $\mathbf{54.72}_{\pm1.89}$ | $\underline{37.38}_{\pm5.86}$ | $\mathbf{44.66}_{\pm2.35}$ |
| | SoftMax+Min | $65.48_{\pm2.69}$ | $71.88_{\pm5.38}$ | $\underline{55.69}_{\pm2.51}$ | $\underline{52.55}_{\pm1.90}$ | $36.48_{\pm5.73}$ | $41.43_{\pm2.14}$ |
| MoCo-1K | Mean | $\mathbf{64.29}_{\pm6.09}$ | $\mathbf{72.17}_{\pm8.65}$ | $49.69_{\pm1.28}$ | $46.86_{\pm0.90}$ | $\mathbf{27.92}_{\pm5.19}$ | $\mathbf{33.32}_{\pm3.58}$ |
| | Max Prob | $58.70_{\pm5.09}$ | $63.89_{\pm10.06}$ | $44.07_{\pm0.90}$ | $37.06_{\pm3.05}$ | $21.35_{\pm3.06}$ | $24.21_{\pm4.51}$ |
| | Min Entropy | $57.84_{\pm5.27}$ | $62.21_{\pm9.97}$ | $42.79_{\pm0.82}$ | $35.51_{\pm3.34}$ | $20.59_{\pm2.95}$ | $22.75_{\pm4.42}$ |
| | SoftMax+Mean | $62.90_{\pm6.00}$ | $\underline{71.71}_{\pm8.20}$ | $50.56_{\pm1.57}$ | $\underline{47.95}_{\pm0.70}$ | $\underline{26.95}_{\pm5.22}$ | $\underline{31.50}_{\pm3.85}$ |
| | SoftMax+Max | $\underline{64.12}_{\pm5.18}$ | $71.51_{\pm8.48}$ | $\mathbf{52.32}_{\pm1.50}$ | $\mathbf{49.06}_{\pm1.35}$ | $25.49_{\pm4.53}$ | $30.44_{\pm4.50}$ |
| | SoftMax+Min | $63.92_{\pm5.03}$ | $71.29_{\pm8.50}$ | $\underline{51.46}_{\pm1.44}$ | $47.93_{\pm1.27}$ | $25.26_{\pm4.52}$ | $29.33_{\pm3.88}$ |

## F.11 MORE RESULTS ON SCALABILITY AND EFFICIENCY OF FLYPROMPT

Table 19: Parameter counts, storage and computational complexity breakdown of FlyPrompt components. $M$: expansion dimension (default 10,000); $T$: number of experts (default 5); $l$: prompt length (default 20); $d$: embedding dimension (default 768). Results are reported in millions.

| Components | Total Param. | Trainable Param. | Storage | Storage Cost | Computation Cost |
|---|---|---|---|---|---|
| G matrix | 0.00 | 0.00 | 100 | $O(M^2)$ | $O(M^3)$ |
| Q matrix | 0.00 | 0.00 | 0.05 | $O(MT)$ | $O(MT)$ |
| Router Head | 0.05 | 0.00 | 0.05 | $O(MT)$ | $O(MT)$ |
| Prompts | 0.38 | 0.38 | 0.38 | $O(ld)$ | $O(l^2d)$ |
| TE$^2$ heads | 0.77 | 0.08 | 0.77 | $O(dT)$ | $O(dT)$ |

# G  PSEUDO-CODE OF FLYPROMPT

## G.1  PSEUDO-CODE: ONLINE REAR UPDATES AND TE$^2$ AGGREGATION

---

**Algorithm 1** FlyPrompt: online REAR maintenance and TE$^2$ inference

---

1: **Inputs**: sessions $\{\mathcal{D}_t\}_{t=1}^T$, backbone $f_{\boldsymbol{\theta}}$, random matrix $\boldsymbol{R}\in\mathbb{R}^{d\times M}$, ridge $\lambda$, EMA decays $\{\alpha_j\}_{j=1}^n$, online iterations $k$.

2: **Initialize**: $\boldsymbol{G}\leftarrow\boldsymbol{0}_{M\times M}$, $\boldsymbol{Q}\leftarrow\boldsymbol{0}_{M\times T}$, prompt set $\mathcal{P}\leftarrow\emptyset$, online head $(\boldsymbol{W},\boldsymbol{b})$, logit mask $\boldsymbol{m}\leftarrow\boldsymbol{0}$

3:

4: **(1) Online Training Phase**

5: **for** $t=1$ to $T$ **do**

6:     Set expert $E_t$: $\boldsymbol{p}_t\leftarrow\frac{1}{t-1}\sum_{i=1}^{t-1}\boldsymbol{p}_i$ if $t>1$ else random

7:     $\mathcal{P}\leftarrow\mathcal{P}\cup\{\boldsymbol{p}_t\}$

8:     Initialize EMA heads: $(\boldsymbol{W}_t^{(j)},\boldsymbol{b}_t^{(j)})\leftarrow(\boldsymbol{W},\boldsymbol{b})$, $\forall j\in\{1,\ldots,n\}$

9:     **for** each batch $(\boldsymbol{X},\boldsymbol{y})\subset\mathcal{D}_t$ **do**

10:         Set logit mask $\boldsymbol{m}$: for any class $c\in\boldsymbol{y}$, $m_c\leftarrow 0$, and for $c'\notin\boldsymbol{y}$, $m_{c'}\leftarrow-\infty$

11:         **for** 1 to $k$ **do**

12:             Update $(\boldsymbol{W},\boldsymbol{b})$ and $\boldsymbol{p}_t$ by minimizing $\text{CE}(f_{\boldsymbol{\theta}}(\boldsymbol{X};\boldsymbol{p}_t)\boldsymbol{W}^\top+\boldsymbol{b}+\boldsymbol{m},\ \boldsymbol{y})$

13:             EMA for $E_t$: $\boldsymbol{W}_t^{(j)}\leftarrow\alpha_j\boldsymbol{W}_t^{(j)}+(1-\alpha_j)\boldsymbol{W}$, $\boldsymbol{b}_t^{(j)}\leftarrow\alpha_j\boldsymbol{b}_t^{(j)}+(1-\alpha_j)\boldsymbol{b}$, $\forall j$

14:         **end for**

15:         $\boldsymbol{H}\leftarrow f_{\boldsymbol{\theta}}(\boldsymbol{X};\boldsymbol{p}_t)$; $\boldsymbol{\Phi}\leftarrow\sigma(\boldsymbol{H}\boldsymbol{R})$

16:         Update REAR stats: $\boldsymbol{G}\leftarrow\boldsymbol{G}+\boldsymbol{\Phi}^\top\boldsymbol{\Phi}$; $\boldsymbol{Q}\leftarrow\boldsymbol{Q}+\boldsymbol{\Phi}^\top\boldsymbol{C}_t$         (one-hot $\boldsymbol{C}_t$ for expert $E_t$)

17:     **end for**

18: **end for**

19: Update closed-form router offline: $\widehat{\boldsymbol{U}}^\top\leftarrow(\boldsymbol{G}+\lambda\boldsymbol{I})^{-1}\boldsymbol{Q}$     (only once after the training phase)

20:

21: **(2) Inference Phase**

22: **for** $\boldsymbol{x}$ from the test dataset **do**

23:     Get routing score: $\boldsymbol{s}(\boldsymbol{x})\leftarrow\sigma(f_{\boldsymbol{\theta}}(\boldsymbol{x})\boldsymbol{R})\widehat{\boldsymbol{U}}^\top$

24:     Select the expert: $e\leftarrow\arg\max_{t\le T} s_t(\boldsymbol{x})$

25:     Get online head outputs: $\boldsymbol{z}^{(0)}\leftarrow f_{\boldsymbol{\theta}}(\boldsymbol{x};\boldsymbol{p}_e)\boldsymbol{W}^\top+\boldsymbol{b}$

26:     Get EMA heads output: $\boldsymbol{z}^{(j)}\leftarrow f_{\boldsymbol{\theta}}(\boldsymbol{x};\boldsymbol{p}_e)\boldsymbol{W}_e^{(j)\top}+\boldsymbol{b}_e^{(j)}$

27:     Get aggregated ensemble output: $\hat{\boldsymbol{z}}(\boldsymbol{x})\leftarrow\max_{j\in\{0,\ldots,n\}}\text{softmax}(\boldsymbol{z}^{(j)}+\boldsymbol{m})$

28:     Final output: $\hat{y}(\boldsymbol{x})=\arg\max_c \hat{z}_c(\boldsymbol{x})$

29: **end for**

---

### G.1.1  COMPLEXITY

- Memory for REAR: storing $\boldsymbol{G}\in\mathbb{R}^{M\times M}$ and $\boldsymbol{Q}\in\mathbb{R}^{M\times T}$ is $O(M^2)$.
- Time complexity of solving the analytic router: $O(M^3)$ (matrix inverse and multiply).
- Per-sample cost: forming $\boldsymbol{\varphi}(\boldsymbol{x})=\sigma(\boldsymbol{h}^\top\boldsymbol{R})$ costs $O(dM)$ (matrix multiply).

