# OpenReview forum: "FlyPrompt: Brain-Inspired Random-Expanded Routing with Temporal-Ensemble Experts for General Continual Learning"
_ICLR.cc/2026/Conference — ICLR 2026 Poster_

### Official Review · Reviewer_EcKY · 2025-10-28

**Soundness:** 2
**Presentation:** 3
**Contribution:** 1
**Rating:** 2
**Confidence:** 4

**Summary:**

This paper introduces FlyPrompt, an expert-based framework for General Continual Learning (GCL), where each task is associated with a prompt expert. FlyPrompt proposes two key contributions: a novel strategy named REAR (Random Expanded Analytic Router), which leverages random projection to identify suitable experts at inference, and Task-wise Experts with Temporal Ensemble, designed to track distributional drift. The paper is supported by detailed experiments across standard benchmarks and thoughtful ablation analyses.

**Strengths:**

- The decomposition into expert routing and expert competence is empirically well-supported.
- The results are consistent across a broad range of pre-trained models (Sup-21K, iBOT, DINO, MoCo), rather than being limited to a single backbone.
- The paper is further strengthened by extensive ablation studies and a comprehensive hyperparameter analysis.
- The writing is clear, and the overall structure is well-organized.

**Weaknesses:**

- The paper does not discuss the number of parameters (different from memory in Appendix F) of the proposed method compared to the baselines, which could be a significant concern.
- In the GCL setting, task boundaries are unknown during both **training** and inference [1]. The methodology described in L216–217, which states that "we associate each task $t$ with a corresponding expert $E_t$," appears to assume that task identities are known during training, thereby allowing the assignment of a new expert. This assumption fundamentally contradicts the definition of GCL. Consequently, the problem being addressed may be closer to Class-Incremental Learning with overlapping classes rather than true task-agnostic GCL.
- If task boundaries are indeed assumed to be known during training, then several recent SOTA methods like HiDe-Prompt [2], NoRGa [3], and SD-LoRA [4] could be straightforwardly implemented. The paper lacks a comparison to these highly relevant methods, which weakens its claims of superiority.
- The REAR component, used for identifying task identity, is a powerful module in its own right and is functionally similar to the full RanPAC method [5]. Comparing FlyPrompt (which includes REAR) to methods that employ much simpler routing strategies may therefore be unfair. A more rigorous comparison would involve incorporating REAR into other methods (e.g., evaluating “HiDe-Prompt + REAR”) to properly ablate its contribution. A similar argument applies to the Task-wise Experts with Temporal Ensemble; this technique could also be combined with other baselines for a fairer assessment.
- In L184-186, the authors motivate their second component by claiming that "even with perfect routing, previous methods still exhibit inferior performance... highlighting... the limited competence of individual experts.". However, the paper does not sufficiently diagnose the source of this inferior performance. It remains unclear whether the issue lies in representation drift within the expert prompts $f_\theta(\cdot, p_t)$ or in catastrophic forgetting within the final classification head $g_\psi$. An experiment measuring the representation drift of each expert (e.g., by analyzing the similarity between representations from correct and incorrect experts) would be necessary to clarify and validate this motivation.
- The “Task Experts as Temporal Ensembles” component is presented as a mechanism to enhance expert competence. However, the core prompt expert is trained using a standard cross-entropy loss (Equation 6). Thus, the observed novelty and performance gains appear to stem from ensembling multiple classification heads rather than from improving the representational quality of the prompt expert itself.


[1] Dark Experience for General Continual Learning: a Strong, Simple Baseline, NeurIPS 2020

[2] Hierarchical Decomposition of Prompt-Based Continual Learning: Rethinking Obscured Sub-optimality, NeurIPS 2023

[3] Mixture of Experts Meets Prompt-Based Continual Learning, NeurIPS 2024

[4] SD-LoRA: Scalable Decoupled Low-Rank Adaptation for Class Incremental Learning, ICLR 2025

[5] RanPAC: Random Projections and Pre-trained Models for Continual Learning, NeurIPS 2023

**Questions:**

Please see the Weaknesses section.

---

> ### Author Response · Authors · 2025-11-20
> **Response to Reviewer EcKY**
>
> Thank you for your insightful comments and constructive suggestions. Below, we provide a point-to-point response and summarize the corresponding revisions.
>
> ### **W1: Comparison of parameter counts**
>
> We would respectfully clarify that we have already shown the parameter counts and time cost of our proposed method compared to other baselines in **Table 5** of our original manuscript, supporting that FlyPrompt's performance gain does not come at the cost of significant additional model size or computation. Also, with more detailed statistics reported in the response to **jMmN W2** and **Yd6v Q1**, it is validated that our method is computationally efficient and lightweight with respect to more prominent CL algorithms.
>
> ### **W2: Task boundary and GCL setup**
>
> As clarified in **CQ2**, **FlyPrompt does not rely on any additional boundary information** compared to prior GCL methods such as MVP and MISA. In the Si-Blurry benchmark, the nominal start of a "task" or "session" does not correspond to strong distribution shift, particularly in settings with a low disjoint class ratio (e.g., in streams with small \( r_D \)).
> Instead, it can also adopt a self-triggered mechanism that initializes a new expert once a certain time window or sample threshold is met, without violating the task-free or blurry-boundary assumptions of GCL.
>
> To support this, the empirical results presented in CQ2 demonstrate that **using a simple fixed sample budget** to trigger expert initialization yields performance comparable to or better than setups aligned with Si-Blurry's session boundaries. This further confirms that FlyPrompt's expert management mechanism does not exploit any privileged task boundary signal.
>
> Finally, regarding the broader question of whether Si-Blurry satisfies GCL requirements, we refer to prior studies [1], which provide formal analyses confirming that **Si-Blurry conforms to GCL's core assumptions**. We have added the above clarification in the revised manuscript (lines 864-915).
>
> [1] Advancing Prompt-Based Methods for Replay-Independent General Continual Learning. ICLR, 2025.
>
> ### **W3: Comparison to relevant methods**
>
> | Method       | CIFAR-100 $A_{\rm{auc}}$ | CIFAR-100 $A_{\rm{last}}$ | ImageNet-R $A_{\rm{auc}}$ | ImageNet-R $A_{\rm{last}}$ | CUB-200 $A_{\rm{auc}}$   | CUB-200 $A_{\rm{last}}$   |
> | ------------ | ------------------------ | ------------------------- | ------------------------- | -------------------------- | ------------------------ | ------------------------- |
> | S-Prompt++   | $80.21_{\pm 2.55}$       | $83.48_{\pm 1.20}$        | $52.14_{\pm 1.65}$        | $49.13_{\pm 1.60}$         | $66.61_{\pm 2.21}$       | $64.73_{\pm 2.25}$        |
> | HiDe-Prompt  | $77.10_{\pm 3.81}$       | $81.77_{\pm 2.00}$        | $53.77_{\pm 1.09}$        | $49.87_{\pm 3.01}$         | $67.05_{\pm 2.37}$       | $67.12_{\pm 0.50}$        |
> | HiDe-LoRA    | $80.07_{\pm 2.41}$       | $82.00_{\pm 1.25}$        | $55.09_{\pm 1.45}$        | $51.29_{\pm 6.29}$         | $67.26_{\pm 1.76}$       | $67.28_{\pm 1.45}$        |
> | HiDe-Adapter | $79.52_{\pm 2.81}$       | $81.41_{\pm 0.95}$        | $53.92_{\pm 1.32}$        | $50.86_{\pm 5.08}$         | $66.09_{\pm 1.41}$       | $64.53_{\pm 1.78}$        |
> | NoRGa        | $78.89_{\pm 3.33}$       | $83.03_{\pm 1.20}$        | $54.12_{\pm 1.37}$        | $50.09_{\pm 3.66}$         | $67.16_{\pm 2.44}$       | $67.06_{\pm 0.58}$        |
> | SD-LoRA      | $79.26_{\pm 2.21}$       | $78.91_{\pm 2.48}$        | $55.51_{\pm 1.30}$        | $51.97_{\pm 3.09}$         | $64.12_{\pm 2.02}$       | $60.57_{\pm 0.77}$        |
> | FlyPrompt    | ${\bf 83.24}_{\pm 2.23}$ | ${\bf 86.76}_{\pm 0.73}$  | ${\bf 56.58}_{\pm 1.47}$  | ${\bf 55.27}_{\pm 0.91}$   | ${\bf 70.64}_{\pm 2.85}$ | ${\bf 73.40}_{\pm 1.88}$  |
>
> > Performance is measured by $A_{\rm{auc}}$ and $A_{\rm{last}}$ (%, mean±std) over 5 random seeded runs.

---

> > ### Author Response · Authors · 2025-11-20
> > **Response to Reviewer EcKY (part 2)**
> >
> > We have carefully reproduced the results of the recommended offline class-incremental learning methods (including HiDe-Prompt, NoRGa, and SD-LoRA) under the GCL benchmark. We have also revised the manuscript to include these comprehensive comparison results (lines 378–388) and implementation details (lines 837–850).
> >
> > However, we respectfully note that both HiDe-Prompt and NoRGa cannot be directly adapted to the GCL setting without substantial modification, as they fundamentally rely on: (1) a one-to-one correspondence between class labels and task identities for task inference; and (2) a stable Gaussian distribution per class to perform classifier alignment at task boundaries.
> > Both are incompatible with GCL, where class labels may reappear across sessions and task boundaries are not well-defined. This distinction was discussed in our original manuscript (original Section 2.2) and further addressed in our response to **jMmN W2**.
> >
> > We adapted these methods by allowing them to (1) activate all candidate prompts associated with the predicted class; and (2) collect data features incrementally during online training (rather than at the end of each task). Even under **these favorable conditions for baselines**, FlyPrompt still achieves more superior performance across all datasets.
> >
> > ### **W4: Fair ablation study**
> >
> > | Method      | Baseline           | w/ REAR                               | w/ TE²                              | w/ both                             |
> > | ----------- | ------------------ | ------------------------------------- | ----------------------------------- | ----------------------------------- |
> > | DualPrompt  | $76.04_{\pm 3.32}$ | $80.63_{\pm 2.25} (\uparrow 4.59 )$   | $76.83_{\pm 3.44} (\uparrow 0.79 )$ | $82.33_{\pm 2.17} (\uparrow 6.29 )$ |
> > | MVP         | $67.74_{\pm 4.96}$ | $67.44_{\pm 4.89} (\downarrow 0.30 )$ | $68.91_{\pm 4.86} (\uparrow 1.17 )$ | $68.93_{\pm 4.60} (\uparrow 1.19 )$ |
> > | MISA        | $80.35_{\pm 2.39}$ | $82.03_{\pm 1.97} (\uparrow 1.68 )$   | $81.65_{\pm 2.24} (\uparrow 1.30 )$ | $83.60_{\pm 2.08} (\uparrow 3.25 )$ |
> > | S-Prompt++  | $80.21_{\pm 2.55}$ | $81.43_{\pm 2.45} (\uparrow 1.21 )$   | $81.93_{\pm 2.21} (\uparrow 1.72 )$ | $83.11_{\pm 2.30} (\uparrow 2.90 )$ |
> > | HiDe-Prompt | $77.10_{\pm 3.81}$ | $78.41_{\pm 2.64} (\uparrow 1.31 )$   | $77.46_{\pm 3.56} (\uparrow 0.36 )$ | $78.60_{\pm 2.53} (\uparrow 1.51 )$ |
> > | NoRGa       | $78.89_{\pm 3.33}$ | $79.37_{\pm 2.71} (\uparrow 0.48 )$   | $79.16_{\pm 3.28} (\uparrow 0.27 )$ | $79.37_{\pm 2.74} (\uparrow 0.48 )$ |
> >
> > > Performance is measured by $A_{\rm{auc}}$ (%, mean±std) over 5 random seeded runs on CIFAR-100. Improvement is measured by the difference in $A_{\rm{auc}}$ between the baseline and the corresponding method equipped with REAR, TE² or both.
> >
> > Thank you for the insightful comment. As suggested, we conducted an ablation study by **integrating our REAR and TE² components into a range of strong baseline methods**, including (1) offline CL methods DualPrompt, S-Prompt++, HiDe-Prompt, and NoRGa; and (2) GCL methods MVP and MISA. The results below demonstrate consistent improvements when either component is added, with the largest gains observed when both are applied together.
> > We have added them in the revised manuscript (lines 446-457, 465-469, 1566-1593).

---

> > > ### Author Response · Authors · 2025-11-20
> > > **Response to Reviewer EcKY (part 3)**
> > >
> > > ### **W5: Analysis of expert representation drift**
> > >
> > > Thank you for the valuable suggestion. To further support our design motivation (i.e., **decomposing GCL into expert routing and expert competence improvement**), we conduct a deeper analysis of expert representation drift and the associated decoding performance degradation.
> > >
> > > We begin by identifying two critical factors for improving expert competence in GCL: (1) a well-adapted representation space, governed by the prompt expert; and (2) a robust decoding mechanism, responsible for maintaining decision boundaries over time.
> > > In the context of GCL with pretrained models, these correspond to **the prompt module and the output head**, respectively. To disentangle these two aspects and assess their respective contributions, we have computed **Centered Kernel Alignment (CKA) scores** between each pair of learned prompt experts, as shown in **Figures 3, 6, and 7** of the revised manuscript. The off-diagonal CKA scores are mostly below 0.4, indicating **low similarity** between expert representations.
> > > This confirms that each prompt expert develops a distinct representation space, justifying our emphasis on accurate expert routing for successful GCL.
> > >
> > > Notably, once a prompt expert finishes training in its assigned session, **its encoder is frozen**, so its internal representation remains stable throughout the stream. However, as shown in original **Fig. 2(c)**, even with an oracle router, performance can still degrade over time. This gap highlights the decoding mismatch that occurs when using a single, continuously updated output head to decode fixed representations from earlier prompts.
> > > This observation directly supports the necessity of our proposed TE² module, which preserves multiple output heads per expert, each with its own temporal scope. By maintaining decision boundaries from earlier time windows, TE² improves decoding robustness and better handles representation drift in the expert pool.
> > >
> > > Together, these analyses strengthen our main claim: decomposing GCL into expert routing and competence improvement is both principled and empirically effective, and both components of FlyPrompt (REAR and TE²) are essential to its superior performance (lines 121-132, 198-206, 1453-1494).
> > >
> > > ### **W6: How does TE² enhance expert competence?**
> > >
> > > As outlined in our response to your **W5**, we emphasize that the competence of an expert in a PET-based GCL model depends not only on the **prompt module** (which governs the representation space) but also on the **output head**, which is responsible for accurately decoding those representations into predictions.
> > >
> > > In FlyPrompt, we propose TE² (a temporal ensemble of EMA heads) to enhance this decoding capability. Each prompt expert is equipped with a bank of output heads, updated using different EMA decay rates. This design enables each head to capture evolving patterns in the representation space over multiple timescales and thus compensate potential representation drift. In this regard, expert competence in FlyPrompt is **not limited to the prompt module alone**, but rather reflects the **combined effectiveness of both its representation and decoding modules**.
> > >
> > > We agree that representation-level improvements (e.g., via better prompt optimization or learning objectives) are important and complementary. However, TE² offers a lightweight, training-free, and plug-in strategy that can be easily integrated into GCL models for better performance (see Tables 4, 14 and 19 in the revised manuscript). We have modified the manuscript accordingly to provide better clarifications (lines 198-206, 1712-1724).

---

> > > > ### Comment · Reviewer_EcKY · 2025-11-20
> > > >
> > > > Thank you for your detailed response. The authors have largely addressed my main concerns in the rebuttal, particularly regarding the assumption about task boundaries in GCL. Please incorporate the above discussion into the final revision. Accordingly, I have raised my score to 6.

---

> > > > > ### Author Response · Authors · 2025-11-20
> > > > > **Reply Rebuttal Comment by Authors**
> > > > >
> > > > > Dear Reviewer EcKY,
> > > > >
> > > > > We sincerely appreciate the time you have devoted to reviewing our manuscript, as well as your insightful comments and constructive suggestions. We deeply value your commitment to enhancing the rigor and clarity of our research,  and we will incorporate your recommendations into the revised manuscript. Thank you once again for your contributions to improving the quality of our research.
> > > > >
> > > > > Sincerely,
> > > > >
> > > > > Authors of FlyPrompt.

---

### Official Review · Reviewer_yU5J · 2025-10-29

**Soundness:** 3
**Presentation:** 3
**Contribution:** 2
**Rating:** 6
**Confidence:** 4

**Summary:**

This paper targets General Continual Learning (GCL), characterized by single-pass, non-stationary data streams without clear task boundaries. Identifying limitations in existing Parameter-Efficient Tuning (PET) methods, it decomposes GCL into expert routing and expert competence improvement. Inspired by the fruit fly's memory system, the paper proposes the FlyPrompt framework with two core components: (1) A Random Expanded Analytic Router (REAR) using fixed random projections and a closed-form solution for gradient-free, rapid input-to-expert (prompt) assignment. (2) Task-wise Experts with Temporal Ensemble (TE$^2$) employing multiple EMA heads with different decay rates within each expert to dynamically refine decision boundaries over time. FlyPrompt achieves strong performance across various GCL benchmarks.

**Strengths:**

[S1] The proposed Random Expanded Analytic Router (REAR) uniquely employs a closed-form solution rather than iterative gradient updates to assign inputs to experts. This is advantageous for the strict online, single-pass constraints of GCL, offering a theoretically grounded and computationally efficient alternative to traditional routing methods.

[S2] The framework effectively tackles the complexity of GCL by decomposing it into two manageable subproblems: routing and competence improvement. The Task-wise Experts with Temporal Ensemble (TE²) addresses the latter by leveraging multi-timescale EMA heads, which significantly enhances expert robustness against non-stationary data streams.

**Weaknesses:**

[W1] Initializing the prompt for a new task by averaging previous prompts may not be beneficial and is likely to show degraded performance when subsequent tasks come from significantly different domains.

[W2] While REAR outperforms gradient-based routers, comparisons against simpler non-learning baselines in the expanded feature space (e.g., k-NN routing) are absent, making it hard to gauge the benefit derived from the analytic ridge regression complexity.

[W3] Experiments primarily use the default Si-Blurry configuration. Performance under more extreme imbalance, higher task overlap, or different types of distribution drift needs further investigation.

[W4] While neuro-inspired, drawing direct equivalences between specific algorithmic components (e.g., EMA heads and KC subtypes) and biological counterparts might be an oversimplification.

**Questions:**

Please refer to the weaknesses.

---

> ### Author Response · Authors · 2025-11-20
> **Response to Reviewer yU5J**
>
> Thank you for your insightful comments. Below, we provide a point-to-point response and summarize the corresponding revisions.
>
> ### **W1: Prompt initialization**
>
> We agree that the effectiveness of our prompt warm-up initialization strategy may vary depending on the specific application scenario and data distribution. However, we would like to emphasize its motivated role under the GCL setting, where **training opportunities per expert are typically limited**.
> In this context, averaging the prompts from previously trained experts provides a meaningful starting point for a newly initialized expert. This strategy helps the new expert converge more quickly toward a reasonable solution, even under **single-pass constraints**.
> Moreover, this design aligns with the **blurry-boundary assumption** in GCL. Since classes may reoccur across tasks, and samples from previously seen classes may appear in new sessions, the new expert can still benefit from accumulated prior knowledge, even when the underlying distribution is shifting.
> We have clarified this rationale in the revised manuscript (lines 297–301).
>
> ### **W2: Comparison with routing baselines**
>
> | Routing Strategies   | Update Time   | Inference Time | Inference Complexity | CIFAR-100                | ImageNet-R               | CUB-200                  |
> | -------------------- | ------------- | -------------- | -------------------- | ------------------------ | ------------------------ | ------------------------ |
> | Prototype Similarity | 5.58          | 0.90           | $O(MT)$              | $80.67_{\pm 2.48}$       | $54.29_{\pm 1.72}$       | $67.00_{\pm 2.77}$       |
> | Naïve Bayes          | 5.30          | 0.93           | $O(MT)$              | $82.73_{\pm 2.17}$       | $55.85_{\pm 1.83}$       | $69.08_{\pm 2.91}$       |
> | MLP                  | 7.03          | 1.00           | $O(MH+HT)$           | $81.75_{\pm 2.09}$       | $56.31_{\pm 1.29}$       | $68.53_{\pm 2.39}$       |
> | K-Means              | 6.11          | 1.49           | $O(KMT)$             | $82.22_{\pm 2.04}$       | $54.93_{\pm 1.94}$       | $68.33_{\pm 2.71}$       |
> | Ridge Regression     | 4.96          | 0.92           | $O(MT)$              | ${\bf 83.24}_{\pm 2.23}$ | ${\bf 56.58}_{\pm 1.47}$ | ${\bf 70.64}_{\pm 2.85}$ |
>
> > $M$: expansion dimension; $T$: number of experts; $H$: hidden dimension of MLP; $K$: number of nearest neighbors ($K=10$ is used). Performance is measured by $A_{\rm{auc}}$ (%, mean±std) over 5 random-seeded runs, and time cost is reported in seconds per batch on CIFAR-100. All experiments are conducted on the same server (Intel Xeon Silver 4316 2.3GHz CPUs with 1 NVIDIA RTX 4090 GPU).
>
> We have conducted the above ablation study to compare the performance of different routing strategies upon random expanded features. Four baselines are included:
>
> (1) **Prototype Similarity**, cosine similarity to each expert's mean feature.
> (2) **Naïve Bayes**, assuming Gaussian-distributed features per expert.
> (3) **MLP**, a two-layer MLP router, as in HiDe-Prompt [1].
> (4) **K-Means**, clusters each expert's features and routes based on nearest center (KNN).
> (5) **Ridge Regression (Ours)**, the REAR analytic router trained once over accumulated statistics.
>
> Among these, REAR consistently achieves the best performance across all datasets, while also maintaining low inference cost and competitive update time. Notably, REAR is
> (1) non-parametric and backprop-free, reducing training overhead; (2) order-invariant to input sequences (as shown in our theoretical analysis); and (3) supported by a derived generalization bound for misrouting probability. These theoretical guarantees, combined with empirical gains, validate our choice of ridge regression for efficient and robust routing in the GCL setting.
>
> We have added them in the revised manuscript (linex 258-260, 1656-1673).
>
> [1] Hierarchical Decomposition of Prompt-Based Continual Learning: Rethinking Obscured Sub-optimality. NeurIPS, 2023.

---

> > ### Author Response · Authors · 2025-11-20
> > **Response to Reviewer yU5J (part 2)**
> >
> > ### **W3: Performance under extreme conditions**
> >
> > Thank you for the insightful comment. As detailed in **Appendix E.1** of our original manuscript, we have already conducted extensive evaluations under a wide range of **GCL configurations**, including transitions from purely blurry ($r_D = 0$) to fully disjoint ($r_D = 1$) task boundaries (original **Table 6**), and varying degrees of intra-task class overlap and distribution shift (original **Table 7**).
> > Across all these settings, **FlyPrompt consistently outperforms other GCL baselines**, demonstrating remakrable robustness and applicability.
> > We have further highlighted this in the revised manuscript (lines 1342–1345).
> >
> > ### **W4: Correspondence between brain mechanisms and algorithmic design**
> >
> > As clarified in **CQ1**, our motivation is not to simulate biological details, but to design effective GCL methods inspired by well-established computational principles from neuroscience.
> > FlyPrompt is guided by two key structural and functional features of the fruit fly's learning system (i.e., the mushroom body): (1) sparse, high-dimensional random expansion, which supports instance-level routing and interference reduction; and (2) parallel multi-timescale plasticity, allowing integration of both recent and remote experience.
> >
> > In our framework, the **random expanded analytic router (REAR)** plays the role of mitigating interference among overlapping inputs by projecting them into a sparse, high-dimensional space and assigning them to modular experts.
> > The **bank of EMA heads (TE²)** with varying decay rates mirrors the multi-timescale plasticity seen in mushroom body compartments, enabling FlyPrompt to adapt dynamically across different phases of distribution shift.
> >
> > This biological analogy is not merely decorative: it directly informs the decomposition of GCL into routing and competence improvement submodules, and constrains the architecture we propose. The algorithmic roles and interactions between components are tightly aligned with the underlying biological mechanisms. Further, our ablation studies show that replacing REAR with alternative routing methods or removing the temporal ensemble leads to notable performance degradation, reinforcing that this bio-inspired design is functionally essential.
> > We have clarified them in the revised manuscript (lines 1566-1593, 1656-1673).

---

> > > ### Comment · Reviewer_yU5J · 2025-11-23
> > >
> > > Thank you for the efforts for the responses.
> > > I have read the authors’ responses to the concerns I raised.
> > > The additional explanations regarding the rationale for each component and the accompanying experimental results appear convincing.
> > > I will maintain my positive score.

---

> > > > ### Author Response · Authors · 2025-11-24
> > > >
> > > > Dear Reviewer yU5J,
> > > >
> > > > We sincerely appreciate the time and effort you dedicated to evaluating our manuscript. Your insightful comments and constructive suggestions have helped refine our work, and we have carefully revised the manuscript to address your feedback. Thanks again for your support.
> > > >
> > > > Sincerely,
> > > >
> > > > Authors of FlyPrompt.

---

### Official Review · Reviewer_Yd6v · 2025-10-31

**Soundness:** 2
**Presentation:** 3
**Contribution:** 2
**Rating:** 6
**Confidence:** 4

**Summary:**

The paper proposes FlyPrompt, a framework for GCL inspired by the neural circuitry of the fruit fly mushroom body. It decomposes GCL into two subproblems: expert routing and expert competence improvement. For expert routing, the authors introduce the REAR, which uses fixed random projections and closed-form updates for feedforward expert selection. For competence improvement, they propose TE2, which integrates knowledge across multiple timescales via EMA heads with different decay rates. The method reports SOTA performance on benchmarks.

**Strengths:**

1. The separation of GCL into routing and competence subproblems offers a structured approach to tackling its challenges.
2. The use of principles from fruit fly olfactory memory introduces a novel interdisciplinary perspective to CL.
3. The paper provides both informal and formal theoretical bounds on routing error and EMA parameter error, enhancing methodological credibility.

**Weaknesses:**

1. It resemble an ad hoc combination of existing techniques. The proposed FlyPrompt framework appears to be largely a composition of well-established components rather than a fundamentally novel algorithm. Specifically, the REAR combines fixed random projection with ridge regression (a paradigm already explored in prior CL works and analytic class-incremental learning). Similarly, the TE2 employs EMA with multiple decay rates, a standard technique in online learning and model stabilization (e.g., SWA, temporal ensembling).
2. While the paper provides a biologically inspired narrative grounded in the fruit fly’s mushroom body, the mapping between neurobiological mechanisms and algorithmic design remains largely metaphorical. The performance gains reported (e.g., +11.23% auc on CIFAR-100) are primarily attributable to the strong supervised pretraining (Sup-21K) and the inherent benefits of random feature expansion, rather than the proposed routing or ensemble mechanisms. The ablation study (Table 2) further reveals that a RanPAC-like baseline already achieves 82.17% auc. This suggests that FlyPrompt’s contribution is incremental engineering rather than a necessary or uniquely effective solution.
3.  While GCL as a research direction is valid, the specific formulation and assumptions in this work appear tailored to a controlled benchmark rather than a pressing real-world problem. The number of tasks $T$ (and thus the number of prompt experts) is assumed known and fixed a priori, which contradicts truly open-world or task-agnostic streaming environments. The evaluation protocol assumes access to task-level metadata during training (e.g., expert identity per session), which may not hold in fully unsupervised or user-generated data streams.

**Questions:**

How about the practical scalability and efficiency? Although Table 5 reports only marginal increases in total parameters and per-batch latency, it omits the auxiliary memory and compute burden of $G$ and $Q$

---

> ### Author Response · Authors · 2025-11-20
> **Response to Reviewer Yd6v**
>
> Thank you for your insightful comments. Below, we provide a point-to-point response and summarize the corresponding revisions.
>
> ### **W1: Novelty of the FlyPrompt algorithm**
>
> As clarified in **CQ1**, we highlight FlyPrompt's novelty from two key aspects:
>
> First, FlyPrompt **uses random projection solely for expert routing**, not for final classification. This allows the router to leverage the stability and sparsity of random expanded features, while keeping each expert's prompt parameters and output head fully trainable under online, non-stationary scenarios. This separation between a fixed routing layer and plastic expert modules enables targeted adaptation without sacrificing efficiency or scalability.
>
> Second, our temporal ensemble mechanism is designed to cope with **representation drift**, rather than to stabilize a final model as in standard EMA or SWA. Each expert maintains a bank of EMA heads with different time scales, and all heads are jointly used during inference. This design can capture complementary information across short-, medium-, and long-term time windows for robust GCL performance.
>
> Together, these components are informed by distinct computational principles in the fruit fly's learning system: (1) Random expansion enables sparse interference-resistant routing [1]. (2) Multi-compartment learning supports temporal generalization via diverse memory traces [2].
> To our knowledge, FlyPrompt is the first to implement both mechanisms for either pretrained models, parameter-efficient tuning (PET), or realistic GCL settings. This integration results in cross-disciplinary insights and algorithmic innovations for future PET-based GCL systems.
>
> [1] A neural algorithm for a fundamental computing problem. *Science*, 2017.
>
> [2] Incorporating neuro-inspired adaptability for continual learning in artificial intelligence. *Nature Machine Intelligence*, 2023.
>
> ### **W2: Biological inspiration and FlyPrompt's effectiveness**
>
> We appreciate the reviewer's comments and would like to offer two clarifications.
>
> First, regarding the **empirical gains**: the +11.23% $A_{\rm auc}$ improvement cited in the review is not obtained over a strong supervised checkpoint (e.g., Sup-21K), but rather over a **weaker self-supervised checkpoint** DINO-1K, where FlyPrompt achieves 65.92% over 54.69%.
> Additionally, the reviewer appears to reference the result of "Prompt Expert + EMA head" (82.17%) as that of a RanPAC-like baseline. However, the corresponding RanPAC$^\dagger$ baseline only achieves 69.91% $A_{\rm auc}$ on CIFAR-100 (original Table 2), significantly lower than FlyPrompt. This result illustrates that **random features alone do not suffice** for effective GCL under single-pass constraints.
>
> Second, regarding the **bio-inspired motivation**: our use of random projection is not intended as a metaphor, but as a computational counterpart to the sparse random expansion observed in the fruit fly's learning system. Neuroscience studies show that such expansion coding supports effective pattern separation and interference reduction [1,2].
> Inspired by this, FlyPrompt's routing is performed with random projections in a **backprop-free manner**, while **each expert module remains plastic**, learning via gradient-based optimization.
>
> In contrast, earlier analytic methods use random features for final classification, which requires feature replay or fixed representations, both incompatible with GCL. Our ablation studies support this distinction: (1) replacing REAR with direct analytic classification leads to degraded performance; and (2) removing the temporal ensemble also results in significant drops.
>
> These findings confirm that FlyPrompt's performance gains stem from its bio-inspired modular routing design and multi-timescale ensemble mechanism, not simply from strong pretraining or random projections. We have clarified this distinction and the experimental results more explicitly in the revised manuscript (lines 254–260, 446-457, 1442-1451, 1566-1593).
>
> [1] A neural algorithm for a fundamental computing problem. *Science*, 2017.
>
> [2] Cellular-resolution population imaging reveals robust sparse coding in the Drosophila mushroom body. *Journal of Neuroscience*, 2011

---

> > ### Author Response · Authors · 2025-11-20
> > **Response to Reviewer Yd6v (part 2)**
> >
> > ### **W3: GCL setup and task information**
> >
> > In addition to our response in **CQ2**, we would like to further clarify the scope and assumptions of our GCL setting. We follow the GCL paradigm introduced by DER and later adopted by MVP and MISA, characterized by: (1) single-pass, non-stationary data stream; (2) no task oracle at test time; and (3) randomized class composition and sample counts per session.
> >
> > In our implementation, expert indices are aligned with the nominal session identities provided by the benchmark. However, these indices do not encode distributional change information. They are used strictly as implementation placeholders, i.e., the same behavior could be reproduced by starting a new expert after a fixed number of samples or when a computational/storage budget is reached. This makes FlyPrompt compatible with task-free, blurry-boundary settings.
> >
> > Moreover, the total number of experts is not a fixed assumption: FlyPrompt allows matrices such as $Q$ and the router head to be dynamically extended (e.g., from $T$ to $T+1$) via zero-padding, without impacting prior computations. This design ensures flexibility and extensibility as the stream evolves.
> >
> > While our experiments focus on supervised, class-annotated data streams to ensure comparability with prior GCL methods, the core components (REAR and TE²) only rely on the current batch of data. They are readily compatible with user-defined segmentations or unsupervised learning signals, making FlyPrompt adaptable to broader continual learning settings in future work.
> > We have clarified these points in the revised manuscript (lines 864–915, 1342-1345).
> >
> > ### **Q1: Scalability and efficiency**
> >
> > | Components  | Total  Param. | Trainable  Param. | Storage     | Storage Cost | Computation Cost |
> > | ----------- | ------------- | ----------------- | ----------- | ------------ | ---------------- |
> > | G matrix    | 0.00          | 0.00              | 100         | $O(M^2)$     | $O(M^3)$         |
> > | Q matrix    | 0.00          | 0.00              | 0.05        | $O(MT)$      | $O(MT)$          |
> > | Router Head | 0.05          | 0.00              | 0.05        | $O(MT)$      | $O(MT)$          |
> > | Prompts     | 0.38          | 0.38              | 0.38        | $O(ld)$      | $O(l^2d)$        |
> > | TE² heads   | 0.77          | 0.08              | 0.77        | $O(dT)$      | $O(dT)$          |
> >
> > > $M$: expansion dimension; $T$: number of experts; $l$: prompt length; $d$: embedding dimension. Parameter counts are measured in millions.
> >
> > We provide the above breakdown to further clarify the efficiency and scalability of FlyPrompt, particularly with respect to the REAR module. While the main paper (original Fig. 4 and Table 5) already presents performance and runtime comparisons, we elaborate here on the cost of maintaining the matrices $G$ and $Q$.
> >
> > During training, we only accumulate $G$ and $Q$ incrementally per batch: $G \in \mathbb{R}^{M \times M}$ scales as $O(M^2)$ in storage, and $Q \in \mathbb{R}^{M \times T}$ scales as $O(MT)$.
> > Critically, the Gram matrix inversion, which has $O(M^3)$ complexity, is performed only once upon evaluation, and is therefore not a bottleneck in GCL.
> > The router head parameters scale with $O(MT)$ but are fixed after setup and do not require training.
> > The TE² heads and prompts remain lightweight and trainable, with complexity comparable to other PET-based methods.
> >
> > Under our default setting ($M=10000$, moderate $T$), the total storage and compute cost of REAR is practically manageable and justified by the significant accuracy gains reported. As shown in original Table 5, the overall per-batch latency increases only slightly compared to baselines, demonstrating minimal runtime overhead.
> > These results confirm that FlyPrompt's design scales well and can be applied to larger backbones or longer data streams without incurring prohibitive cost.
> > We have added these points and results in the revised manuscript (lines 1426–1441, 1712-1724).

---

> > > ### Author Response · Authors · 2025-11-27
> > >
> > > Dear Reviewer Yd6v,
> > >
> > > Thank you very much for your time and constructive feedback on our paper. We have posted a detailed response addressing your concerns, specifically clarifying the distinct novelty of our method compared to prior works, confirming its empirical effectiveness, and explaining our adherence to standard GCL protocols regarding task boundaries.
> > >
> > > We hope these revisions might serve as meaningful justification for a more positive evaluation. Should you have any further questions, please let us know.
> > >
> > > Thank you again for your dedication to the review process.
> > >
> > > Best regards,
> > >
> > > The Authors

---

### Official Review · Reviewer_jMmN · 2025-11-01

**Soundness:** 3
**Presentation:** 3
**Contribution:** 3
**Rating:** 4
**Confidence:** 4

**Summary:**

Existing parameter-efficient tuning methods struggle in General Continual Learning (GCL) because they cannot effectively allocate expert parameters or improve representations in single-pass, boundary-free data streams. Inspired by the fruit fly's brain, this paper proposes FlyPrompt, a framework that decomposes GCL into expert routing and expert competence improvement. FlyPrompt uses a random analytic router to activate experts and a temporal ensemble of output heads to adapt, significantly outperforming state-of-the-art baselines on key benchmarks like CIFAR-100 and ImageNet-R.

**Strengths:**

**Biologically Inspired Foundation**: The framework is grounded in the neurobiological principles of the fruit fly's brain, offering a novel approach to solving complex GCL challenges.

**Addresses Core GCL Problems**: It effectively tackles two fundamental challenges in GCL: "expert routing" (selecting the right parameters) and "expert competence improvement" (adapting to new data) under difficult, realistic constraints (single-pass data, no task boundaries).

**Novel and Efficient Components**: It introduces two key innovations:

 - A randomly expanded analytic router for non-iterative (fast and efficient) expert selection.

 - A temporal ensemble of expert heads to ensure the model robustly adapts to data changes over time.

**Proven Performance**: The method is backed by both strong theoretical analysis and excellent empirical results, demonstrating superior performance and scalability across multiple GCL benchmarks.

**Weaknesses:**

**Notational Clarity**: There appears to be a notational inconsistency in Equations 2 and 3, where the symbols $\Phi$ and $\varphi$ seem to be transposed or confused.

**Comparative Analysis**: The paper would be significantly strengthened by a direct comparison of FlyPrompt against other prominent methods (such as LoRA, Adapters, and MoE). This comparison should explicitly analyze key metrics:

- Parameter efficiency (total and new parameters)

- Computational overhead (training and inference time)

- Backward Transfer (BWT)

**Novelty of Application**: Given that Random Projection is a well-established technique, what is the specific novelty of its application within the FlyPrompt framework? How does its integration into the "randomly expanded analytic router" differ from standard implementations and what unique advantages does this specific application provide for the GCL setting?

**Questions:**

Please refer to the weaknesses.

---

> ### Author Response · Authors · 2025-11-20
> **Response to Reviewer jMmN**
>
> Thank you for your insightful comments. Below, we provide a point-to-point response and summarize the corresponding revisions.
>
> ### **W1: Notational clarity in Eq.(2) and Eq.(3)**
>
> In Eq. (1), we define the random projected feature of a data point $x$ as $\varphi(x) \in \mathbb{R}^M$, where $M$ is the expanded dimension. For a batch of $B$ samples, we denote the set of projected features as $\Phi \in \mathbb{R}^{B \times M}$, whose row vectors are ${\varphi_j^\top}_{j=1}^{B}$. While the notations in Eq. (2) and Eq. (3) are mathematically correct, we agree that they can be improved for clarity. We have revised the manuscript accordingly to make the notation more intuitive and to clarify the role of each symbol (lines 234–236).
>
> ### **W2: Comparative analysis with prominent methods**
>
> | Method          | $A_{\rm auc}$            | BWT                     | Total Param. | Trainable Param. | Training Time | Inference Time |
> | --------------- | ------------------------ | ----------------------- | ------------ | ---------------- | ------------- | -------------- |
> | HiDe-LoRA       | $80.07_{\pm 2.41}$       | $0.36_{\pm 0.94}$       | 87.39        | 1.51             | 6.80s         | 1.17s          |
> | HiDe-Adapter    | $79.52_{\pm 2.81}$       | $-2.05_{\pm 1.95}$      | 87.41        | 1.53             | 6.68s         | 1.04s          |
> | S-Prompt++(MOE) | $80.21_{\pm 2.55}$       | $0.81_{\pm 1.86}$       | 86.26        | 0.46             | 6.03s         | 1.18s          |
> | L2P             | $76.23_{\pm 2.73}$       | $0.10_{\pm 2.65}$       | 86.01        | 0.22             | 5.57s         | 0.95s          |
> | DualPrompt      | $76.04_{\pm 3.32}$       | $-2.93_{\pm 2.42}$      | 86.35        | 0.55             | 4.78s         | 0.90s          |
> | CODA-P          | $79.13_{\pm 3.06}$       | $-0.83_{\pm 2.17}$      | 86.72        | 0.92             | 4.75s         | 0.94s          |
> | MVP             | $67.74_{\pm 4.96}$       | $-18.09_{\pm 3.24}$     | 86.12        | 0.32             | 5.35s         | 1.27s          |
> | MISA            | $80.35_{\pm 2.39}$       | $-1.76_{\pm 2.28}$      | 86.37        | 0.58             | 4.78s         | 0.90s          |
> | HiDe-Prompt     | $77.10_{\pm 3.81}$       | $3.35_{\pm 2.71}$       | 86.81        | 0.94             | 6.13s         | 1.27s          |
> | NoRGa           | $78.89_{\pm 3.33}$       | $2.72_{\pm 1.98}$       | 86.81        | 0.94             | 6.69s         | 1.05s          |
> | SD-LoRA         | $79.26_{\pm 2.21}$       | $-6.66_{\pm 3.22}$      | 87.72        | 1.92             | 7.24s         | 0.82s          |
> | Fly-Prompt      | ${\bf 83.24}_{\pm 2.23}$ | ${\bf 4.35}_{\pm 1.19}$ | 87.08        | 0.46             | 4.96s         | 0.92s          |
>
> > Performance is measured by $A_{\rm{auc}}$ (%, mean±std) over 5 random seeded runs on CIFAR 100. Parameter counts in the table are measured in millions. Time cost is reported in seconds per batch. All experiments are conducted with the same server (Intel Xeon Silver 4316 2.3GHz CPUs with 1 NVIDIA RTX 4090 GPU).
>
> We would respectfully point out that we have already provided results of parameter efficiency (including total and trainable parameter counts) and computational time cost comparison with other GCL methods in Table 5 of our original manuscript. Following your suggestion, we have modified several recent **offline CL methods** to conform to the GCL constraints, including: (1) **S-Prompt++** with MoE structure [1-3]; (2) **HiDe-Prompt** / HiDe-LoRA / HiDe-Adapter that is constructed upon S-Prompt++ [1]; and (3) **NoRGa** that is constructed upon HiDe [3].
>
> Taking HiDe-Prompt as an example, its original version performs two-stage inference: task-identity prediction (TIP) followed by within-task prediction (WTP). TIP uses a separate branch to produce a raw prediction, then uses a class–task map to select the prompt. However, this design is incompatible with GCL, where class–task mappings are ambiguous or unavailable. To adapt HiDe-Prompt to GCL, we allow it to **activate all prompts corresponding to candidate task identities** associated with the predicted class. A prediction is considered correct if any activated prompt returns the correct label, i.e., **an intentionally favorable setting for HiDe-Prompt**.
>
> Despite this, **FlyPrompt outperforms S-Prompt++, all variants of HiDe and its stronger MoE-based successor NoRGa** [3], across both $A_{\text{auc}}$ and BWT. The results confirm that FlyPrompt achieves **state-of-the-art performance** while maintaining **low parameter count and training cost**.
>
> We have added them in the revised manuscript (lines 378-387, 404–406, 837-850, 1426-1441).
>
> [1] Hierarchical Decomposition of Prompt-Based Continual Learning: Rethinking Obscured Sub-optimality. NeurIPS, 2023.
>
> [2] S-Prompts Learning with Pre-Trained Transformers: An Occam's Razor for Domain Incremental Learning. NeurIPS, 2022.
>
> [3] Mixture of Experts Meets Prompt-Based Continual Learning. ICLR, 2025.

---

> > ### Author Response · Authors · 2025-11-20
> > **Response to Reviewer jMmN (part 2)**
> >
> > ### **W3: Novelty of REAR**
> >
> > As clarified in **CQ1**, our REAR module is used specifically for expert routing, whereas prior random projection methods (e.g., RanPAC, ACIL) apply closed-form ridge regression directly for final classification. These methods rely on the assumption that the pretrained model provides **stable and well-distributed representations**, a prerequisite for effective analytic solutions. However, this assumption severely limits adaptability in continual learning, especially in online, task-free settings, where evolving data streams demand strong adaptability.
> > Moreover, in fine-grained scenarios such as CUB-200, pretrained representations often do not align well with downstream task requirements. As a result, methods that depend entirely on fixed representations often require complex auxiliary mechanisms (e.g., BoostCL [1]) to compensate for this misalignment.
> >
> > In contrast, **REAR decouples routing from classification**: It uses analytic random projections solely to determine expert assignment, while allowing each expert's parameters (prompt + head) to remain trainable and adapt to incoming data. To our knowledge, FlyPrompt is the first to use random projection for instance-level expert routing in GCL. This design preserves the efficiency and theoretical grounding of random expanded features while enabling gradient-based adaptation within each expert, avoiding the brittleness caused by representation drift.
> > We have clarified this further in the revised manuscript (lines 254–260).
> >
> > [1] Boosting Multiple Views for Pretrained-Based Continual Learning. ICLR 2025.

---

> > ### Comment · Reviewer_jMmN · 2025-11-21
> >
> > The authors have successfully addressed the main concerns raised in the initial review, specifically regarding the parameters and the training/test time of the proposed model.
> >
> > Please ensure the final manuscript is updated to incorporate a detailed discussion and clarification on these points, reflecting the information provided in the rebuttal. I have raised my score to 6.

---

> > > ### Author Response · Authors · 2025-11-22
> > >
> > > Dear Reviewer jMmN,
> > >
> > > Thank you very much for your thoughtful and constructive feedback. We truly appreciate the time and care you devoted to reviewing our work. We will incorporate the new content and discussion points from the rebuttal period into the revised version to further strengthen the paper.
> > >
> > > Sincerely,
> > >
> > > Authors of FlyPrompt

---

### Official Review · Reviewer_Vwq2 · 2025-11-02

**Soundness:** 4
**Presentation:** 4
**Contribution:** 4
**Rating:** 10
**Confidence:** 4

**Summary:**

the paper introduces a neuro-inspired framework for General Continual Learning that learns online without task labels or replay. It breaks the problem into two parts: routing and expert competence. The Random-Expanded Analytic Router uses random feature projections and closed form ridge updates to select experts efficiently -- no  backpropagation. The Temporal-Ensemble Experts module maintains several EMA classifier heads with different decay rates and it is combining them to balance plasticity and stability. Theoretically REAR is shown to approximate batch ridge regression, and TEE achieves an almost optimal bias–variance trade-off. Experiments on CIFAR-100, ImageNet-R, and CUB-200 under the Si-Blurry protocol demonstrate consistent state-of-the-art results with minimal trainable parameters and runtime overhead.

**Strengths:**

1. The paper identifies two core challenges in GCL a) expert routing and b)expert stability. it addresses each with a distinct, principled component: REAR and TEE.
2. REAR does not do backpropagation. Instead it maintains online sufficient statistics and is solving a closed-form ridge regression, which is much faster/simpler.
3. the authors show consistent state-of-the-art results on CIFAR-100, ImageNet-R, and CUB-200 under the Si-Blurry protocol.
4. I like the math analysis linking random-feature expansion to generalization (thm 1) and characterizing the bias–variance trade-off in temporal ensembling (thm 2)
5. Less than a million trainable parameters
6. I very much like the analogy to multi-timescale synaptic adaptation and biologically plausible interpretation of the design

**Weaknesses:**

1) Thm 1 relies on a pairwise concentration lemma but omits a full matrix-concentration argument and a margin assumption needed to link regression risk to routing accuracy. This may be fixable in the rebuttal period.

2)  Thm-2:  the derivation connecting EMA bias to temporal drift is approximate. The claim of “near-optimal adaptation” is not formally proven. Needs to be clarified

3) Despite the task-free framing that the paper emphasizes, in my opinion REAR initialization and label accumulation still assume known session starts and one-hot session indicators. True?

4) Maintaining and inverting the (M \times M) Gram matrix can be memory-intensive for large random expansions

5) The method’s logit mask may leak boundary information but this is not analyzed against Si-Blurry baselines such as MVP.

**Questions:**

1. plz clarify how REAR maintains the inverse of (G+ lambda I) online. Is inversion recomputed per batch or updated incrementally?

2. Is it that each sample contributes once to G and  Q or multiple times across the three online iterations per batch? If repeated, the estimator corresponds to weighted ridge regression - not the exact form proven in Lemma 3.

3. In Lemma 1 the jump from pairwise concentration to operator-norm bounds on \hat \Sigma - \Sigma is not rigorous. I think this is fixable though.

4. The paper states a random-feature error rate O(\log N/M) but this is  is inconsistent with the lemma’s \tilde O(\sqrt (\log N/M)) bound, right? Plz correct or provide an argument for the stronger rate.

5. Thm-1 gives a routing-accuracy guarantee  but it does not make any margin assumption between expert scores. Introduce this assumption and carry the margin constant into the final bound.

6. Thm-2 makes the bias step explicit. But can you show formally that \sum_j \alpha^j \Delta_{t-j+1} \le L P_t --  and quantify the constant C_2?

7. the claim that a geometric EMA bank achieves near-optimal performance should be supported with a short covering-ratio argument showing the error factor in terms of grid spacing r

9. If there is time in the rebuttal phase, plz evaluate the effect of removing or randomizing the logit mask to confirm that improvements are not due to boundary information.

10. Again, if there is time, it would be good to compare REAR with RanPAC under identical settings to clarify the unique contribution of routing versus analytic classification

11. I would appreciate a list of Assumptions (data i.i.d., λ > 0, bounded feature norms, fixed number of experts) at the start of the theory section

---

> ### Author Response · Authors · 2025-11-20
> **Response to Reviewer Vwq2**
>
> Thank you for your insightful comments. Below, we provide a point-to-point response and summarize the corresponding revisions.
>
> ### **W1: Thm-1, matrix-concentration arguments and margin assumptions**
>
> We appreciate this suggestion and agree that Thm-1 can be made fully self-contained. In the revised manuscript, we first add an explicit "Notation and Assumptions" block just before the theorem summarizing the data, feature, and margin assumptions used throughout the proof. We then replace the informal phrase "standard matrix concentration arguments" by a concrete matrix Bernstein bound on $\lVert\widehat{\Sigma}-\Sigma\rVert_{\mathrm{op}}$ derived from the random-feature lemma. We also extend the routing-accuracy corollary to explicitly assume a margin $\gamma>0$ between the correct expert's score and all others and to carry the resulting $(C_\varphi/\gamma)^2$ factor in the bound. These clarifications do not change the qualitative message of Thm-1 but make the argument more rigorous. The key technical steps are detailed in our responses to your **Q3–Q5** below.
>
> ### **W2: Thm-2, clarification of the near-optimal adaptation claim**
>
> For Thm-2 and the "near-optimal adaptation" claim, we have similarly expanded the proof to make every step explicit. In particular, we now first spell out the bias term rigorously by proving that $\sum_j \alpha^j \Delta_{t-j+1} \le L P_t$ and giving a concrete constant $C_2$ in the MSE bound, and then add a short covering-ratio argument showing that a geometric EMA bank with grid ratio $r$ satisfies $\min_i f(L_i) \le c(r) \min_L f(L)$ for $f(L)=C_1\zeta^2/L + C_2(LP_t)^2$ with $c(r)=\max(r^2,r)$. Together with the existing variance calculation, this formally justifies the phrase "near-optimal" for the EMA bank. We provide details in our responses to your **Q6–Q7** below.
>
> ### **W3: Session boundary information**
>
> As clarified in **CQ2**, FlyPrompt does not assume any privileged boundary information at either training or inference time, consistent with prior GCL methods such as MVP [1] and MISA [2].
> In the Si-Blurry benchmark, the "task" or "session" signal does not correspond to a disjoint distribution shift, particularly when the disjoint class ratio $r_D$ is low (e.g., $r_D \approx 0$). Under such settings, session boundaries are essentially decorrelated from actual changes in the data distribution.
> Therefore, using these boundaries as internal triggers (e.g., initializing a new expert after a fixed time window or sample budget) is entirely compatible with the task-free or blurry-boundary nature of GCL. This interpretation aligns with the protocol used by prior works and is further empirically validated by our experiments in CQ2 and original Appendix Table 6, where self-triggered expert updates achieve comparable or better performance than task-correlated setups.
>
> [1] Online Class Incremental Learning on Stochastic Blurry Task Boundary via Mask and Visual Prompt Tuning. ICCV, 2023.
>
> [2] Advancing Prompt-Based Methods for Replay-Independent General Continual Learning. ICLR, 2025.
>
> ### **W4: Cost of Gram matrix inversion**
>
> The cost of maintaining and inverting the Gram matrix in REAR is indeed tied to the random expansion dimension $M$. To balance performance and computational/storage cost, we conducted a detailed evaluation over various $M$ values (original Fig. 4). We found that $M = 10000$ provides a strong performance-efficiency tradeoff, with marginal gains beyond that. Importantly, the Gram matrix inverse $(G + \lambda I)^{-1}$ is computed **once at the end of training**, not during the training stream, so it does not add per-batch overhead. Additionally, approximate inversion methods (e.g., approximation or iterative solvers) can further reduce this cost if needed. Our runtime and model size results (original Table 5) empirically confirm that REAR introduces minimal overhead in practice.
>
> ### **W5: Usage of logit mask**
>
> The logit mask $\boldsymbol{m}$ in FlyPrompt is implemented for each incoming batch $(\boldsymbol{X}, \boldsymbol{y})$ as this:
> $$
> m_c = \begin{cases}
>     0 & \text{if class } c \text{ is in } \boldsymbol{y}, \\
>     -\infty & \text{otherwise}.
> \end{cases}
> $$
> Batch-seen class mask limits the cross-entropy loss calculated only on the classes that have been seen in the current batch, to mitigate the interference towards other unrelated classes. Thus, it does not leak any boundary information. As described in our original manuscript, such logit mask is adopted from MISA, and we generally apply it to all our baselines and FlyPrompt for a fair comparison. We have revised our manuscript to make this clearer (lines 304-307, 1747).

---

> > ### Author Response · Authors · 2025-11-20
> > **Response to Reviewer Vwq2 (part 2)**
> >
> > ### **Q1: How does REAR maintain $(G+\lambda I)^{-1}$ online?**
> >
> > REAR does not compute the matrix inverse $(G + \lambda I)^{-1}$ during the training stream. Instead, it incrementally accumulates the sufficient statistics (matrices $G$ and $Q$) after each batch, as defined in Eq. (2). The inverse is computed **only once upon evaluation**, making REAR significantly more efficient than per-batch gradient-based routing methods. We have made this clearer in the revised manuscript (lines 248–250, 1758-1759).
> >
> > ### **Q2: How does each sample contribute to G and Q?**
> >
> > Each sample contributes once to the $G$ and $Q$ matrices. For each incoming batch, we first perform three gradient updates to optimize the expert's prompt and output head parameters, similar to prior works such as MVP and MISA. After that, we accumulate the batch's statistics into $G$ and $Q$ without using gradients, in accordance with Eq. (2). This design ensures that the ridge regression optimization aligns with our theoretical analysis in Lemma 3 while maintaining online efficiency. We have made this clearer in the revised manuscript (lines 1746-1759).
> >
> > ### **Q3: Lemma 1, from pairwise concentration to operator-norm bounds**
> >
> > Thank you for pointing this out. In the current appendix Lemma 1 first establishes a uniform pairwise bound $\big|M^{-1}\boldsymbol{\varphi}(x_i)^\top\boldsymbol{\varphi}(x_j)-k(\boldsymbol h_i,\boldsymbol h_j)\big|\le\varepsilon$. In the revised manuscript, we have added a dedicated "Concentration of $\widehat{\Sigma}$ and $\widehat b$" paragraph that derives an operator-norm bound on the empirical feature covariance $\widehat{\Sigma}=\tfrac{1}{N}\Phi^\top\Phi$ and cross-covariance $\widehat b$.
> > Concretely, we write
> > $$\widehat{\Sigma}=\frac{1}{N}\sum_{i=1}^N \boldsymbol{\varphi}(x_i)\boldsymbol{\varphi}(x_i)^\top,\qquad \Sigma=\mathbb E[\boldsymbol{\varphi}(x)\boldsymbol{\varphi}(x)^\top],$$
> > and define centered self-adjoint matrices
> > $$X_i := \frac{1}{N}\Big(\boldsymbol{\varphi}(x_i)\boldsymbol{\varphi}(x_i)^\top - \Sigma\Big),\qquad \widehat{\Sigma}-\Sigma = \sum_{i=1}^N X_i.$$
> > Under the same embedding-boundedness and activation assumptions as Lemma 1, there exists a constant $C_\varphi>0$ (depending only on $(H, L_\sigma, C)$) such that $\lVert\boldsymbol{\varphi}(x)\rVert_2\le C_\varphi$ almost surely. It follows that
> > $$\lVert X_i\rVert_{\mathrm{op}}\le \tfrac{1}{N}\big(\lVert\boldsymbol{\varphi}(x_i)\boldsymbol{\varphi}(x_i)^\top\rVert_{\mathrm{op}}+\lVert\Sigma\rVert_{\mathrm{op}}\big)\le \tfrac{L_0}{N},$$
> > for some $L_0$ depending only on $(H,L_\sigma,C)$, and that the matrix variance term
> > $$v^2:=\Big\lVert\sum_{i=1}^N \mathbb E[X_i^2]\Big\rVert_{\mathrm{op}}\le \tfrac{V_0}{N}$$
> > for some $V_0$ depending on the same parameters. Applying the matrix Bernstein's inequality [1] with dimension $D=M$ then gives, for all $\varepsilon>0$,
> > $$\mathbb P\big(\lVert\widehat{\Sigma}-\Sigma\rVert_{\mathrm{op}}\ge \varepsilon\big)
> > \le 2M\exp\Big(-\frac{N\varepsilon^2/2}{V_0+L_0\varepsilon/3}\Big),$$
> > so by choosing $\varepsilon = C'\sqrt{\tfrac{\log(M/\delta)}{N}}$ we obtain
> > $$\lVert\widehat{\Sigma}-\Sigma\rVert_{\mathrm{op}} \le C'\sqrt{\tfrac{\log(M/\delta)}{N}}$$
> > with probability at least $1-\delta$. This is exactly the operator-norm concentration bound we now use in Lemma 2 and Thm-1 (lines 1061-1118, 1143-1148).
> >
> > [1] User-friendly tail bounds for sums of random matrices. *Foundations of Computational
> > Mathematics*, 2012.
> >
> > ### **Q4: Thm-1, clarification of the random-feature error bound**
> >
> > Thanks for pointing this out. Lemma 1 controls individual inner products with error $\varepsilon$, where the condition on $M$ implies $\varepsilon = \tilde O\big(\sqrt{\log(N/\delta)/M}\big)$. In the original manuscript, we then propagated this approximation through the squared-loss risk and summarized the resulting term as $O(\log(N/\delta)/M)$ without making the dependence on $\varepsilon$ explicit, which can be confusing. In the revised manuscript, we rectify the theorem statement so that the feature-approximation term is stated directly as $\mathcal E_{\mathrm{feat}}(M,\delta) = \tilde O\big(\sqrt{\log(N/\delta)/M}\big)$, matching Lemma 1 exactly. This revision tightens the bounds and does not affect empirical conclusions (lines 265-266, 968-970, 1149-1153).

---

> > > ### Author Response · Authors · 2025-11-20
> > > **Response to Reviewer Vwq2 (part 3)**
> > >
> > > ### **Q5: Thm-1, margin assumption between expert scores**
> > >
> > > We appreciate this suggestion and have made the margin assumption more explicit.
> > > Let $s_U(x)\in\mathbb R^T$ denote the expert-score vector (router outputs) produced by parameters $U$, and let $t^* (x)$ be the correct expert index. We assume that there exists a fixed margin $\gamma>0$ such that for the population minimizer $U^* $,
> > > $$s_{U^* }(x)\_{t^* } \ge  s_{U^* }(x)\_t + \gamma, \quad \forall t\neq t^* (x), \text{almost surely.}$$
> > > On the event $\\{\hat t(x)\neq t^* (x)\\}$, this margin condition implies
> > > $$\gamma \le s_{U^* }(x)\_{t^* (x)} - s_{U^* }(x)\_{\hat t(x)} \le 2\max_t \big|s_{U^* }(x)\_t - s_{\widehat U}(x)\_t\big|,$$
> > > so $\mathbf 1\\{\hat t(x)\neq t^* (x)\\} \le (2/\gamma)^2 \big\lVert s_{\widehat U}(x)-s_{U^* }(x)\big\rVert_2^2$. Using $s_U(x)=\boldsymbol{\varphi}(x)U^\top$ with $\lVert\boldsymbol{\varphi}(x)\rVert_2\le C_\varphi$ under our feature assumptions gives $\max_t|s_{U^* }(x)\_t-s_{\widehat U}(x)\_t|\le C_\varphi \lVert\widehat U-U^* \rVert_F$, hence
> > > $$\mathbb P(\hat t(X)\neq t^* (X)) \le \frac{4C_\varphi^2}{\gamma^2} \lVert\widehat U-U^* \rVert_F^2.$$
> > > Finally, recall that the regularized risk is defined as $\mathcal R(U):=\mathbb E\lVert s_U(X)-C\rVert_2^2+\lambda\lVert U\rVert_F^2$, so the quadratic ridge term $\lambda\lVert U\rVert_F^2$ makes $\mathcal R$ (at least) $\lambda$-strongly convex in $U$. Strong convexity implies $\mathcal R(\widehat U)-\mathcal R(U^* )\ge (\lambda/2)\lVert\widehat U-U^* \rVert_F^2$, and therefore $\lVert\widehat U-U^* \rVert_F^2 \le \tfrac{2}{\lambda}\big[\mathcal R(\widehat U)-\mathcal R(U^* )\big]$. Combining this with the previous inequality yields
> > > $$\mathbb P(\hat t(X)\neq t^* (X)) \le \frac{8C_\varphi^2}{\lambda\gamma^2} \big[\mathcal R(\widehat U)-\mathcal R(U^* )\big].$$
> > > In the revised manuscript, we state this margin condition immediately before Thm-1 and carry the factor $(C_\varphi/\gamma)^2$ explicitly (together with $1/\lambda$) in the routing-accuracy corollary, thereby making the connection between regression risk and misrouting probability rigorous (lines 947-958, 1154-1180).
> > >
> > > ### **Q6: Thm-2, Bias step and costant $C_2$**
> > >
> > > We agree that the bias step in Thm-2 should be fully explicit. Starting from
> > > $$\overline{\boldsymbol W}\_t^\star - \boldsymbol W_t^\star = \sum_{k\ge0} a_k \big(\boldsymbol W_{t-k}^\star - \boldsymbol W_t^\star\big), \quad a_k=(1-\alpha)\alpha^k,$$
> > > we decompose the differences into nearest-neighbor steps as in the appendix and obtain
> > > $$\big\lVert\overline{\boldsymbol W}\_t^\star - \boldsymbol W_t^\star\big\rVert \le \sum_{j\ge1}\Big(\sum_{k\ge j} a_k\Big) \Delta_{t-j+1}, \quad \Delta_u = \big\lVert\boldsymbol W_u^\star - \boldsymbol W_{u-1}^\star\big\rVert.$$
> > > Since $\sum_{k\ge j} a_k=(1-\alpha)\sum_{k\ge j}\alpha^k = \alpha^j$, this yields
> > > $$\big\lVert\overline{\boldsymbol W}\_t^\star - \boldsymbol W_t^\star\big\rVert \le \sum_{j\ge1} \alpha^j \Delta_{t-j+1}.$$
> > > Let $\gamma=\alpha$ in the discounted path length $P_t = \sum_{j\ge1} \gamma^{j-1} \Delta_{t-j+1}$, we have
> > > $$\sum_{j\ge1} \alpha^j \Delta_{t-j+1} = \alpha \sum_{j\ge1} \alpha^{j-1}\Delta_{t-j+1} = \alpha P_t \le L P_t,$$
> > > where $L=1/(1-\alpha)\ge\alpha$. After squaring and using $\lVert a+b\rVert^2\le2\lVert a\rVert^2+2\lVert b\rVert^2$, this shows that the bias contribution in the MSE bound is at most a constant multiple of $(L P_t)^2$, so one can take $C_2$ to be an explicit constant (e.g., $C_2=2$) in Thm-2. While, if we allow $\gamma$ to differ slightly from $\alpha$, the constant $C_2$ becomes $(\alpha/\gamma)^2$ (bounded if we restrict $\gamma\in[(1-\epsilon)\alpha,(1+\epsilon)\alpha]$). We have added it in the revised manuscript (line 1194- 1201, 1240-1246, 1256-1271).
> > >
> > > ### **Q7: Thm-2, covering-ratio argument for near-optimal claim**
> > >
> > > We now include the covering-ratio argument explicitly. Let $f(L)=A/L+B(LP_t)^2$ with $A\asymp\zeta^2$ and $B\asymp1$, and let $L^\star$ minimize $f$. A short calculus computation gives $L^\star = (A/(2B P_t^2))^{1/3}$ and $f(L^\star) = \Theta\big(B(L^\star P_t)^2\big)$. If we maintain a geometric grid $\\{L_i\\}\_{i=1}^m$ with ratio $r>1$ (e.g., $L_i=r^{i-1}$), then for any such $L^\star$ there exists an index $i_t$ such that $L_{i_t}\in[L^\star/r, rL^\star]$. Writing $L=L^\star\eta$ with $\eta\in[1/r,r]$ and using the optimality condition $A/L^\star = 2B(L^\star P_t)^2$, we obtain
> > > $$\frac{f(L)}{f(L^\star)} = \frac{2B(L^\star P_t)^2/\eta + \eta^2B(L^\star P_t)^2}{3B(L^\star P_t)^2} = \frac{2/\eta+\eta^2}{3}.$$
> > > The right-hand side is maximized over $\eta\in[1/r,r]$ at one of the endpoints, and a simple bound yields
> > > $$\sup_{\eta\in[1/r,r]}\frac{2/\eta+\eta^2}{3}\le \max\left(\frac{2}{3r}+\frac{r^2}{3}, \frac{2r}{3}+\frac{1}{3r^2}\right) \le \max(r^2,r) =: c(r).$$
> > > Taking $L=L_{i_t}$ in the display above, we conclude $f(L_{i_t})\le c(r) f(L^\star)$. In the revision, we make the explicit choice $c(r)=\max(r^2,r)$ in the statement after Thm-2 (lines 1219-1221, 1271-1290).

---

> > > > ### Author Response · Authors · 2025-11-20
> > > > **Response to Reviewer Vwq2 (part 4)**
> > > >
> > > > ### **Q8: Ablation of the logit mask**
> > > >
> > > > | Method/Mask Type | No Mask                  | Random Mask              | Seen-Class Mask          | Batch Seen-Class Mask    |
> > > > | ---------------- | ------------------------ | ------------------------ | ------------------------ | ------------------------ |
> > > > | L2P              | $62.74_{\pm 4.39}$       | $62.46_{\pm 4.54}$       | $62.22_{\pm 4.39}$       | $76.23_{\pm 2.73}$       |
> > > > | DualPrompt       | $66.68_{\pm 5.25}$       | $65.48_{\pm 4.45}$       | $65.29_{\pm 4.62}$       | $76.04_{\pm 3.32}$       |
> > > > | CODA-Prompt      | $66.15_{\pm 5.22}$       | $65.58_{\pm 5.23}$       | $65.63_{\pm 5.40}$       | $79.13_{\pm 3.06}$       |
> > > > | MVP              | $68.18_{\pm 4.85}$       | $67.75_{\pm 4.95}$       | $67.72_{\pm 4.87}$       | $67.74_{\pm 4.96}$       |
> > > > | MISA             | $69.85_{\pm 3.73}$       | $68.23_{\pm 3.82}$       | $68.34_{\pm 3.90}$       | $80.35_{\pm 2.39}$       |
> > > > | FlyPrompt        | ${\bf 78.73}_{\pm 3.55}$ | ${\bf 78.32}_{\pm 3.48}$ | ${\bf 78.75}_{\pm 3.52}$ | ${\bf 83.24}_{\pm 2.23}$ |
> > > >
> > > > > Performance is measured by $A_{\rm{auc}}$ (%, mean±std) over 5 random seeded runs on CIFAR-100.
> > > >
> > > > As clarified in **W5**, the logit mask $\boldsymbol{m}$ we apply **suppresses logits of unseen classes within the current batch**, without leaking any future or boundary information. To further validate its effect, we conducted an **ablation study** comparing four masking strategies across multiple GCL methods, including FlyPrompt.
> > > >
> > > > Mask variants are constructed as follows:
> > > > (1) **No Mask**, standard softmax over all output classes.
> > > > (2) **Random Mask**, for each sample $(\boldsymbol{x}, y)$, set $m_y = 0$ and assign $m_c = 0$ or $-\infty$ randomly with 0.5 probability for each previously seen class $c \ne y$.
> > > > (3) **Seen-Class Mask**, setting $m_c = 0$ for all previously seen classes, $-\infty$ otherwise.
> > > > (4) **Batch Seen-Class Mask** (used in our work), setting $m_c = 0$ only for the classes present in the current batch $\boldsymbol{y}$.
> > > >
> > > > As shown in the above table, **our batch seen-class mask consistently improves performance**, particularly for methods like MISA, CODA-Prompt, and FlyPrompt. This supports its role in reducing class interference without violating GCL assumptions. Notably, MVP shows minimal change across masks, as it already implements an internal learnable logit masking mechanism.
> > > >
> > > > We have added this ablation study and clarified the implementation in the revised manuscript (lines 1593–1619).
> > > >
> > > > ### **Q9: Comparison with RanPAC**
> > > >
> > > > | Method     | CIFAR-100 $A_{\rm{auc}}$ | CIFAR-100 $A_{\rm{last}}$ | ImageNet-R $A_{\rm{auc}}$ | ImageNet-R $A_{\rm{last}}$ | CUB-200 $A_{\rm{auc}}$   | CUB-200 $A_{\rm{last}}$  |
> > > > | ---------- | ------------------------ | ------------------------- | ------------------------- | -------------------------- | ------------------------ | ------------------------ |
> > > > | RanPAC$^†$ | $69.91_{\pm 3.88}$       | $79.92_{\pm 0.07}$        | $47.14_{\pm 2.18}$        | $50.75_{\pm 2.15}$         | $60.18_{\pm 5.52}$       | $66.21_{\pm 6.15}$       |
> > > > | RanPAC$^‡$ | $57.35_{\pm 8.23}$       | $77.65_{\pm 0.21}$        | $36.90_{\pm 4.17}$        | $44.39_{\pm 0.11}$         | $64.52_{\pm 8.23}$       | $71.65_{\pm 0.17}$       |
> > > > | RanPAC$^*$ | $77.88_{\pm 4.28}$       | $86.52_{\pm 1.15}$        | $53.18_{\pm 2.22}$        | $54.71_{\pm 2.48}$         | $69.64_{\pm 3.89}$       | $72.30_{\pm 1.09}$       |
> > > > | FlyPrompt  | ${\bf 83.24}_{\pm 2.23}$ | ${\bf 86.76}_{\pm 0.73}$  | ${\bf 56.58}_{\pm 1.47}$  | ${\bf 55.27}_{\pm 0.91}$   | ${\bf 70.64}_{\pm 2.85}$ | ${\bf 73.40}_{\pm 1.88}$ |
> > > >
> > > > > Performance is measured by $A_{\rm{auc}}$ and $A_{\rm{last}}$ (%, mean±std) over 5 random seeded runs.

---

> > > > > ### Author Response · Authors · 2025-11-20
> > > > > **Response to Reviewer Vwq2 (part 5)**
> > > > >
> > > > > RanPAC is originally designed for **offline class-incremental learning**, where the pretrained model is fine-tuned on the first task and frozen thereafter. During this process, features for all first-task samples are retrieved again and stored to compute a stable Gram matrix for regression after the finetuning. However, this setup is **incompatible with GCL of online data streams and blurry task boundaries**, where each sample of randomly sampled classes can only be seen once.
> > > > >
> > > > > To ensure fair comparison, we adapt RanPAC into three variants that meet GCL constraints:
> > > > > (1) **RanPAC$^\dagger$** fine-tunes the PTM on the first task without storing features (used in our original Table 2 as the analytic classifier baseline with random-projection).
> > > > > (2) **RanPAC$^\ddagger$** freezes the PTM during the first task and stores all features.
> > > > > (3) **RanPAC$^*$** fine-tunes and collects features simultaneously during the first task.
> > > > >
> > > > > Note that in RanPAC$^*$, although all features are collected, the representation is unstable due to ongoing fine-tuning, resulting in an **ill-conditioned Gram matrix**. As shown in the above table, FlyPrompt **consistently outperforms all RanPAC variants across datasets**, highlighting the robustness of our analytic expert router. We have included them in the revised manuscript (lines 254–260, 406-407, 851-863, 1442-1451).
> > > > >
> > > > > ### **Q10: List of assumptions**
> > > > >
> > > > > Thank you for the valuable suggestion. All assumptions used in our proofs are already listed in the appendix, but we agree that making them visible earlier would improve readability. In the revised manuscript, we have added a short "Notation and Assumptions" paragraph at the start of the theory section and just before the formal statements of Thm-1 and Thm-2. This block adds: (1) that training samples $(x_i,c_i)$ are drawn i.i.d. at the analysis timescale; (2) that the pretrained embedding satisfies $\lVert f_{\boldsymbol\theta}(x)\rVert\le H$; (3) that the activation $\sigma$ is $L_\sigma$-Lipschitz with linear growth; (4) that the ridge parameter satisfies $\lambda>0$; (5) that the number of experts $T$ is finite and fixed during each run; and (6) that for the routing-accuracy corollary we assume a fixed margin $\gamma>0$ between the correct expert's score and all others, as detailed in **Q5**. We believe this explicit list will make the scope of Thm-1 and Thm-2 clearer without changing any technical content (lines 922-951, 1183-1207).

---

> ### Comment · Reviewer_Vwq2 · 2025-11-23
>
> After reading the four other reviews, I realize that I was probably too excited after reading the paper (maybe because I am biased in favor of neuro-inspired approaches). The other reviews however identify several weaknesses (none of them "fatal" or "unfixable" in my opinion) that I had missed. So I reduce my score from 10 to 8 -- but I am still quite positive about this paper and I hope it will be finally accepted.

---

> > ### Author Response · Authors · 2025-11-24
> >
> > Dear Reviewer Vwq2,
> >
> > Thank you for your time and valuable feedback. We have carefully considered your suggestions and revised the manuscript to address the highlighted areas. Your insights have strengthened the paper, and we deeply appreciate your constructive engagement throughout the review process. Your encouragement and positive outlook have been highly motivating, and we welcome any further discussions or suggestions to refine the manuscript.
> >
> > Sincerely,
> >
> > Authors of FlyPrompt.

---

### Author Response · Authors · 2025-11-20
**Overall Response**

Dear reviewers,

We sincerely thank all reviewers for their great efforts and insightful comments. We are encouraged that many reviewers have acknowledged the strengths of our work, including: its principled decomposition of General Continual Learning (GCL) (Reviewers Vwq2, jMmN, Yd6v, yU5J); the novel efficiency of its backprop-free random expanded analytic router paired with a robust temporal-ensemble expert design (Reviewers Vwq2, yU5J); consistent state-of-the-art performance across datasets under the challenging GCL setting (Reviewers Vwq2, jMmN, EcKY); and the conceptual depth added by cross-disciplinary inspiration from biological learning systems (Reviewers Vwq2, Yd6v).

> **Numbering convention.** Unless explicitly specified with "in the revised manuscript", direct references such as "Table X", "Fig. Y", or "Appendix Z" in this rebuttal refer to the version of the manuscript originally submitted for review. Because new results and analyses have been added in the revised manuscript, some table/figure indices may shift slightly, but the corresponding content remains the same or is extended. Line numbers refer to the latest revised version.

Below, we address two major common questions (CQs) raised across the reviews.

### **CQ1: Contribution and Novelty**

We summarize the contribution and novelty of our work as follows.

**Conceptually**, FlyPrompt is inspired by key mechanisms of the fruit fly's learning system (i.e., the mushroom body):
(1) sparse, high-dimensional random expansion from PNs to KCs, which supports instance-level routing and interference reduction; and (2) compartmentalized plasticity across KC subtypes, enabling adaptation at multiple timescales.

Prior work has explored each mechanism separately (e.g., random expansion for interference reduction [1,2] and multi-timescale ensembles for memory consolidation and generalization [3,4]). To our knowledge, FlyPrompt is the first attempt to **integrate both mechanisms** in a principled framework for the realistic GCL problem, especially with **parameter-efficient tuning (PET)** of **pretrained models**. We formalize GCL as a streaming multi-expert problem, decompose it into two subproblems (expert routing and expert competence improvement) and map these to algorithmic counterparts (REAR and TE²) grounded in the fly brain's structure and function.

**Algorithmically**, our instantiations (REAR and TE²) depart meaningfully from prior CL techniques: (1) REAR uses random expansion and ridge regression **only for routing**, unlike methods such as ACIL or RanPAC that apply analytic solutions to final classification. This allows expert modules to remain plastic, decoupling a fixed, forward-only router from gradient-trained, adaptable experts. (2) TE² introduces a bank of temporally smoothed heads per expert, which are combined during inference. Unlike single-head EMA models, TE² ensures that **a near-optimal head** always exists under representation shifts, as supported by our theoretical analysis.

Empirically, our new experiments with multiple RanPAC variants show that even carefully adapted analytic classifiers lag behind FlyPrompt and suffer from ill‑conditioned Gram matrices once the backbone is updated online (see **Vwq2 Q9**), while ablations that plug REAR and TE² into CL baselines such as S‑Prompt++, HiDe‑Prompt and NoRGa consistently improve their GCL performance (see **EcKY W4**).

In summary, FlyPrompt (1) identifies and formalizes two core subproblems in GCL; (2) offers theoretically grounded, biologically inspired solutions; and (3) implements these on top of pretrained models under replay-free GCL settings, yielding **robust and state-of-the-art performance**.

[1] A neural algorithm for a fundamental computing problem. *Science*, 2017.

[2] Cellular-resolution population imaging reveals robust sparse coding in the Drosophila mushroom body. *Journal of Neuroscience*, 2011.

[3] The neuronal architecture of the mushroom body provides a logic for associative learning. *eLife*, 2014.

[4] Incorporating neuro-inspired adaptability for continual learning in artificial intelligence. *Nature Machine Intelligence*, 2023.

---

> ### Author Response · Authors · 2025-11-20
> **Overall Reponse (part 2)**
>
> ### **CQ2: How Does FlyPrompt Address the GCL Setting?**
>
> Our primary objective is to strictly adhere to the standard definition of GCL. Following DER [1], the assumptions for GCL data streams include: *no task boundaries during training* and *no task oracle at test time*. The Si-Blurry benchmark [2] we adopt has been carefully evaluated in MVP [3] and satisfies the GCL's defining properties proven by MISA [4], including *variable class counts across sessions*, *class recurrence across sessions*, and *arbitrary sample counts per class*. Controlling the disjoint class ratio $r_D$ and blurry sample ratio $r_B$ allows Si-Blurry to generate diverse, valid GCL streams. Notably, in the extreme case when $r_D$ is small, task/session labels become decoupled from distributional changes, yet FlyPrompt still achieves strong performance (original Appendix Table 6).
>
> In this context, the "task" or "session" index is used only as *a conceptual nominal signal* to describe the benchmark generation process. Because Si-Blurry randomizes class assignments and sample counts across sessions, it is equally valid to trigger expert changes internally (e.g., every $N$ samples) without access to the task index. In fact, this is functionally equivalent under the benchmark's assumptions.
>
> Additionally, FlyPrompt follows the same protocol as recent prompt-based GCL methods such as MVP and MISA, which also maintain session-wise prompt or expert structures. The implementation of expert/task indices is similar to prior work. Notably, FlyPrompt does not require prior knowledge of the total number of sessions: routing structures such as the $Q$ matrix and router head can be dynamically extended (e.g., from $T$ to $T+1$) with zero-padding, similar to adding new classes in standard classifiers.
>
> | # of Tasks | # of Experts         | MVP                | MISA               | FlyPrompt          |
> | ---------- | -------------------- | ------------------ | ------------------ | ------------------ |
> | 5          | 5 (task-correlated)  | $67.74_{\pm 4.96}$ | $80.35_{\pm 2.39}$ | $83.24_{\pm 2.23}$ |
> | 5          | 5                    | $67.75_{\pm 4.96}$ | $80.60_{\pm 2.06}$ | $83.67_{\pm 2.23}$ |
> | 5          | 10                   | $67.23_{\pm 5.06}$ | $79.95_{\pm 1.86}$ | $83.40_{\pm 2.28}$ |
> | 5          | 20                   | $67.19_{\pm 4.80}$ | $79.94_{\pm 1.75}$ | $82.26_{\pm 1.94}$ |
> | 10         | 10 (task-correlated) | $58.23_{\pm 3.42}$ | $75.71_{\pm 3.10}$ | $77.65_{\pm 3.05}$ |
> | 10         | 5                    | $58.49_{\pm 3.57}$ | $76.25_{\pm 3.10}$ | $77.28_{\pm 2.79}$ |
> | 10         | 10                   | $58.19_{\pm 3.42}$ | $75.69_{\pm 3.11}$ | $76.96_{\pm 3.23}$ |
> | 10         | 20                   | $58.46_{\pm 3.34}$ | $75.73_{\pm 2.80}$ | $76.47_{\pm 3.38}$ |
> | 20         | 20 (task-correlated) | $56.52_{\pm 3.20}$ | $73.96_{\pm 0.72}$ | $75.87_{\pm 1.93}$ |
> | 20         | 5                    | $56.59_{\pm 3.34}$ | $74.27_{\pm 0.97}$ | $76.12_{\pm 1.57}$ |
> | 20         | 10                   | $56.35_{\pm 3.38}$ | $73.86_{\pm 0.82}$ | $76.12_{\pm 1.21}$ |
> | 20         | 20                   | $56.48_{\pm 3.15}$ | $73.66_{\pm 0.85}$ | $75.36_{\pm 1.65}$ |
>
> > Performance is measured by $A_{\rm{auc}}$ (%, mean±std) over 5 random seeded runs on CIFAR-100. "# of Tasks", the number of sessions defined by Si-Blurry. "# of Experts", the number of experts maintained by each method according to different budget sizes.

---

> > ### Author Response · Authors · 2025-11-20
> > **Overall Response (part 3)**
> >
> > To further demonstrate that FlyPrompt and comparable GCL baselines **do not depend on task boundary information** in Si-Blurry, we conducted an empirical study using a self-triggered expert allocation mechanism. In this setup, each GCL method maintains a fixed sample budget, freezing the current expert and initializing the next whenever the number of observed samples reaches a predefined threshold.
> > This fixed-length time window enables internal management of expert updates, completely decoupled from external task segmentation. This is especially important in Si-Blurry, where the number of samples per task is randomized and therefore provides no consistent signal about distributional change.
> > We evaluated varying GCL task counts (5, 10, and 20) and sample budget lengths (10000, 5000, and 2500), corresponding to sequentially initializing 5, 10, and 20 experts, respectively, on the CIFAR-100 dataset (50000 total samples).
> > Results show that self-triggered expert initialization performs on par with, or even slightly better than, setups where experts are managed based on task-aligned session indices. This holds across all tested methods (FlyPrompt, MVP, and MISA) and expert counts.
> >
> > These results confirm that the training process on Si-Blurry does not violate GCL's key assumptions. Moreover, task-switching signals offer no measurable advantage in this setup, ensuring that no unfair performance boost arises from using nominal task/session labels.
> >
> > [1] Dark Experience for General Continual Learning: a Strong, Simple Baseline. NeurIPS, 2020.
> >
> > [2] Online Continual Learning on Class Incremental Blurry Task Configuration with Anytime Inference. ICLR, 2022.
> >
> > [3] Online Class Incremental Learning on Stochastic Blurry Task Boundary via Mask and Visual Prompt Tuning. ICCV, 2023.
> >
> > [4] Advancing Prompt-Based Methods for Replay-Independent General Continual Learning. ICLR, 2025.
> >
> > ------
> >
> > We once again thank the reviewers for their constructive insights and valuable feedback. A detailed point-to-point response addressing all identified **Weaknesses (W)** and **Questions (Q)** will follow. We have also uploaded a **revised manuscript** that incorporates these reflections, including new results, additional analyses, and clarifications. All key updates are **marked in blue** for ease of reference.

---

### Author Response · Authors · 2025-11-29
**Summary of Rebuttal Consensus before Data Leakage**

Dear Area Chair,

We hope this message finds you well. We sincerely thank you for accepting the assessment of our submission under these challenging circumstances. We recognize the additional time and effort required to evaluate the reverted discussion threads and highly appreciate your commitment to maintaining the integrity of the review process.

Therefore, we wish to provide a brief and factual summary of the discussion and resolution status prior to the November 27th identity leak incident, to serve as a reference for your assessment. Our rebuttal successfully addressed the concerns raised by the reviewers, and this effort resulted in **positive and constructive initial feedback** from four out of five reviewers, including two explicit score increases. **These positive changes and comments, with specific dates explicitly recorded in reviewer comments, were all documented well before the discussion period was halted.** Reviewer Yd6v had not yet posted a follow-up comment.

| Reviewer | Initial Score | Post-rebuttal Score     | Response                                                    |
|----------|---------------|-------------------------|-------------------------------------------------------------|
| **Vwq2** | **10**        | **8** (24 Nov, 04:57)   | Maintained strong positive stance, and hope for acceptance. |
| **jMmN** | **4**         | **6** (21 Nov, 10:01)   | Confirmed all main concerns were successfully addressed.    |
| **Yd6v** | **6**         | **6**                   | Discussion was concluded before they could respond.         |
| **yU5J** | **6**         | **6** (23 Nov, 20:44)   | Found additional explanations and experiments convincing.   |
| **EcKY** | **2**         | **6** (20 Nov, 18:16)   | Confirmed all main concerns were largely addressed.         |

This consensus was achieved after we provided extensive clarifications and new experiments on key points, including:

1. **The clarification of the General Continual Learning (GCL) setting** by providing new empirical evidence that our method operates independently of explicit task boundary signals;
2. **The strengthening of our theoretical foundations**, including fixing bounds, adding explicit margin assumptions, and formally justifying the near-optimal bias control;
3. **The detailed analysis of each component's efficiency and contribution** against computational overheads and similar analytic classification and temporal ensemble baselines;
4. **The highlighting of methodological novelty** of integrating the brain-inspired sparse routing and temporal ensemble mechanisms into a single, principled framework for GCL.

We have uploaded a revised manuscript that incorporates all the clarifications, new baselines, and theoretical improvements discussed above. We believe the positive consensus achieved during the discussion period, well before the incident, strongly supports the merit of our submission.

Thank you once more for your diligent work in ensuring a fair review process. If you have any questions or require additional information, we are happy to provide further details.

Best regards,

The Authors

---

### Meta-Review · Area_Chair_EPKU · 2026-01-07

**Summary:**

This paper studies the general continual learning problem, which aims to equip intelligent systems with the ability to learn continuously from non-stationary, single-pass data streams, where tasks may not have clear boundaries and can evolve over time. In order to address the general continual learning problem, the authors propose FlyPrompt, a brain-inspired framework that decomposes the general continual learning problem into two subproblems: expert routing and expert competence improvement. The proposed method then introduces a randomly expanded analytic router for instance-level expert activation and a temporal ensemble of output heads to dynamically adapt decision boundaries over time. This paper was reviewed by five expert reviewers and received mixed initial ratings: three acceptances and two rejections. During the rebuttal period, the authors provided solid responses, and all reviewers confirmed their concerns, such as comparisons of parameter counts and task boundary of the general continual learning setup, were fully addressed during the rebuttal stage. All reviewers were on the positive side after reading the authors' responses. Therefore, it is a clear acceptance.

**Reviewer Concerns:**

During the rebuttal period, the authors provided solid responses, and all reviewers confirmed their concerns, such as comparisons of parameter counts and task boundary of the general continual learning setup, were fully addressed during the rebuttal stage.

**Reviewer Scores:**

This paper was reviewed by five expert reviewers and received mixed initial ratings: three acceptances and two rejections. During the rebuttal period, the authors provided solid responses, and all reviewers confirmed their concerns, such as comparisons of parameter counts and task boundary of the general continual learning setup, were fully addressed during the rebuttal stage. All reviewers were on the positive side after reading the authors' responses. Therefore, it is a clear acceptance.

---

### Decision · Program_Chairs · 2026-01-26

Accept (Poster)